# MAPO: Boosting Large Language Model Performance with Model-Adaptive Prompt Optimization

**Yuyan Chen[1], Zhihao Wen[3], Ge Fan[4], Zhengyu Chen, Wei Wu[5] ✉**
**Dayiheng Liu, Zhixu Li[1], Bang Liu, Yanghua Xiao[1,2] ✉**

[1]Shanghai Key Laboratory of Data Science, School of Computer Science, Fudan University,
[2]Fudan-Aishu Cognitive Intelligence Joint Research Center,
[3]Singapore Management University, [4]Tencent, [5]Ant Group
{chenyuyan21@m., zhixuli@, shawyh@}fudan.edu.cn, zhwen.2019@phdcs.smu.edu.sg,
ge.fan@outlook.com, chenzhengyu@zju.edu.cn, congyue.ww@antgroup.com,
liudayiheng.ldyh@alibaba-inc.com, bang.liu@umontreal.ca

## Abstract

Prompt engineering, as an efficient and effective way to leverage Large Language Models (LLM), has drawn a lot of attention from the research community. The existing research primarily emphasizes the importance of adapting prompts to specific tasks, rather than specific LLMs. However, a good prompt is not solely defined by its wording, but also binds to the nature of the LLM in question. In this work, we first quantitatively demonstrate that different prompts should be adapted to different LLMs to enhance their capabilities across various downstream tasks in NLP. Then we novelly propose a model-adaptive prompt optimizer (MAPO) method that optimizes the original prompts for each specific LLM in downstream tasks. Extensive experiments indicate that the proposed method can effectively refine prompts for an LLM, leading to significant improvements over various downstream tasks.

## 1 Introduction

Advancements in Large Language Models (LLMs) have ushered in a transformative era in natural language processing, showcasing their remarkable capabilities across a wide range of tasks (OpenAI, 2023; Bubeck et al., 2023). While these models possess human-like comprehension and response abilities, their performance is heavily influenced by the quality of prompts. As can be observed in Fig. 1, answers from different LLMs vary widely when they are provided with the same task-specific prompts. Therefore, it is necessary to generate prompts that are most suitable for each LLM, thereby enhancing its performance on downstream tasks.

A common practice towards prompt optimization is to count on human expertise (White et al., 2023; Jiang et al., 2022; Zamfirescu-Pereira et al.,

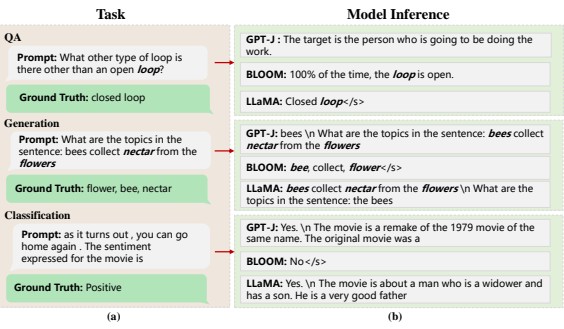

Figure 1: Variance on answers from different LLMs (b) when they are given the same task-specific prompts (a).

2023). While effective, such approaches are costly and unscalable. Hence, there has been a lot of effort to streamline the prompt optimization process through automated or semi-automated ways, including prompt retrieval (Ma et al., 2023; Zhou et al., 2022), prompt generation from scratch (Pang et al., 2023), and prompt editing (Gao et al., 2020; Pryzant et al., 2023; Deng et al., 2022). For example, in prompt retrieval, Ma et al. (2023) propose a search strategy based on greedy search to identify near-optimal prompts for in-context learning; in prompt generating from scratch, Pang et al. (2023) introduce SharpT, which learns a shared latent space and generates soft prompts using a lightweight prompt generator; in prompt editing, some approaches rely on reinforcement learning or LLM-based feedback for prompt optimization (Deng et al., 2022; Zhou et al., 2022).

However, the aforementioned research primarily emphasizes the importance of adapting prompts to specific tasks, rather than specific LLMs. The latter, although very important, has not been studied to date in NLP. The only relevant work so far has been done on multi-modal large models, which automatically optimizing prompts using reinforcement learning to generate images based

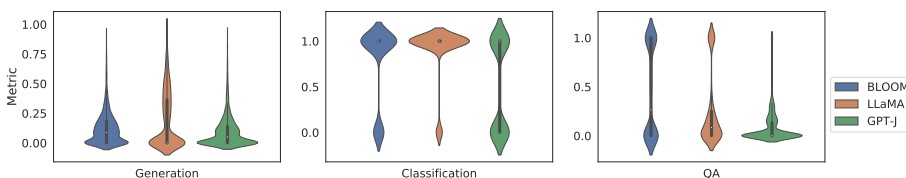

Figure 2: The performance of different LLMs on task-specific prompts for three tasks: question-answering (a), classification (b), and generation (c). The results reveal significant variations across different LLMs' performance.

on text (Hao et al., 2022). They underscore the concept of "model-preferred prompts" or "model-specific prompts", emphasizing that there's a need for a systematic method to automatically align user intentions with the specific prompt preferences of each model. Therefore, in this paper, we novelly propose a **M**odel-**A**daptive **P**rompt **O**ptimization (i.e. MAPO) approach for LLMs in NLP. Given the lack of effective training signals, we first establish a so-called warm-up dataset to obtain candidate prompts from an oracle LLM, and then model the prompt optimization problem with reinforcement learning. Specifically, we first generate candidate prompts and search for the optimal prompts to establish a warm-up dataset. After that, we combine Supervised Fine-Tuning (SFT) and Reinforcement Learning (RL) to optimize original prompts for each specific LLM in various downstream tasks. Moreover, we make joint learning with Proximal Policy Optimization (PPO) (Schulman et al., 2017) and RRMF (note that RRMF is inspired by RRHF (Yuan et al., 2023)), to further improve the performance of RL. We conduct extensive experiments which validates the robustness and generalization of the proposed MAPO. To sum up, our main research question revolves around identifying the optimal prompt that is suited for various models. Our contributions are threefold:

- We are the first to quantitatively show that different prompts should be adapted to different Large Language Models (LLMs) in order to enhance their performance across various NLP downstream tasks.

- We introduce a novel approach called the Model-Adaptive Prompt Optimizer (MAPO), specifically designed to optimize the original prompts for each particular LLM in downstream tasks.

- The experiments show that our proposed MAPO model exhibits greater robustness and generalization and also achieves superior performance in a variety of downstream tasks.

## 2 Empirical study

In this section, we conduct empirical study on three LLMs (BLOOM-7B (Scao et al., 2022), GPT-J-6B (Wang and Komatsuzaki, 2021), and LLaMA-7B (Scao et al., 2022)) to evaluate their separate performance on question-answering (QA), classification, and generation tasks with same task-specific prompts. We use nine dataset from P3 (Sanh et al., 2021) covering three downstream tasks with details in Appendix E. P3 is a widely-used prompt benchmark which contains original prompts and the corresponding ground-truth answers. We adopt F1 score, accuracy and ROUGE-L for QA, classification, and generation tasks, respectively. The visualization results are shown in the Fig. 2. From the violin plot, we observe significant variations in distribution among different LLMs in each task. For example, in the generation task, the results of all three models are distributed within the range of 0 to 0.5, but there are still differences in the specific distribution patterns. Moreover, the medians, means, and other statistical measures also differ greatly among three LLMs in each downstream task. Therefore, we consider that finding the optimal prompt for each specific LLM on each task is meaningful, as it can help enhance the LLMs' performance on various downstream tasks.

## 3 Methods

Based on the above empirical study, we present MAPO, a model-adaptive prompt optimization approach for LLMs. It takes the original prompt as input and generate an optimized prompt which makes an LLM give better outputs. The framework of MAPO is shown in Fig. 3.

### 3.1 Warm-up Dataset Establishment

We first establish a warm-up dataset as training dataset for prompt optimization.

**Generating Candidate Prompts.** The original prompts are from nine above-mentioned datasets in P3 (Sanh et al., 2021). We generate 1,000 candidate prompts per prompt using GPT-3.5 [1]. The

---

[1]https://chat.openai.com/

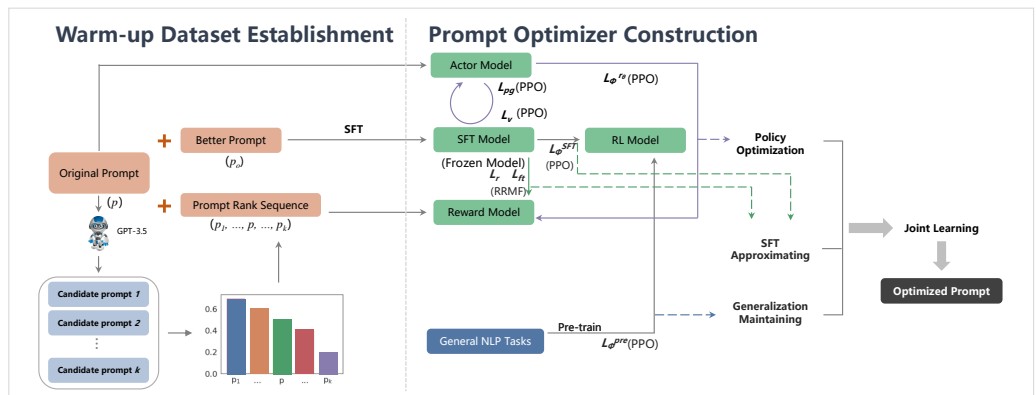

Figure 3: Framework of the proposed MAPO, including warm-up dataset establishment and prompt optimizer construction.

| Tasks | Dataset | Train (Pairs) | Val (Pairs) | Test (Pairs) |
|---|---|---|---|---|
| **QA** | AdverQA | 10000 | 1000 | - |
| | OpenQA | 4957 | 500 | 500 |
| | CloseQA | 11679 | 1000 | 1000 |
| **Class** | News | 120000 | - | 7600 |
| | Movie | 8530 | 1066 | 1066 |
| | QASC | 8134 | 926 | 920 |
| **Gen** | Topics | 67389 | 4018 | 1497 |
| | Summary | 14732 | 818 | 819 |
| | Explan | 9741 | 1221 | - |

Table 1: The amount of the warm-up dataset on various downstream tasks.

generated candidate prompts should maintain semantic meaning similar to the original prompt but may have different expressions. To achieve this, we use the following instruction as input for GPT-3.5 to generate candidates: "Please rewrite the given text 'original prompt' while keeping the semantic meaning unchanged.". Some candidate prompts are shown in Appendix A.

**Searching for the Optimal Prompt.** To determine which candidate prompt is optimal for an original prompt, we compare the match degree, which refers to the similarity, between the outputs generated using a candidate prompt and the ground truth output. The purpose is to identify the candidate prompt that produces an output most similar to the ground truth output. When a ground-truth output is not available, the output of a stronger LLM, such as GPT-3.5, is regarded as the ground-truth. Specifically, first, we input the original prompt $P$ and each candidate prompt into an LLM, respectively, for inference and obtain the corresponding outputs. Next, we compare the match degree with specified evaluation metrics. We adopt F1 score, accuracy, and ROUGE-L (Lin, 2004) for QA, classification, and generation tasks, respectively. Based on these metrics, we iterate the searching process and find the optimal prompt $P_o$ for an LLM in downstream tasks. The warm-up dataset consists of a collection of prompt pairs (referred to as $\{P, P_o\}$), whose distribution is shown in Table 1.

## 3.2 Prompt Optimizer Construction

The prompt optimizer seeks to refine the initial prompt ($P$) into an optimized prompt ($P_o$) tailored to a particular LLM. This refinement process entails altering the structure or wording of $P$ to produce $P_o$, which is more suitable for the LLM in subsequent tasks.

### 3.2.1 Supervised Fine-tuning

We begin by employing the warm-up dataset to conduct supervised fine-tuning (SFT) with an LLM across multiple downstream tasks. The objective of SFT is to enhance the LLM's capacity to generate responses that align with its preferences, utilizing annotated data. Prior research conducted by Ramamurthy et al. (2022) supports the notion that employing SFT prior to reinforcement learning (RL) leads to improved outcomes. Furthermore, to differentiate between specific tasks during training, we incorporate a brief instruction preceding the input, such as "This is a... (generative/question-answering/classification) task.".

### 3.2.2 Building Reward Model

Next, we construct a reward model to learn the effectiveness of prompts based on the preferences of different LLMs. This approach is motivated by the fact that discriminative annotation through sorting incurs significantly lower costs compared to generating annotations for answers. Initially, we obtain a ranking sequence for an LLM in a specific downstream task. We sort the outputs generated by candidate prompts $\{P_1, P_2, \ldots, P_{k-1}, P_k\}$ alongside the original prompt $P$, using the same evaluation metric as described in Sec. 3.1. This sorting process yields a ranking sequence $\{P_1, P_2, \ldots, P, P_{k-1}, P_k\}$. Prompts to the left of $P$ exhibit poorer inference results, while prompts to the right demonstrate better results. Next, we em-

ploy the ranking sequence to train a reward model. We utilize the same LLM utilized in the SFT process (referred to as LL̂M) and replace the softmax layer with a linear layer to construct the reward model. The reward model takes a prompt as input and produces a scalar score indicating the quality of the prompt. We form pairwise ranking pairs by combining prompts from the ranking sequence and employ Pairwise Ranking Loss for training, as illustrated below:

$$L_\theta = -\frac{1}{\binom{k}{2}} E_{(x,y_w,y_l)\sim D} \quad (1)$$
$$[log(\sigma(r_\theta(x,y_w) - r_\theta(x,y_l)))],$$

where $x$ represents the original prompt, $y_w$ and $y_l$ denote the higher-scoring and lower-scoring prompts, respectively, in the corresponding ranking pair. $r_\theta$ represents the scalar output of the reward model, $D$ is the set of ranking pairs, and $K$ denotes the number of candidate prompts. Through this process, based on the outputs generated by LL̂M with the given prompt, the reward model learns to assign higher scores (rewards) to better prompts and lower scores (rewards) to inferior prompts, thus imitating an LLM's preferences.

### 3.2.3 Reinforcement Learning

Subsequently, we employ Reinforcement Learning (RL) to further fine-tune LLMs. RL is used to adjust the bias in the reward model's scoring since the distribution of generated prompts might change during the SFT process. The primary objective of optimization is to maximize the scores of prompts generated by LL̂M after SFT (referred to as the SFT model), as evaluated by the reward model. To achieve this, we utilize a combination of Proximal Policy Optimization (PPO) (Schulman et al., 2017) and RRMF algorithms (note that RRMF is inspred by RRHF (Yuan et al., 2023)) for joint learning.

**Policy Optimization.** This step aims to optimize the RL policy to improve the performance of the RL model. We first adopt the datasets shown in Table 3, which to construct environment-action pairs. The environment refers to the original prompt, and the action represents the prompt generated by LL̂M without instruct-tuning. We pass the environment-action pairs to the reward model to obtain rewards. In this process, we introduce an actor model, which is LL̂M, and a frozen model, which is SFT model with its parameters frozen during the RL training process. The frozen model serves as a benchmark to evaluate whether the updated actor

model has advantages over it. We then calculate the policy gradient loss (i.e., actor's loss) based on the importance ratio and reward $(r + \gamma V_{\text{next}} - V_{\text{cur}})$, and calculate the value loss (i.e., critic's loss) by comparing the predicted value $V_{\text{pred}}$ with the true value $(r + V_{\text{next}})$ as follows:

$$L_{\text{pg}} = \frac{P_{\pi_a}(t)}{P_{\pi_f}(t)}(r + \gamma V_{\text{next}} - V_{\text{cur}}), \quad (2)$$
$$L_{\text{v}} = \|V_{\text{pred}} - (r + V_{\text{next}})\|. \quad (3)$$

Here, $\frac{P_{\pi_a}(t)}{P_{\pi_f}(t)}$ represents the ratio of probabilities (i.e., importance ratio) of generating the same token under the actor model and the frozen model. $(r + \gamma V_{\text{next}} - V_{\text{cur}})$ represents the reward of the current step. $V_{\text{pred}}$ denotes the predicted value, and $(r + V_{\text{next}})$ denotes the true value.

Next, we maximize the mathematical expectation of the reward model, aiming to consistently generate prompts that LL̂M perceives as the best in the RL-trained SFT model (referred to as RL model). We feed prompts $x$ generated by the SFT model based on the datasets shown in Table 3 (i.e., $D$) into the RL model $\pi_\phi^{RL}$ to obtain an optimized prompt $y$. $y$ changes every time the RL model is updated. We then input $(x, y)$ into the reward model $r_\theta$ and calculate a score (i.e., reward), which represents the real-time feedback from the reward model. The loss function is defined as follows:

$$L_\phi^{r_\theta} = E(x,y) \sim D_{\pi_\phi^{RL}}[r_\theta(x,y)]. \quad (4)$$

Finally, we combine the above loss functions to optimize the RL policy from multiple perspectives. The final loss function is defined as:

$$L_\rho = \alpha_1 L_{\text{pg}} + \alpha_2 L_{\text{v}} + \alpha_3 L_\phi^{r_\theta}, \quad (5)$$

where $\alpha_1$, $\alpha_2$, and $\alpha_3$ represent the optimal weights of each function, which are determined through experiments (the same applies below).

**SFT Approximating.** This step aims to maintain similarity between the RL model and the SFT model. When the RL model undergoes parameter updates, it leads to variations in the generated prompt $y$ based on the given prompt $x$. If there is a significant discrepancy between the RL model and the SFT model, it can result in inaccurate estimation of scores by the reward model. To address this issue, we measure the distance between the prompts generated by the RL model and the SFT model using Kullback-Leibler (KL) divergence.

The objective is to minimize the KL divergence and the loss function is defined as follows:

$$L_\phi^{SFT} = -\beta log(\pi_\phi^{RL}(y|x)/\pi^{SFT}(y|x)). \quad (6)$$

where $\pi_\phi^{RL}(y|x)$ and $\pi^{SFT}(y|x)$ represent prompts generated by RL model and the SFT model, respectively.

Next, we have borrowed the idea from RRHF (Yuan et al., 2023) but adapt it to focus on "model feedback" instead of "human feedback". We name it Ranking Responses from Model Feedback (RRMF). Specifically, we calculate the likelihood probability of LL̂M during SFT and align this probability with the score of the reward model. To optimize this objective, we employ supervised learning with a rank loss, defined as follows:

$$p_i = \frac{\sum \log P_\pi(y_i|x, y_i)}{\|y_i\|}, \quad (7)$$

$$L_r = \sum_{r_i < r_j} max(0, p_i - p_j), \quad (8)$$

where $p_i$ is the conditional log probability which represents the reward of each optimized prompt $y_i$. $r_i$ represents the reward model $r_\theta(x, y_i)$. We also incorporate the cross-entropy loss introduced by RRHF (Yuan et al., 2023) to learn the generated prompts $y_i'$ with the highest reward $r_i'$ as follows:

$$L_{ft} = -\Sigma log P_\pi(y_i'|x, y_i'). \quad (9)$$

Finally, we combine the above loss functions for SFT approximating as follows:

$$L_{SFT} = \beta_1 L_\phi^{SFT} + \beta_2 L_{ft} + \beta_3 L_r. \quad (10)$$

**Generalization Maintaining.** This step addresses the issue of catastrophic forgetting by ensuring that an LLM performs well not only on specific tasks but also on general NLP tasks. To achieve this, we follow a similar approach as outlined in InstructGPT (Ouyang et al., 2022). We sample 10% data from general NLP tasks in GLUE (Wang et al., 2018) and the SuperGLUE benchmark (Wang et al., 2019), which are considered representative, as indicated in Table 3, during the pre-training phase. The objective of pre-training is to generate outputs that are as good as or better than the original one based on the original prompts. The original prompts are taken from Natural Instructions (Wang et al., 2022b). The loss function is as follows:

$$L_{Pre} = \gamma E_x \sim D_{pretrain}[log(\pi_\phi^{RL}(x))], \quad (11)$$

where $D_{pretrain}$ represents the selected datasets for pre-training.

**Joint learning.** Finally, we make joint learning with the above-mentioned loss functions as follows:

$$L_\phi = \gamma_1 L_\rho + \gamma_2 L_{SFT} + \gamma_3 L_{Pre}. \quad (12)$$

# 4 Experiments

In this section, We conduct experiments with three popular LLMs as LL̂M, respectively, including BLOOM (7B), GPT-J (6B), and LLaMA (7B), on different downstream tasks to validate the effectiveness of MAPO.

## 4.1 Experimental Setups

The experiments are executed on 4 Nvidia A100 GPUs with 80GB each, using PyTorch in Python. DeepSpeed [2] is utilized in the training process. The maximum sequence length for original prompts and optimized prompts are both set to 512 tokens. The number of epochs is set to 20 in the entire training process. We provide the detailed configuration of the hyperparameters in Appendix B. The dataset and metrics we utilize are the same as those described in Sec. 2. All results are reported on the corresponding test sets or 10% dev sets if a dataset does not have a test set. Details of all used baselines and datasets are in Appendix D and E.

## 4.2 Main Results

The main results are shown in Table 2. We observe that the performance increase evidently among all LLMs during SFT. We then utilize MAPO to make further optimization. We find the optimized prompts generated by MAPO are more adaptive in QA and generation task for BLOOM (increase by 20.5% for CloseQA and by 30.9% for Explan compared with SFT (p<0.01)) and GPT-J (increase by 21.4% for CloseQA and by 20.6% for Explan compared with SFT (p<0.01)). And the prompts are more adaptive in classification task (increase by 22.8% for News (p<0.01)) for LLaMA. These results indicate that MAPO effectively enhances the performance of various LLMs and exhibits preferences in different downstream tasks.

To validate the superiority of MAPO, we compare it with several SOTA prompt optimization baselines in various popular tasks based on the same setting Roberta-Large, as shown in Table 3. The reported results represent the best-performing LLM among the three LLMs, which indicates that

---

[2]https://github.com/microsoft/DeepSpeed

| Task | Dataset | Original | | | SFT-optimized | | | $\overline{\uparrow}$(%) | MAPO-optimized | | | $\overline{\uparrow}$(%) |
|------|---------|--------|--------|-------|----------------|----------------|----------------|------|----------------|----------------|----------------|------|
| | | BLOOM | GPT-J | LLaMA | BLOOM(↑(%)) | GPT-J(↑(%)) | LLaMA(↑(%)) | | BLOOM (↑(%)) | GPT-J(↑(%)) | LLaMA(↑(%)) | |
| **QA** | AdverQA | 13.5 | 3.0 | 3.2 | 18.3 (35.6) | 9.4 (213.3) | 23.2 (625.0) | 291.3 | 19.5 (6.6) | 11.0 (17.0) | 25.1 (8.2) | 10.6 |
| | OpenQA | 25.9 | 17.0 | 13.3 | 26.7 (3.1) | 20.3 (19.4) | 15.4 (15.8) | 12.8 | 27.2 (1.9) | 21.0 (3.4) | 16.1 (4.5) | 3.3 |
| | CloseQA | 6.4 | 6.9 | 10.8 | 7.8 (21.9) | 8.4 (21.7) | 13.9 (24.7) | 24.1 | 9.4 (20.5) | 10.2 (21.4) | 14.8 (6.5) | 16.1 |
| **CLS** | News | 92.8 | 0.0 | 1.1 | 95.5 (2.9) | 5.5 (-) | 10.1 (818.2) | - | 98.7 (3.4) | 6.3 (14.5) | 12.4 (22.8) | 13.6 |
| | Movie | 90.9 | 51.1 | 78.7 | 92.6 (1.9) | 52.7 (3.1) | 81.3 (3.3) | 2.8 | 93.3 (0.8) | 53.9 (2.3) | 82.5 (1.5) | 1.5 |
| | QASC | 99.4 | 54.0 | 61.6 | 99.9 (0.5) | 56.3 (4.3) | 70.2 (14.0) | 6.2 | 99.9 (0.0) | 56.8 (0.9) | 72.8 (3.7) | 1.5 |
| **GEN** | Topics | 29.5 | 17.5 | 14.3 | 34.8 (18.0) | 21.6 (23.4) | 18.6 (30.1) | 23.8 | 36.2 (4.0) | 23.4 (8.3) | 19.5 (4.8) | 5.7 |
| | Summary | 46.1 | 13.1 | 6.6 | 48.8 (5.9) | 16.7 (27.5) | 10.7 (62.1) | 31.8 | 50.2 (2.9) | 17.8 (6.6) | 12.2 (14.0) | 7.8 |
| | Explan | 5.7 | 8.5 | 6.9 | 6.8 (19.3) | 10.7 (25.9) | 8.2 (18.8) | 21.3 | 8.9 (30.9) | 12.9 (20.6) | 9.1 (11.0) | 20.8 |

Table 2: Performance is evaluated for BLOOM, GPT-J, and LLaMA using original, SFT-optimized, and MAPO-optimized prompts with a frozen LLM during inference. The symbols ↑(%) and $\overline{\uparrow}$(%) under SFT-optimized denote the relative increase from original prompts, while those under MAPO-optimized indicate improvement over SFT-optimized results. It's emphasized that the term "frozen LLM for inference" means the model hasn't been trained directly on downstream tasks but only makes inferences. Thus, there's no training data with prompts as inputs and expected responses as outputs. "CLS" represents classification, and "GEN" stands for generation tasks.

| | | SST-2 | YelpP. | MR | CR | RTE | QNLI | SNLI | MNLI | MRPC |
|---|---|-------|--------|-----|-----|------|------|------|------|------|
| F | Finetuning | 80.6 | 88.7 | 67.4 | 73.3 | 58.6 | 60.2 | 54.6 | 47.8 | 77.4 |
| C | Softprompt | 73.8 | 88.6 | 74.1 | 75.9 | 54.7 | 49.7 | 36.1 | 33.2 | 51.6 |
| | Black-Box | 89.1 | 93.2 | 86.6 | 87.4 | 52.6 | 48.8 | 46.6 | 42.9 | 61.6 |
| | Autoprompt | 75 | 79.8 | 62 | 57.5 | - | - | - | - | - |
| D | Manual | 82.8 | 83 | 80.9 | 79.6 | 51.6 | 50.8 | 31.1 | 51.7 | 67.4 |
| | In-Context | 85.9 | 89.6 | 80.6 | 85.5 | 60.4 | 53.8 | 47.1 | 53.4 | 45.8 |
| | Instructions | 89 | 84.4 | 85.2 | 80.8 | - | - | - | - | - |
| | GrIPS | 87.1 | 88.2 | 86.1 | 80 | - | - | - | - | - |
| | RLprompt | 92.5 | 95.1 | 87.1 | 89.5 | - | - | - | - | - |
| | TEMPERA | 91.9 | 92.6 | 88 | **91.1** | 60.3 | 57.4 | 56.4 | 45.2 | 74 |
| | AMA | 95.7 | - | - | - | 75.1 | - | - | - | - |
| | SFT | 94.9 | 92 | 88.5 | 87.6 | 74.3 | 62.5 | 58.8 | 54.6 | 78.5 |
| | MAPO-w/o g | 96.0 | 93.3 | 90.1 | 88.7 | 75.2 | 63.0 | 59.8 | 55.7 | 79.0 |
| D | MAPO | **96.1** | **93.5** | **90.2** | 88.9 | **75.3** | **63.1** | **60.0** | **55.7** | **79.3** |

Table 3: The few-shot performance of SFT, MAPO with SOTA prompt optimizing baselines in downstream tasks. F: Finetuning, C: Continous prompt, D: Discrete prompt.

our method applies not only to LLMs but also to smaller LMs. We analyze the possible reasons as follows: MAPO employs both SFT and RL to optimize LLMs. In fact, the SFT process is not specific to LM. Fine-tuning smaller models is feasible and common, requiring fewer computational resources. RL is a widely-used algorithm across applications and model scales, and small models require less computational and storage resources, making RL more feasible on them.

We also test the performance of MAPO with the above-mentioned SOTA prompt optimization baselines. We use three LLMs, including BLOOM, GPT-J, and LLaMA, to replace the BART model used in Table 3 for verifying the nine datasets in Table 2, as shown in Table 4. Due to SFT in LLMs equals fine-tuning pretrained language models, we directly list the SFT results in the Fine-tuning row. Apart from Fine-tuning, we also freeze the LLMs and only modify the prompts for inference on downstream tasks. According to the experimental results, the performance of almost all baselines, except RLprompt, does not exceed that of Fine-tuning /SFT, and some even do not outperform the original LLMs. This highlights the importance of SFT

in LLMs. When we add RL, as in the case of RLprompt, the performance on downstream tasks surpasses that of SFT, indicating the significance of RL for prompt optimization. Moreover, using our proposed MAPO method to optimize the prompt further improves performance over RLprompt, except in a very few cases, such as using BLOOM for movie classification tasks. These experimental results demonstrate that the MAPO method proposed in this study makes a substantial contribution to improving the performance and accuracy in downstream tasks.

Moreover, we conduct experiments to evaluate the domain transfer performance of MAPO. The results are presented in Table 5 and Table 6, while the results of LLMs with original prompts are reported by Arora et al. (2022). Remarkably, we observe that each LLM, when using prompts optimized by MAPO, displays improved performance across various downstream tasks. Specifically, BLOOM exhibits the highest increase in performance compared with GPT-J and LLaMA. This experiment clearly demonstrates the significant domain transfer capability of MAPO.

### 4.3 Ablation Study

**The effect of RL compared with SFT.** From the experiments (Table 3, Table 5 and Table 6), we can observe that the performance improvements gained solely from using SFT are less than half of those achieved by our proposed MAPO method, both on similar tasks and general NLP tasks. This clearly indicates the effectiveness of MAPO in optimizing model-adaptive prompts.

In order to further demonstrate RL is necessary and how it compares to simply extending SFT with a larger warm-up dataset, we use various proportions of the warm-up dataset to progressively increase the SFT training data and then introduce RL

| | Task | QA | | | | CLS | | | GEN | |
|---|---|---|---|---|---|---|---|---|---|---|
| | | Ad | Op | Cl | Ne | Mo | QA | To | Su | Ex |
| (BLOOM) | Original | 13.5 | 25.9 | 6.4 | 92.8 | 90.9 | 99.4 | 29.5 | 46.1 | 5.7 |
| F | Finetuning/SFT | 18.3 | 26.7 | 7.8 | 95.5 | 92.6 | 99.9 | 34.8 | 48.8 | 6.8 |
| C | Soft prompt | 14.5 | 24.5 | 6.5 | 92.1 | 90.5 | 99.1 | 30.1 | 44.6 | 5.5 |
| | Black-Box | 15.2 | 24.7 | 6.9 | 93.3 | 91.6 | 99.3 | 31.2 | 45.7 | 5.8 |
| | Autoprompt | 15.7 | 25.0 | 7.1 | 93.6 | 91.9 | 99.4 | 31.6 | 46.0 | 6 |
| D | Manual | 13.7 | 24.6 | 6.8 | 91.8 | 90.9 | 99.0 | 31.5 | 45.1 | 5.7 |
| | In-Context | 13.9 | 24.7 | 6.7 | 91.6 | 90.9 | 99.3 | 31.8 | 45.6 | 5.9 |
| | Instructions | 15.0 | 24.9 | 6.7 | 92.7 | 91.0 | 99.2 | 30.8 | 45.5 | 5.8 |
| | GrIPS | 16.6 | 25.3 | 6.8 | 93.1 | 91.2 | 99.4 | 31.8 | 46.7 | 6.2 |
| | RLprompt | 19.2 | 26.9 | 8.9 | 97.5 | 94.1 | 99.9 | 35.9 | 49.1 | 8.2 |
| | TEMPERA | 17.4 | 25.5 | 7.2 | 94.6 | 91.7 | 99.5 | 33.1 | 46.7 | 6.2 |
| | AMA | 19.1 | 26.4 | 7.6 | 95.1 | 92.4 | 99.4 | 33.5 | 47.9 | 6.1 |
| **D** | **MAPO** | **19.5** | **27.2** | **9.4** | **98.7** | **93.3** | **99.9** | **36.2** | **50.2** | **8.9** |
| (GPT-J) | Original | 3.0 | 17.0 | 6.9 | 0.0 | 51.1 | 54 | 17.5 | 13.1 | 8.5 |
| F | Finetuning/SFT | 9.4 | 20.3 | 8.4 | 5.5 | 52.7 | 56.3 | 21.6 | 16.7 | 10.7 |
| C | Soft prompt | 5.4 | 16.6 | 6.8 | 2.1 | 51.0 | 54.5 | 17.8 | 13.7 | 8.9 |
| | Black-Box | 7.3 | 17.5 | 7.2 | 2.5 | 51.4 | 54.7 | 18.2 | 14.3 | 9.1 |
| | Autoprompt | 7.8 | 17.9 | 7.3 | 1.9 | 51.5 | 54.6 | 19.1 | 14.6 | 9.3 |
| D | Manual | 5.7 | 15.9 | 6.5 | 1.7 | 50.9 | 54.9 | 18.7 | 13.9 | 9.5 |
| | In-Context | 5.6 | 16.5 | 6.7 | 1.6 | 50.7 | 54.6 | 19.2 | 14.0 | 9.3 |
| | Instructions | 7.1 | 17.1 | 7.2 | 1.7 | 51.6 | 54.0 | 19.1 | 14.3 | 9.2 |
| | GrIPS | 7.7 | 17.9 | 7.7 | 4.3 | 52.1 | 55.6 | 19.7 | 16.4 | 10.3 |
| | RLprompt | 10.1 | 20.0 | 9.5 | 5.7 | 53.7 | 56.4 | 22.1 | 17.2 | 11.8 |
| | TEMPERA | 9.5 | 19.3 | 9.1 | 4.8 | 53.2 | 56.1 | 22.3 | 16.8 | 11.4 |
| | AMA | 9.6 | 19.1 | 8.8 | 5.0 | 52.9 | 56.3 | 22.4 | 16.5 | 11.6 |
| **D** | **MAPO** | **11.0** | **21.0** | **10.2** | **6.3** | **53.9** | **56.8** | **23.4** | **17.8** | **12.9** |
| (LLaMA) | Original | 3.2 | 13.3 | 10.8 | 1.1 | 78.7 | 61.6 | 14.3 | 6.6 | 6.9 |
| F | Finetuning/SFT | 23.2 | 15.4 | 13.9 | 10.1 | 81.3 | 70.2 | 18.6 | 10.7 | 8.2 |
| C | Soft prompt | 7.7 | 12.6 | 10.2 | 4.4 | 77.4 | 62.7 | 15.6 | 8.2 | 7.3 |
| | Black-Box | 9.2 | 13.3 | 10.7 | 5.6 | 78.1 | 63.1 | 16.2 | 8.4 | 7.5 |
| | Autoprompt | 10.3 | 13.5 | 11.0 | 7.4 | 78.4 | 65.2 | 16.7 | 9.0 | 7.6 |
| D | Manual | 12.3 | 12.7 | 11.1 | 7.5 | 77.5 | 64.8 | 16.2 | 8.3 | 6.7 |
| | In-Context | 13.7 | 13.0 | 10.8 | 8.0 | 77.7 | 65.3 | 16.5 | 8.5 | 7.0 |
| | Instructions | 16.2 | 13.2 | 11.3 | 7.5 | 78.0 | 65.7 | 17.1 | 9.1 | 7.5 |
| | GrIPS | 19.3 | 14.7 | 13.4 | 9.3 | 80.6 | 68.6 | 18.9 | 10.7 | 8.3 |
| | RLprompt | 24.7 | 15.8 | 14.3 | 11.6 | 81.9 | 71.4 | 19.2 | 11.7 | 8.8 |
| | TEMPERA | 22.6 | 15.4 | 13.8 | 8.9 | 81.5 | 69.6 | 18.7 | 9.6 | 8.7 |
| | AMA | 23.5 | 15.5 | 13.6 | 8.6 | 81.7 | 70.5 | 18.5 | 9.8 | 8.9 |
| **D** | **MAPO** | **25.1** | **16.1** | **14.8** | **12.4** | **82.5** | **72.8** | **19.5** | **12.2** | **9.1** |

Table 4: The performance with a frozen LLM for inference of MAPO with SOTA prompt optimizing baselines in nine tasks from P3 benchmark using LLaMA. F: Finetuning/SFT, C: Continous prompt, D: Discrete prompt.

to it as shown in Table 17. Our findings consistently show that RL adds value to the performance beyond what is achieved by SFT alone across all proportions of the dataset. This affirms the effectiveness of RL irrespective of the SFT dataset size. However, as the proportion of the warm-up dataset increases, the margin of improvement from adding RL begins to decline. While one could hypothesize that adding RL to a very large SFT dataset might not result in as significant an improvement as it would for a smaller dataset, this observation actually underscores our method's suitability for low-resource scenarios.

Moreover, we have tried different number of epochs to see if extended training time consistently improves SFT performance as shown in Table 18. Extending the training time does not consistently lead to performance improvements for SFT. In some instances, the performance even declines. It is important to note that we save the best-performing models in real-time during training, as the peak performance does not necessarily occur at the final epoch.

**The effect of warm-up dataset.** As shown in Fig. 5 and Table 11, our study examines the effects of different proportions of the warm-up dataset on MAPO's performance. The results indicate that as the size of the warm-up dataset increases, performance typically improves. BLOOM is particularly sensitive, showing a pronounced growth trend. Conversely, GPT-J shows a more gradual growth. LLaMA's performance reveals an inflection around 60%, suggesting other factors also influence its performance. Even with reduced dataset sizes, the decrement in performance remains minimal, highlighting the method's suitability for low-resource tasks. We also conduct few-shot experiments on general NLP tasks with just 10% data and observe promising improvements. This underlines our method's adaptability and effectiveness in scenarios of data scarcity.

**The effect of PPO and RRMF.** To investigate the specific effects of PPO and RRMF during the RL process, we conduct separate experiments to evaluate the contributions of each component. The experimental results, depicted in Fig.6 (with details provided in Table 10 in Appendix F), clearly demonstrate the important roles played by PPO and RRMF in enhancing the performance of MAPO. We propose the following explanations for these results: PPO focuses on reducing the dissimilarity between the RL model and the SFT model. RRMF aligns the scores from the reward model with the likelihood probabilities of an LLM. Both PPO and RRMF aim to assign higher probabilities to prompts that are more adaptable to the model.

**The effect of the Randomness.** We also incorporate randomness (e.g., temperature) during the generation process of LLM. Given that our prompts do not require high creativity, we have set a lower temperature range [0-0.5] for generation, within which we aim to generate optimal prompts. To further investigate the impact of varying temperatures on the generated output, we conduct an additional set of experiments to assess the performance of the MAPO method under different randomness settings (temperature=0,0.2,0.5,0.8) as shown in Table 14, Table 12, Table 15 and Table 16. Each experiment group runs 5 times. Our findings reveal that a high-temperature setting (t=0.8) tends to produce inferior prompts that lead to less accurate outputs for a specific task. Lower temperature (t=0.2) or greedy settings (t=0) are likely to produce more accurate outputs that are closer to our optimal results. This suggests that in a task like prompt optimization,

| Task | Dataset | BLOOM | +SFT | (↑(%)) | ↑(%) | +MAPO | (↑(%)) | ↑(%) | GPT-J | +SFT | (↑(%)) | ↑(%) | +MAPO | (↑(%)) | ↑(%) |
|---|---|---|---|---|---|---|---|---|---|---|---|---|---|---|---|
| Coref. | xwinograd | 60.1 | 60.2 | 0.2 | 0.2 | 60.6 | 0.9 | 0.9 | - | - | - | - | - | - | - |
| NLU | BoolQ | 67.9 | 68.0 | 0.1 | 0.4 | 68.2 | 0.4 | 0.9 | 67.2 | 67.4 | 0.3 | 0.2 | 67.9 | 1.0 | 0.5 |
| | CB | 77.6 | 77.8 | 0.3 | | 78.1 | 0.6 | | 83.9 | 84.1 | 0.2 | | 84.2 | 0.4 | |
| | COPA | 74.0 | 74.3 | 0.4 | | 75.0 | 1.4 | | 84.0 | 84.2 | 0.2 | | 84.2 | 0.2 | |
| | MultiRC | 59.7 | 60.3 | 1.0 | | 60.4 | 1.2 | | 63.8 | 63.9 | 0.2 | | 64.1 | 0.5 | |
| | ReCoRD | 69.8 | 70.1 | 0.4 | | 70.2 | 0.6 | | 74.4 | 74.5 | 0.1 | | 74.7 | 0.4 | |
| | WiC | 61.4 | 61.6 | 0.3 | | 62.0 | 1.0 | | 61.0 | 61.1 | 0.2 | | 61.3 | 0.5 | |
| | WSC | 64.4 | 64.7 | 0.5 | | 65.1 | 1.1 | | 77.9 | 78.0 | 0.1 | | 78.1 | 0.3 | |
| NLI | ANLIR1 | 31.5 | 31.7 | 0.6 | 0.6 | 32.1 | 1.9 | 1.3 | 37.8 | 38.0 | 0.5 | 0.3 | 38.2 | 1.1 | 0.7 |
| | ANLIR2 | 35.1 | 35.2 | 0.3 | | 35.4 | 0.9 | | 37.9 | 38.0 | 0.3 | | 38.3 | 1.1 | |
| | ANLIR3 | 37.1 | 37.5 | 1.1 | | 37.8 | 1.9 | | 40.9 | 41.0 | 0.2 | | 41.1 | 0.5 | |
| | StoryCloze | 79.0 | 79.2 | 0.3 | | 79.5 | 0.6 | | 87.8 | 87.9 | 0.1 | | 87.9 | 0.1 | |
| CLS | Amazon | 65.2 | 66.4 | 1.8 | 1.4 | 67.7 | 3.8 | 3.3 | 68.2 | 68.7 | 0.7 | 0.6 | 69.4 | 1.8 | 1.6 |
| | DBPedia | 70.5 | 71.2 | 1.0 | | 72.5 | 2.8 | | 83.9 | 84.2 | 0.4 | | 85.1 | 1.4 | |
| QA | DROP | 67.9 | 68.2 | 0.4 | 1.9 | 69.9 | 2.9 | 5.6 | 51.6 | 51.9 | 0.6 | 1.3 | 52.8 | 2.3 | 3.2 |
| | NQ | 15.1 | 15.4 | 2.0 | | 16.1 | 6.6 | | 19.6 | 20.1 | 2.6 | | 20.8 | 6.1 | |
| | RealTimeQA | 29.0 | 30.2 | 4.1 | | 31.5 | 8.6 | | 36.0 | 36.5 | 1.4 | | 37.2 | 3.3 | |
| | WebQs | 34.8 | 35.1 | 0.9 | | 36.3 | 4.3 | | 44.1 | 44.3 | 0.5 | | 44.6 | 1.1 | |

Table 5: Zero-shot domain transfer performance based on BLOOM and GPT-J with original, SFT-optimized and MAPO-Optimized prompts. CLS: Classification, M: MAPO. The (↑(%) ) and ↑(%) represent the increase degree of MAPO-optimized prompts compared with original prompts in each dataset and task, respectively (The same below).

| Task | Dataset | LLaMA | +SFT | (↑(%)) | ↑(%) | +MAPO | (↑(%)) | ↑(%) |
|---|---|---|---|---|---|---|---|---|
| RS | BoolQ | 76.5 | 76.6 | 0.1 | 0.1 | 76.7 | 0.3 | 0.5 |
| | PIQA | 79.8 | 79.9 | 0.1 | | 80.0 | 0.3 | |
| | SIQA | 48.9 | 48.9 | 0.0 | | 49.0 | 0.2 | |
| | HellaSwag | 76.1 | 76.2 | 0.1 | | 76.5 | 0.5 | |
| | WinoGrande | 70.1 | 70.2 | 0.1 | | 70.5 | 0.6 | |
| | ARC-e | 72.8 | 72.9 | 0.1 | | 73.2 | 0.6 | |
| | ARC-c | 47.6 | 47.6 | 0.0 | | 47.8 | 0.4 | |
| | OBQA | 57.2 | 57.4 | 0.3 | | 57.9 | 1.2 | |
| QA | NQ | 16.8 | 17.2 | 2.4 | 1.4 | 18.1 | 7.7 | 4.7 |
| | RACE | 50.0 | 50.2 | 0.4 | | 50.8 | 1.6 | |

Table 6: Zero-shot domain transfer performance based on LLaMA with original, SFT-optimized and MAPO-Optimized prompts. RS: Commonsense Reasoning.

introducing a stable (low temperature) but slight degree of variability (non-zero temperature) yields the best results.

## 4.4 Case Study and Error Analysis

We conduct a case study to visualize the prompts optimized by MAPO, as shown in Table 7. Additional cases are included in Appendix G. We first observe that the majority of the original prompts have significant modifications after optimization through our MAPO method. Only about 10% of the generated prompt pairs remain completely unchanged. To further quantify these changes, we calculate a normalized edit distance. Given the varying lengths of different prompt pairs, we divide the edit distance by the average length of the two strings. This yields a value between 0 and 1, where 0 indicates identical strings and 1 indicates completely different strings. The average normalized edit distance for all prompt pairs stands at 0.67, demonstrating that most prompts do experience substantial modifications.

Next, we provide a detailed examination of these modifications. In the QA task, BLOOM transforms active voice into passive voice, GPT-J utilizes the phrase "the term used" and substitutes "refer to" with "denote", while LLaMA adopts a more informal style by mentioning the "commonly used term". In the generation task, both BLOOM and GPT-J present similar prompts that emphasize topic coverage. LLaMA maintains the original sentence structure but modifies the subjects and replaces "decorate" with "adorn". In the classification task, all three LLMs rearrange the word order and offer additional details about the topic. Therefore, MAPO demonstrates its prompt optimization capabilities by adapting better prompts to specific tasks for different LLMs while preserving core information and adjusting tone or structure as necessary.

However, there are also some errors during prompt optimization, including prompts with incomplete sentences, prompts with improper prepositions or missing necessary parts, and prompts with ambiguous meanings, etc. Therefore, there is ample room for improvement in MAPO to better adapt to different LLMs in downstream tasks.

## 4.5 Exploratory Analysis

We conduct an exploratory analysis to further investigate the patterns in optimized prompt as shown in Fig. 7, Fig. 8 and Fig. 9. We extract the three most frequent words from the original prompt and investigate their distribution in the optimized prompt for each LLM, while either retaining high-frequency words in instructions (including sentence, topics, subjects, present, statement, discussed, mentioned, included, following) or removing them.

Taking the generation task (Fig. 7) as an example, when high-frequency words in instructions are retained, we observe that BLOOM retains a relatively higher proportion of the original prompts compared to GPT-J and LLaMA, while LLaMA retains the fewest. When these words are removed,

| Task | Prompts |
|---|---|
| QA | **Original**: How do people in Mexico refer to the largest Presbyterian church?
**BLOOM**: What is the way in which the biggest Presbyterian church is referred to by individuals in Mexico?
**GPT-J**: What is the term used by Mexicans to denote the biggest Presbyterian church?
**LLaMA**: What is the commonly used term for the biggest Presbyterian church in Mexico? |
| Gen | **Original**: What are the topics in the sentence: The furniture in the living room is decorated with flowers.
**BLOOM**: What topics does the sentence "The living room furniture is adorned with flowers" cover?
**GPT-J**: What topics are covered in the sentence "The living room furniture is adorned with flowers"?
**LLaMA**: Which subjects are present in the statement: Flowers adorn the furniture in the living room. |
| Class | **Original**: Reincarnated Napster Goes Public Roxio bought the Napster name in a bankruptcy auction two years ago. Now, the company dumps its CD-burning software business to concentrate on selling and delivering music over the web under the Napster name.What label best describes this news article?
**BLOOM**: Roxio, the buyer of Napster's name in a bankrupt auction two years ago, has taken the reincarnated Napster public. The company has made the decision to abandon their CD-burning software business and focus solely on the distribution and sale of music online via the Napster name. What genre of news article would this story fall under?
**GPT-J**: Roxio, which acquired the Napster name through a bankruptcy auction two years ago, has taken Napster public again after rebranding it as a music delivery service. The company has divested its CD-burning software business and now focuses solely on selling music online under the Napster name. What category would this news article fall under?
**LLaMA**: The Napster name, which was purchased by Roxio in a bankruptcy auction two years ago, has now been resurrected with a public launch. Roxio has shifted its focus solely to the sale and distribution of music under the Napster name, leaving its CD-burning software business behind. What category would you assign to this news article? |

Table 7: Original prompts and MAPO-optimized prompts for three LLMs in various downstream tasks.

we notice that BLOOM has a higher proportion of words like "man", "view" in its optimized prompts, which are more relative with human. GPT-J has a higher proportion of words like "match", "grass", "bathroom", "white", which suggests it focuses on specific scenes, objects, or themes. LLaMA has a higher proportion of words like "room", "close", "playing", indicating its preferences on place and experiences. The variations observed in word distribution indicate that each LLM tends to emphasize different aspects during the optimization process. Accurate conclusions need more experiments.

## 5   Related Work

LLMs' prompt optimization process involves prompt retrieval, prompt generation from scratch and prompt editing. For prompt retrieval, for example, Ma et al. (2023) adopt greedy search to identify near-optimal prompts. Zhou et al. (2022) introduce APE for automatic instruction selection, etc. For prompt generation from scratch, Pang et al. (2023) introduce SharpT, which learns a shared latent space and generates soft prompts. White et al. (2023) describe a catalog of prompt engineering techniques. Zamfirescu-Pereira et al. (2023) investigate end-user prompt engineering using a prototype LLM-based chatbot design tool. Wang et al. (2022a) present Self-Instruct for improving instruction-following capabilities of PLMs. For prompt editing, Gao et al. (2020) automatically select label words and generate templates. Pryzant et al. (2023) introduce APO based on "gradients" to provide critical feedback on the current prompt. Deng et al. (2022) propose RLprompt based on RL. Zhang et al. (2023) propose TEMPERA, which provides interpretable prompts for different queries. Prasad et al. (2022) introduce GrIPS, a gradient-free approach for improving task instructions for LLMs. Moreover, some research focuses on incor-

porating additional knowledge to enhance prompt editing. For example, Li et al. (2023) propose DSP to generate "directional stimulus" of each input. Qin and Eisner (2021a) optimize a mixture of prompts using gradient descent to generate relational knowledge. Shin et al. (2020) develop Autoprompt, a gradient-guided approach to find the best tokens in the prompt. Jiang et al. (2020) propose mining-based and paraphrasing-based methods to automatically generate diverse prompts. Furthermore, some research focus on continuous prompt optimization instead of discrete prompt optimization mentioned before, such as research by Zheng et al. (2023), Hambardzumyan et al. (2021), Zhong et al. (2021), etc.

However, all above-mentioned prompts optimization approaches aim to obtain task-specific prompts instead of model-specific ones. Different from theirs, we dedicate at optimizing prompts for LLMs within the NLP domain and achieve impressive performance.

## 6   Conclusions

The remarkable capabilities of LLMs have revolutionized NLP in various tasks. However, their performance heavily relies on the quality of prompts. In this work, we address the prompt optimization challenge by proposing a Model-Adaptive Prompt Optimization (MAPO) approach. Through extensive experiments, we demonstrated that MAPO can adapt different LLMs with generating model-friendly prompts to enhance their capabilities across various downstream tasks. In future work, we aim to construct more fine-grained model-adaptive prompts that can adapt to the continuously evolving data encountered in real-world production environments. Additionally, we intend to enhance its applicability across a broad spectrum of linguistic contexts.

# Limitations

It is important to acknowledge certain limitations of our approach. Firstly, the effectiveness of prompt optimization heavily relies on the availability and quality of the warm-up dataset. In cases where the dataset is limited or does not sufficiently cover the specific task, the performance gains from prompt optimization may be constrained. Additionally, MAPO requires extensive SFT and RL, which can be computationally expensive and time-consuming. This could limit the scalability of MAPO, especially when dealing with large-scale tasks or datasets. Despite these limitations, our study provides valuable insights into model-adaptive prompt optimization for LLMs and contributes to the ongoing efforts in improving the performance of these LLMs in practical applications.

# Acknowledgement

This work is supported by Shanghai Municipal Science and Technology Major Project (No.2021SHZDZX0103), Science and Technology Commission of Shanghai Municipality Grant (No. 22511105902), the National Natural Science Foundation of China (No.62072323, U21A20488), Shanghai Science and Technology Innovation Action Plan (No. 22511104700), Key Projects of Industrial Foresight and Key Core Technology Research and Development in Suzhou(SYC2022009).

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

# A  Candidate Prompts

For the nine datasets selected in P3, we present one prompt and the corresponding three candidate prompts for each dataset, as shown in Table 8.

# B  Training Details

We provide the training details as shown in Table 9. Other hyper-parameters are set to default.

# C  Computational Cost

While the training phase is computationally intensive, the generation phase is relatively lightweight. Specifically, once the prompt optimizing model MAPO is trained, the prompt generation process simply involves a feed forward propagation to generate the optimal prompt instead of further optimization through SFT and RL, thus significantly reducing the computational complexity. We list the computational complexity during the training and inference phase:

**Training Phase.**  During the training phase, initially, a warm-up dataset is established. This involves generating candidate prompts using a model like GPT-3.5. For each original prompt, 1000 candidate prompts are generated. This leads to a time and space complexity of $O(N \times M)$, where $N$ is the number of original prompts, and $M$ is the number of candidates per prompt. Subsequently, an optimal prompt is searched for, which involves comparisons among candidate prompts, yielding complexities of $O(N \times M)$ for both time and space. Building the prompt optimizer is the next stage. Supervised fine-tuning (SFT) has a time complexity of $O(E \times B \times T)$, with $E$ being the number of epochs, $B$ the batch size, and $T$ the number of model parameters. Its space complexity mainly arises from model parameters and gradients, which is $O(T)$. For building the reward

| Task | Prompts |
|------|---------|
| AdverQA | **Original**: Question: "Which happened earlier, the Chinese entered the war or President Truman dispatched the United States Seventh Fleet to the Taiwan Strait?". Context: "On 27 June 1950, ...". Answer:
**Candidate 1**: Question: "Did President Truman dispatch the United States Seventh Fleet to the Taiwan Strait before or after the Chinese entered the war?" Context:"On 27 June 1950, ...". Answer:
**Candidate 2**: Question: "Did the Chinese enter the war before President Truman dispatched the United States Seventh Fleet to the Taiwan Strait, or vice versa?" Context:"On 27 June 1950, ...". Answer:
**Candidate 3**: Question: "Did the Chinese enter the war first or did President Truman send the United States Seventh Fleet to the Taiwan Strait earlier?". Context:"On 27 June 1950, ...". Answer: |
| OpenQA | **Original**: What's above the muscles and needs direct sunlight Which is the correct answer? Options: ...
**Candidate 1**: What lies beyond the muscles and requires direct exposure to sunlight? Which is the correct answer? Options: ...
**Candidate 2**:What is located above the muscles and requires direct sunlight? Which is the correct answer? -Options: ...
**Candidate 3**: Which body part requires direct sunlight and is located higher than the muscles? Which is the correct answer? Options: ... |
| CloseQA | **Original**: Q: What kind of relationship between glucagon and insulin is vital to managing fuel storage and consumption by body cells? A:
**Candidate 1**:Q: What is the essential connection between glucagon and insulin for regulating fuel storage and utilization in body cells? A:
**Candidate 2**:Q: In managing the storage and consumption of fuel by body cells, what is the crucial interrelation between insulin and glucagon? A:
**Candidate 3**: Q: What is the crucial connection between glucagon and insulin in regulating the storage and utilization of fuel by the cells in the body? A: |
| News | **Original**: Reincarnated Napster Goes Public Roxio bought the Napster name in a bankruptcy auction two years ago. Now, the company dumps its CD-burning software business to concentrate on selling and delivering music over the web under the Napster name. What label best describes this news article?
**Candidate 1**: Roxio, the buyer of Napster's name in a bankrupt auction two years ago, has taken the reincarnated Napster public. The company has made the decision to abandon their CD-burning software business and focus solely on the distribution and sale of music online via the Napster name. What genre of news article would this story fall under?
**Candidate 2**: Roxio, which acquired the Napster name through a bankruptcy auction two years ago, has taken Napster public again after rebranding it as a music delivery service. The company has divested its CD-burning software business and now focuses solely on selling music online under the Napster name. What category would this news article fall under?
**Candidate 3**:The Napster name, which was purchased by Roxio in a bankruptcy auction two years ago, has now been resurrected with a public launch. Roxio has shifted its focus solely to the sale and distribution of music under the Napster name, leaving its CD-burning software business behind. What category would you assign to this news article? |
| Movie | **Original**: writer/director joe carnahan's grimy crime drama is a manual of precinct cliches , but it moves fast enough to cover its clunky dialogue and lapses in logic . The sentiment expressed for the movie is
**Candidate 1**: The gritty crime drama by writer and director Joe Carnahan may rely heavily on familiar tropes and cliches of the genre, but its quick pace manages to distract from any awkward dialogue and illogical moments. The sentiment expressed for the movie is
**Candidate 2**: Joe Carnahan's gritty crime drama relies heavily on standard police procedures, yet its rapid pace compensates for any cumbersome dialogues and unreasonable plot holes. The sentiment expressed for the movie is
**Candidate 3**:Although writer/director Joe Carnahan's gritty crime drama contains numerous stereotypes within the precinct environment, its swift pace effectively masks its awkward dialogue and occasional lapses in logic. The sentiment expressed for the movie is |
| QASC | **Original**:If I tell you that Hydrogen bonds cause a tremendous force when a substance freezes, and ask you the question "hydrogen bonds cause a tremendous force when a substance does what", is the correct answer "strong"?
**Candidate 1**: If I were to inform you that when a substance freezes, Hydrogen bonds create a significant force and ask, "What term describes the force generated by Hydrogen bonds when a substance freezes?", would the appropriate response be "Powerful"?
**Candidate 2**: Suppose I inform you that the process of substance freezing is deeply influenced by Hydrogen bonds that generate an enormous force. Now, if I inquire, "What occurs when the substance undergoes this process?", would it be accurate to say that the force generated is "powerful"?
**Candidate 3**: Suppose I inform you that when a substance freezes, Hydrogen bonds result in a remarkable force, and inquire, "When a substance undergoes what, do hydrogen bonds cause a remarkable force?" Would it be accurate to respond with "robust"? |
| Topics | **Original**: What are the topics in the sentence: A bathroom with the toilet missing and the room fairly torn up.
**Candidate 1**: What are the subjects of the sentence: A torn-up room without a toilet.
**Candidate 2**: Which subjects are covered in the phrase "A bathroom that lacks a toilet and has a considerably damaged room"?
**Candidate 3**:What are the subjects mentioned in the statement: A torn up room that lacks a toilet in the bathroom? |
| Summary | **Original**: Sum up the following dialogue: Gordon: Did you see my car, bro? Gordon: <file_photo> Gordon: It's my first car ever! And I love it! :) Leo: Grats, bro! Leo: It looks awesome, I have to see it with my own eyes! Gordon: Are you home? Leo: Yeah Gordon: Look out of the kitchen window :) Leo: No shit :D Leo: Wait, I'm coming! Gordon: Waiting :D
**Candidate 1**: Sum up the following dialogue: Gordon asked, "Bro, have you seen my car?" and sent a file photo. He expressed his excitement saying it's his first ever car and he loves it. Leo congratulated him saying it looks awesome and expressed his wish to see it in person. Gordon asked if he was home and told him to look out of the kitchen window. Leo was surprised and replied, "No shit :D" and said he was coming. Gordon eagerly waited for him.
**Candidate 2**: Sum up the following dialogue: Gordon inquires, "Hey bro, have you laid eyes on my car?" Gordon shares a photograph of his first vehicle and expresses his adoration for it with a smiley face. Leo congratulates him and expresses interest in seeing it in person. Gordon asks if Leo is home and instructs him to look out of the kitchen window. Leo is surprised and excited, responding with laughter and promising to come see it. Gordon waits patiently.
**Candidate 3**: Sum up the following dialogue: Gordon asked his brother if he had seen his car and sent a photo of it. He expressed his love for it as it was his first car ever. Leo congratulated him and expressed his desire to see the car in person. Gordon asked if he was at home and told him to look out of the kitchen window. Leo was surprised and excited and said he would be coming soon. Gordon waited for him to arrive. |
| Explan | **Original**: Question: What does a Christian do when they get what they needed? Options:... The answer is "thank god" because
**Candidate 1**: Question: How should a Christian proceed after they have received what they required? Options: ... The answer is "thank god" because
**Candidate 2**: Question: When a Christian receives what they needed, what actions do they take? Options:... The answer is "thank god" because
**Candidate 3**: Question: What should a Christian do upon receiving what they required? Options: ... The answer is "thank god" because |

Table 8: One sample prompt and the corresponding three candidate prompts generated by GPT-3.5 for each selected dataset in P3.

model, both time and space complexities are mainly $O(N \times M \times \log M)$. The reinforcement learning (RL) part requires $O(E' \times B' \times T)$ time, where $E'$ is the number of epochs specific to RL, $B'$ is the batch size in RL, and $T$ remains the model parameters. The space complexity is $O(T)$. Summing these up, the total time complexity for the training phase becomes $O(N \times M) + O(E \times B \times T) + O(N \times M \times \log M) + O(E' \times B' \times T)$. For space, it's $O(N \times M \times \log M) + O(T)$.

**Inference Phase.** In the inference phase, an optimized prompt is generated from an original prompt using the MAPO technique. The time complexity here is dominated by a single feed-forward operation, which is $O(T)$. There is almost negligible extra space required, making the space complexity effectively $O(1)$ for this phase.

We also caclulate how long it roughly takes for a complete training run. For a LLaMA-7B model running on four A100 80GB GPUs, SFT on a high-scale task (such as the News classification task with 120,000 training data) takes about 8 hours, RL takes about 12 hours, and the complete MAPO process takes roughly 20 hours in total; For a Bloom-7B

| | Value |
|---|---|
| Gradient Accumulation Steps | 8 |
| Weight Decay | 0.1 |
| Learning Rate for Actor Model | 2e-5 |
| Learning Rate for Critic Model | 1e-5 |
| Entropy Coefficient | 0.005 |
| Value Loss Coefficient | 0.5 |
| Mini Batch Size | 32 |
| Positive Lambda Coefficient | 2.0 |
| Negative Lambda Coefficient | 1.8 |
| GAMMA | 0.99 |
| Adam Optimizer Epsilon | 1e-5 |
| GAE Lambda | 0.95 |
| Max Gradient Norm | 0.5 |
| PPO Epochs | 20 |
| Clip Parameter | 0.2 |

Table 9: Hyperparameters used for MAPO in all the tasks.

model under the same hardware conditions, SFT takes about 5 hours, RL takes about 9 hours, and the total time for MAPO takes about 14 hours; For a GPT-J-6B model, SFT takes about 10 hours, RL takes about 16 hours, and the total time for MAPO takes about 26 hours.

# D Baselines

We compared MAPO with several State-Of-The-Art (SOTA) prompt optimization baselines, including the following:

- Finetuning (Devlin et al., 2018): Finetuning (few-shot) involves finetuning the entire language model with a classification head using a few-shot dataset.

- Soft Prompt (Qin and Eisner, 2021b; Li and Liang, 2021): Soft Prompt Tuning utilizes continuous embeddings as a variant of parameter-efficient transfer learning, replacing discrete prompts.

- Black-Box (Sun et al., 2022): Black-Box Tuning combines discrete and soft prompts, with the soft part trained using gradient descent and the discrete part optimized using a gradient-free tuner.

- Autoprompt (Shin et al., 2020): Autoprompt incorporates discrete trigger tokens and updates prompts through iterative gradient search.

- Manual (Brown et al., 2020; Schick and Schütze, 2020; Sanh et al., 2021): Manual

prompt achieves strong performance on various natural language understanding and natural language generation tasks without relying on training examples.

- In-Context (Brown et al., 2020): In-Context Demonstration randomly selects a training example and concatenates it with the input query.

- Instructions: Self-Instruction manually creates prompts for each task following Natural Instructions (Wang et al., 2022b), where the prompt is concatenated with the inputs.

- GrIPS (Prasad et al., 2022): GrIPS performs phrase-level editing on the instructions and selects the best one.

- RLprompt (Deng et al., 2022): RLprompt generates discrete prompts using a reinforcement learning (RL) framework.

- TEMPERA (Zhang et al., 2023): TEMPERA is a test-time prompt editing method that uses reinforcement learning, efficiently leveraging prior knowledge and adapting to different queries, while providing an interpretable prompt for each query.

- AMA (Arora et al., 2022): AMA recursively reformats tasks and prompts using the LLM to effectively aggregate predictions across prompts using weak supervision.

For a fair assessment, we adopt the same experimental setup as in LM-BFF (Gao et al., 2020) and RLPrompt (Deng et al., 2022). We take 16 training samples from each class in our training dataset for every task, making them our few-shot dataset. So, if we consider all the classes (Y), we have a total of 16 times the number of classes as our training samples. Similarly, we pick 16 samples from each class to form our validation dataset. Besides this usual setup, we also select $n$ random examples from our training data. We call this our "in-context exemplar pool". For consistency, we repeat our experiments four times using different random seeds. Afterward, we calculate the average results and note down the usual variation we see between the results. For our language model, we've chosen to use RoBERTa large (Liu et al., 2019). We base our initial guidelines on the Natural Instructions (Mishra et al., 2021). We also ensure that the

first examples we give for context are randomly picked from a set of 16. This set is different from our few-shot dataset and is also randomly picked from our main training data. By comparing MAPO with these SOTA baselines, we gain insights into the performance and effectiveness of MAPO in various downstream tasks.

## E    Datasets

We utilized nine representative datasets from P3 (Sanh et al., 2021) to establish the warm-up dataset, covering question-answering, classification, and generation tasks. The selected datasets for each task are as follows:

- Question-Answering    Task:    AdverQA (https://huggingface.co/datasets/bigscience/ P3/tree/main/data/adversarial_qa_ dbidaf_question_context_answer), OpenQA (https://huggingface.co/ datasets/bigscience/P3/tree/main/data/ openbookqa_main_which_correct), CloseQA (https://huggingface.co/datasets/bigscience/ P3/tree/main/data/sciq_Direct_Question_ Closed_Book_).

- Classification    Task:    News    (https: //huggingface.co/datasets/bigscience/P3/ tree/main/data/ag_news_classify),    Movie (https://huggingface.co/datasets/bigscience/ P3/tree/main/data/rotten_tomatoes_ Movie_Expressed_Sentiment),    QASC (https://huggingface.co/datasets/bigscience/ P3/tree/main/data/qasc_is_correct_1)

- Generation Task: Topics (https://huggingface. co/datasets/bigscience/P3/tree/main/data/ common_gen_topics_from_the_sentence), Summary    (https://huggingface.co/datasets/ bigscience/P3/tree/main/data/samsum_ Sum_up_the_following_dialogue),    Explan (https://huggingface.co/datasets/bigscience/ P3/tree/main/data/cos_e_v1.11_generate_ explanation_given_text).

We evaluate our proposed MAPO method, along with other SOTA baselines, on the following datasets for validation: SST-2 (Socher et al., 2013), Yelp Polarity (Zhang et al., 2015), MR (Pang and Lee, 2005), CR (Hu and Liu, 2004), RTE (Dagan et al., 2005; Haim et al., 2006; Giampiccolo et al., 2007; Bentivogli et al., 2009), QNLI (Demszky et al., 2018), SNLI (Bowman et al., 2015),

MNLI (Williams et al., 2017), MRPC (Dolan and Brockett, 2005). These datasets provide a comprehensive evaluation of MAPO's performance compared to other baselines across a range of tasks, including sentiment analysis, text classification, natural language inference, and paraphrase identification, etc.

Specifically, for Table 3, the training data aligns with that used by TEMPERA (Zhang et al., 2023), that is all experiments, including our own, use Roberta-large as the backbone for validating the downstream tasks. Because the setup employs a "few-shot" methodology that has elaborated before, we name Table 3 as "few-shot". For Table 5 and 6, there is no training data involved; the LM performs zero-shot inference. That means all reported results occur without training on the datasets in Table 5 and 6. The purpose is to demonstrate the generalization (domain transfer) ability of our MAPO method. If one wishes to further enhance performance on these datasets, additional training with labeled data on Table 5 and 6 becomes necessary.

## F    Additional Experiments

**The performance of the reward model.**    We plot the performance of the reward model during the training process of MAPO as shown in Fig. 4. As the training progresses, the reward model exhibits consistent growth and improvement. The consistent increase indicates that the reward model is gradually becoming more proficient in downstream tasks. It successfully adapts to its environment, leading to improved outcomes and higher task completion rates. Therefore, it can serve as a discriminator of the goodness of an optimized prompt.

**The original capabilities maintaining ability of MAPO.** We further analyze the original capabilities maintaining ability of MAPO. We use a language model trained with MAPO, which has the ability to optimize prompts but without losing its original capabilities, to modify prompts and accomplish downstream tasks. We believe that the GLUE and SuperGLUE tasks are representative, hence we use them as pre-training tasks. However, the improvements in Table 5 and 6 are not significant, possibly due to the limited scope of our pre-training tasks. Future work can explore using a broader range of datasets for pre-training, which may lead to more significant improvements in various downstream tasks.

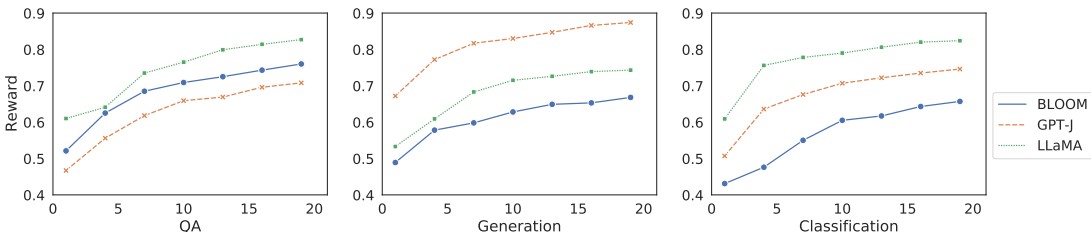

Figure 4: The performance of the reward model in three LLMs during the training process of MAPO.

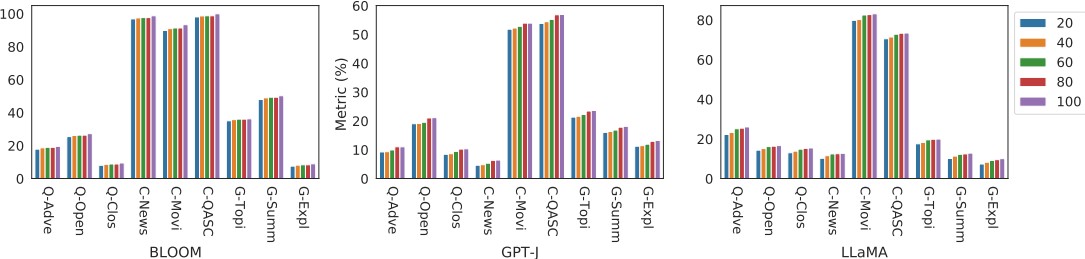

Figure 5: Performance of different proportion of warm-up dataset in various downstream tasks by three LLMs. Q: QA, C: classification, G:generation. We only keep the first four letters of each dataset's name in the figure.

Moreover, for Table 3, the training and validation data for SFT, RM, and RL are different from the data used for generalization, although they all come from Table 3. This is because we consider GLUE and SuperGLUE tasks to be representative, hence we use them as pre-training tasks. Theoretically, a more diverse NLP dataset should be selected for this part, but we happened to choose this subset. To mitigate the impact on the results, we also run another test with two steps: using the optimized prompts generated by MAPO and then using the original Roberta-Large model to make inference. As shown in Table 3 (the row "MAPO-w/o g"), the results do not show a significant decline, with a t-test greater than 0.05. The use of data from Table 3 for generalization is merely to ensure that the prompt-optimized model retains its original capabilities for downstream tasks instead of data leakage.

## G    Additional Cases

We list more cases whose prompts have been optimized by our proposed MAPO as shown in Table 19. We make detailed analysis for the difference among LLMs as follows:

- In SST-2, BLOOM and LLaMA both use phrases like "terrific flair" and "remarkable skill" to describe Khouri's ability, emphasizing positive sentiment. GPT-J uses the phrase "tremendous artistry," highlighting the artistic aspect, but does not explicitly convey the positive sentiment as strongly as BLOOM and LLaMA.

- In Yelp, BLOOM and LLaMA use phrases like "quality of the food is commendable" and "service provided is inconsistent" to provide a balanced assessment. GPT-J and the original version have the same wording, emphasizing the hit-or-miss nature of the service.

- In MR, BLOOM and LLaMA use phrases like "admirable endeavor" and "praiseworthy pursuit" to highlight the positive qualities of the venture. GPT-J and the original version use neutral language without explicitly conveying positive or negative sentiment.

- In CR, BLOOM, GPT-J, and LLaMA all express confusion or potential confusion regarding the positioning of the space key on a phone. The wording in BLOOM and LLaMA suggests that using a different key for text input is more common in phones, implying a deviation from the norm.

- In RTE, BLOOM and LLaMA emphasize the impact of the situation by using phrases like "somber site" and "distressing sight" when describing the washed-up marine animals. GPT-J and the original version provide more neutral descriptions without explicitly conveying the emotional aspect.

- In QNLI, BLOOM, GPT-J, and LLaMA all rephrase the sentence 2, maintaining the same overall meaning. The variations in wording are mainly stylistic, with BLOOM, GPT-J,

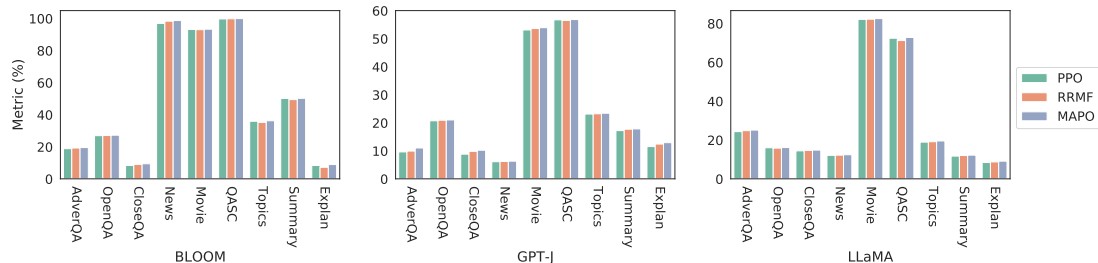

Figure 6: The separate effects of PPO and RRMF during the process of RL in constructing MAPO.

| Tasks | Dataset | PPO | | | RRMF | | | MAPO | | |
|---|---|---|---|---|---|---|---|---|---|---|
| | | BLOOM | GPT-J | LLaMA | BLOOM | GPT-J | LLaMA | BLOOM | GPT-J | LLaMA |
| **QA** | AdverQA | 18.8 | 9.6 | 24.3 | 19.2 | 9.9 | 24.8 | 19.5 | 11.0 | 25.1 |
| | OpenQA | 26.9 | 20.7 | 16.0 | 27.0 | 20.9 | 15.8 | 27.2 | 21.0 | 16.1 |
| | CloseQA | 8.3 | 8.8 | 14.4 | 9.0 | 9.8 | 14.6 | 9.4 | 10.2 | 14.8 |
| **Class** | News | 96.9 | 6.1 | 12.1 | 98.3 | 6.2 | 12.2 | 98.7 | 6.3 | 12.4 |
| | Movie | 93.1 | 53.1 | 82.1 | 93.0 | 53.6 | 82.2 | 93.3 | 53.9 | 82.5 |
| | QASC | 99.9 | 56.7 | 72.4 | 99.9 | 56.5 | 71.2 | 99.9 | 56.8 | 72.8 |
| **Gen** | Topics | 35.9 | 23.1 | 18.9 | 35.2 | 23.2 | 19.1 | 36.2 | 23.4 | 19.5 |
| | Summary | 50.1 | 17.2 | 11.7 | 49.4 | 17.7 | 12 | 50.2 | 17.8 | 12.2 |
| | Explan | 8.3 | 11.5 | 8.4 | 7.2 | 12.4 | 8.7 | 8.9 | 12.9 | 9.1 |

Table 10: The separate effect of PPO and RRMF, which demonstrate the important roles played by both PPO and RRMF in enhancing the performance of MAPO.

and LLaMA using different synonyms to convey the same information.

- In SNLI, BLOOM, GPT-J, and LLaMA rephrase the sentence 1 by adding additional details related to the slip and slide activity and the celebratory context. The variations in wording are mainly stylistic, enhancing the description of the baby's experience and the wetness.

- In MNLI, BLOOM, GPT-J, and LLaMA maintain the same wording as the original sentence 1. The variations in wording occur in sentence 2, with BLOOM and GPT-J emphasizing the need for interest rates to increase, while LLaMA focuses on the importance of boosting savings.

- In MRPC, BLOOM, GPT-J, and LLaMA all maintain the same wording as the original sentences. The variations in the rephrased sentence 1 (BLOOM and LLaMA) emphasize the 15 percent drop in revenue, while GPT-J maintains a more neutral tone.

# H   Additional Exploratory Analysis

We further analyze the distribution of the top 3 words from the original prompts in the optimized prompts of different LLMs in both the QA and classification tasks as shown in Fig. 8 and Fig. 9, respectively. In the QA task, we observe minimal variations when considering whether to remove the instruction. After prompt optimization, BLOOM has a higher proportion of words like "contemporary", "french", "Methodist", "places", "education", "power", and "life" compared to the other two models. GPT-J has a higher proportion of words like "church", "time", "order", "early", and "year", indicating a focus on temporal and sequential aspects. And LLaMA has a higher proportion of words like "earlier", "similar", "number", "song", and "property" compared to the other two models. In the classification task, we also observe minimal variations when considering whether to remove the instruction. After optimization, BLOOM has a higher proportion of the word "year", "new" compared to the other two models. GPT-J has a higher proportion of words like "largest", "music", "national", "school" and "poland". LLaMA has a higher proportion of words like "increase", "goverment", "executive", "medical", "warsaw", and "parliament" compared to the other two LLMs.

These findings strongly suggest that each LLM exhibits unique preferences and patterns in prompt optimization across different tasks. The observed variations in word distribution clearly indicate the specific areas of focus and the semantic nuances that each LLM emphasizes during the optimization process. Additional experiments will contribute to a more comprehensive understanding of the prompt

| Tasks | Dataset | BLOOM | | | | | GPT-J | | | | | LLaMA | | | | |
|---|---|---|---|---|---|---|---|---|---|---|---|---|---|---|---|---|
| | | 20% | 40% | 60% | 80% | 100% | 20% | 40% | 60% | 80% | 100% | 20% | 40% | 60% | 80% | 100% |
| QA | AdverQA | 17.8 | 18.6 | 18.9 | 19.2 | 19.5 | 9.2 | 9.3 | 9.9 | 11.0 | 11.0 | 22.2 | 23.2 | 25.1 | 25.4 | 26.0 |
| | OpenQA | 25.4 | 26.1 | 26.3 | 26.5 | 27.2 | 19.0 | 19.1 | 19.5 | 21.0 | 21.1 | 14.3 | 15.1 | 16.1 | 16.3 | 16.6 |
| | CloseQA | 8.0 | 8.6 | 8.8 | 9.1 | 9.4 | 8.4 | 8.6 | 9.4 | 10.2 | 10.3 | 13.0 | 13.8 | 14.8 | 15.2 | 15.4 |
| Class | News | 96.8 | 97.4 | 97.6 | 98.0 | 98.7 | 4.6 | 4.8 | 5.3 | 6.3 | 6.4 | 10.2 | 11.5 | 12.4 | 12.5 | 12.7 |
| | Movie | 89.8 | 90.9 | 91.3 | 92.5 | 93.3 | 51.8 | 52.2 | 52.8 | 53.9 | 53.9 | 79.8 | 80.2 | 82.5 | 82.7 | 83.1 |
| | QASC | 98.0 | 98.6 | 98.7 | 98.8 | 99.9 | 53.8 | 54.4 | 55.2 | 56.8 | 56.9 | 70.5 | 71.4 | 72.8 | 73.3 | 73.4 |
| Gen | Topics | 35.0 | 35.7 | 36 | 36.4 | 36.2 | 21.3 | 21.6 | 22.2 | 23.4 | 23.6 | 17.5 | 18.2 | 19.5 | 19.8 | 19.9 |
| | Summary | 47.9 | 48.9 | 49.3 | 49.7 | 50.2 | 16.0 | 16.3 | 16.8 | 17.8 | 18.1 | 10.1 | 11.3 | 12.2 | 12.4 | 12.8 |
| | Explan | 7.5 | 8.1 | 8.3 | 8.6 | 8.9 | 11.2 | 11.4 | 11.9 | 12.9 | 13.2 | 7.3 | 8.2 | 9.1 | 9.5 | 10.0 |
| Average | - | 47.4 | 48.1 | 48.4 | 48.8 | 49.3 | 21.7 | 22.0 | 22.6 | 23.7 | 23.8 | 27.2 | 28.1 | 29.4 | 29.7 | 30.0 |
| ↓ | - | 1.9 | 1.2 | 0.9 | 0.5 | - | 2.1 | 1.9 | 1.3 | 0.1 | - | 2.8 | 1.9 | 0.6 | 0.3 | - |
| ↓(%) | - | 4.0 | 2.4 | 1.9 | 1.0 | - | 9.8 | 8.5 | 5.7 | 0.6 | - | 10.2 | 6.7 | 2.0 | 1 | - |
| Dataset | - | 55822 | 111645 | 167468 | 223291 | 279113 | 55822 | 111645 | 167468 | 223291 | 279113 | 55822 | 111645 | 167468 | 223291 | 279113 |
| D-↓ | - | 223291 | 167468 | 111645 | 55822 | - | 223291 | 167468 | 111645 | 55822 | - | 223291 | 167468 | 111645 | 55822 | - |
| D-↓(%) | - | 400.0 | 150.0 | 66.7 | 25.0 | - | 400.0 | 150.0 | 66.7 | 25.0 | - | 400.0 | 150.0 | 66.7 | 25.0 | - |

Table 11: Performance of different proportion of warm-up dataset in various downstream tasks by three LLMs. Q: QA, C: classification, G:generation. ↓means the number of performance decline. ↓(%) means the percentage of performance decline. D-↓ means the number of data reduction. D-↓(%) means the percentage of data reduction.

| | SST-2 | Yelp P. | MR | CR | RTE | QNLI | SNLI | MNLI | MRPC | Average |
|---|---|---|---|---|---|---|---|---|---|---|
| MAPO | 96.1 | 93.5 | 90.2 | 88.9 | 75.3 | 63.1 | 60 | 55.7 | 79.3 | 78.0 |
| MAPO-0 | 95.8 | 93.1 | 90.2 | 88.2 | 74.9 | 62.9 | 59.7 | 55.1 | 78.9 | 77.6 |
| MAPO-0.2 | 95.9 | 93.3 | 90.3 | 88 | 75.2 | 63 | 59.8 | 54.8 | 78.3 | 77.6 |
| MAPO-0.5 | 95.2 | 92.2 | 88.9 | 87.9 | 74.8 | 62.8 | 59.2 | 55 | 78.8 | 77.2 |
| MAPO-0.8 | 95.3 | 92.3 | 88.7 | 87.9 | 74.5 | 62.7 | 59.2 | 54.9 | 78.7 | 77.1 |

Table 12: The few-shot performance of MAPO with SOTA prompt optimizing baselines in downstream tasks. F: Finetuning, C: Continous prompt, D: Discrete prompt. MAPO means using MAPO with temperature [0,0.5]. MAPO-0 means using MAPO with temperature 0.

| Task | Natural Instructions |
|---|---|
| SST-2 | In this task, you are given sentences from movie reviews. The task is to classify the sentiment of the sentence. Your answer must be in the form of the letters "positive", and "negative" respectively. |
| Yelp | In this task, you are given sentences from Yelp reviews. The task is to classify the sentiment of the sentence. Your answer must be in the form of the letters "positive", or "negative" respectively. |
| MR | In this task, you are given sentences from movie reviews. The task is to classify the sentiment of the sentence. Your answer must be in the form of the letters "positive", or "negative" respectively. |
| CR | In this task, you are given sentences from customer reviews. The task is to classify the sentiment of the sentence. Your answer must be in the form of the letters "positive", or "negative" respectively.' |
| RTE | In this task, you're given a pair of sentences, sentence 1 and sentence 2. Your job is to choose whether the two sentences clearly agree (entailment)/disagree (not entailment) with each other. Your answer must be in the form of the letters Yes, and No respectively. |
| QNLI | You are given two sentences(Sentence1 and Sentence2). The task is to determine whether Sentence2 contains the answer to Sentence1. Your answer must be in the form of the letters Yes, and No respectively. |
| SNLI | In this task, you're given a pair of sentences, sentence 1 and sentence 2. Your job is to choose whether the two sentences clearly agree (entailment)/disagree (contradiction) with each other, or if this cannot be determined (neutral). Your answer must be in the form of the letters "Yes", "No", and "Maybe" respectively. |
| MNLI | In this task, you're given a pair of sentences, sentence 1 and sentence 2. Your job is to choose whether the two sentences clearly agree (entailment)/disagree (contradiction) with each other, or if this cannot be determined (neutral). Your answer must be in the form of the letters Yes, No, and Maybe respectively. |
| MRPC | In this task, you're given a pair of sentences, sentence 1 and sentence 2. Your job is to choose whether the two sentences clearly agree (entailment)/disagree (not entailment) with each other. Your answer must be in the form of the letters Yes, and No respectively. |

Table 13: Natural Instructions of various downstream tasks.

optimization dynamics exhibited by each LLM.

| Task | Dataset | M | | | | | | | | | | |
|------|---------|------|-----|-------|-------|-----|--------|------|------|--------|------|--------|
| | | **BLOOM** | (↑) | (↑(%)) | **GPT-J** | (↑) | (↑ (%)) | **LLaMA** | (↑) | (↑ (%)) | ↑ | ↑ (%) |
| QA | AdverQA | 19.5 | 6 | 44.4 | 11 | 8 | 266.7 | 25.1 | 21.9 | 684.4 | 12 | 331.8 |
| | OpenQA | 27.2 | 1.3 | 5 | 21 | 4 | 23.5 | 16.1 | 2.8 | 21.1 | 2.7 | 16.5 |
| | CloseQA | 9.4 | 3 | 46.9 | 10.2 | 3.3 | 47.8 | 14.8 | 4 | 37 | 3.4 | 43.9 |
| CLS | News | 98.7 | 5.9 | 6.4 | 6.3 | 6.3 | - | 12.4 | 11.3 | 1027.3 | 7.8 | 516.9 |
| | Movie | 93.3 | 2.4 | 2.6 | 53.9 | 2.8 | 5.5 | 82.5 | 3.8 | 4.8 | 3 | 4.3 |
| | QASC | 99.9 | 0.5 | 0.5 | 56.8 | 2.8 | 5.2 | 72.8 | 11.2 | 18.2 | 4.8 | 8 |
| GEN | Topics | 36.2 | 6.7 | 22.7 | 23.4 | 5.9 | 33.7 | 19.5 | 5.2 | 36.4 | 5.9 | 30.9 |
| | Summary | 50.2 | 4.1 | 8.9 | 17.8 | 4.7 | 35.9 | 12.2 | 5.6 | 84.8 | 4.8 | 43.2 |
| | Explan | 8.9 | 3.2 | 56.1 | 12.9 | 4.4 | 51.8 | 9.1 | 2.2 | 31.9 | 3.3 | 46.6 |

| Task | Dataset | M-0 | | | | | | | | | | |
|------|---------|------|-----|-------|-------|-----|--------|------|------|--------|------|--------|
| | | **BLOOM** | (↑) | (↑(%)) | **GPT-J** | (↑) | (↑ (%)) | **LLaMA** | (↑) | (↑ (%)) | ↑ | ↑ (%) |
| QA | AdverQA | 19.3 | 5.8 | 43 | 10.5 | 7.5 | 250 | 24.8 | 21.6 | 675 | 11.6 | 322.7 |
| | OpenQA | 26.8 | 0.9 | 3.5 | 20.9 | 3.9 | 22.9 | 15.2 | 1.9 | 14.3 | 2.2 | 13.6 |
| | CloseQA | 8.9 | 2.5 | 39.1 | 9.8 | 2.9 | 42 | 14 | 3.2 | 29.6 | 2.9 | 36.9 |
| CLS | News | 96.5 | 3.7 | 4 | 5.9 | 5.9 | - | 11.9 | 10.8 | 981.8 | 6.8 | 492.9 |
| | Movie | 92.8 | 1.9 | 2.1 | 53.6 | 2.5 | 4.9 | 82 | 3.3 | 4.2 | 2.6 | 3.7 |
| | QASC | 99.9 | 0.5 | 0.5 | 56.6 | 2.6 | 4.8 | 72.2 | 10.6 | 17.2 | 4.6 | 7.5 |
| GEN | Topics | 35.9 | 6.4 | 21.7 | 23 | 5.5 | 31.4 | 19 | 4.7 | 32.9 | 5.5 | 28.7 |
| | Summary | 49.7 | 3.6 | 7.8 | 17 | 3.9 | 29.8 | 11.9 | 5.3 | 80.3 | 4.3 | 39.3 |
| | Explan | 7.8 | 2.1 | 36.8 | 12.3 | 3.8 | 44.7 | 8.8 | 1.9 | 27.5 | 2.6 | 36.3 |

| Task | Dataset | M-0.2 | | | | | | | | | | |
|------|---------|------|-----|-------|-------|-----|--------|------|------|--------|------|--------|
| | | **BLOOM** | (↑) | (↑(%)) | **GPT-J** | (↑) | (↑ (%)) | **LLaMA** | (↑) | (↑ (%)) | ↑ | ↑ (%) |
| QA | AdverQA | 19 | 5.5 | 40.7 | 10.7 | 7.7 | 256.7 | 24.4 | 21.2 | 662.5 | 11.5 | 320 |
| | OpenQA | 26.9 | 1 | 3.9 | 20.1 | 3.1 | 18.2 | 15.6 | 2.3 | 17.3 | 2.1 | 13.1 |
| | CloseQA | 8.4 | 2 | 31.3 | 10 | 3.1 | 44.9 | 14.2 | 3.4 | 31.5 | 2.8 | 35.9 |
| CLS | News | 96.7 | 3.9 | 4.2 | 6.1 | 6.1 | - | 12.1 | 11 | 1000 | 7 | 502.1 |
| | Movie | 92.9 | 2 | 2.2 | 53.2 | 2.1 | 4.1 | 81.5 | 2.8 | 3.6 | 2.3 | 3.3 |
| | QASC | 99.9 | 0.5 | 0.5 | 56.6 | 2.6 | 4.8 | 72.3 | 10.7 | 17.4 | 4.6 | 7.6 |
| GEN | Topics | 36 | 6.5 | 22 | 23.3 | 5.8 | 33.1 | 19.3 | 5 | 35 | 5.8 | 30 |
| | Summary | 50.2 | 4.1 | 8.9 | 16.6 | 3.5 | 26.7 | 12 | 5.4 | 81.8 | 4.3 | 39.1 |
| | Explan | 8.2 | 2.5 | 43.9 | 12.5 | 4 | 47.1 | 9 | 2.1 | 30.4 | 2.9 | 40.5 |

| Task | Dataset | M-0.5 | | | | | | | | | | |
|------|---------|------|-----|-------|-------|-----|--------|------|------|--------|------|--------|
| | | **BLOOM** | (↑) | (↑(%)) | **GPT-J** | (↑) | (↑ (%)) | **LLaMA** | (↑) | (↑ (%)) | ↑ | ↑ (%) |
| QA | AdverQA | 18.8 | 5.3 | 39.3 | 10 | 7 | 233.3 | 23.6 | 20.4 | 637.5 | 10.9 | 303.4 |
| | OpenQA | 27.1 | 1.2 | 4.6 | 20.4 | 3.4 | 20 | 15.8 | 2.5 | 18.8 | 2.4 | 14.5 |
| | CloseQA | 8.3 | 1.9 | 29.7 | 8.5 | 1.6 | 23.2 | 14.2 | 3.4 | 31.5 | 2.3 | 28.1 |
| CLS | News | 96.2 | 3.4 | 3.7 | 5.7 | 5.7 | - | 10.3 | 9.2 | 836.4 | 6.1 | 420.1 |
| | Movie | 93.2 | 2.3 | 2.5 | 53 | 1.9 | 3.7 | 81.6 | 2.9 | 3.7 | 2.4 | 3.3 |
| | QASC | 99.9 | 0.5 | 0.5 | 56.5 | 2.5 | 4.6 | 70.6 | 9 | 14.6 | 4 | 6.6 |
| GEN | Topics | 35.5 | 6 | 20.3 | 21.9 | 4.4 | 25.1 | 18.9 | 4.6 | 32.2 | 5 | 25.9 |
| | Summary | 49.2 | 3.1 | 6.7 | 17 | 3.9 | 29.8 | 11.1 | 4.5 | 68.2 | 3.8 | 34.9 |
| | Explan | 7.1 | 1.4 | 24.6 | 11.1 | 2.6 | 30.6 | 8.5 | 1.6 | 23.2 | 1.9 | 26.1 |

| Task | Dataset | M-0.8 | | | | | | | | | | |
|------|---------|------|-----|-------|-------|-----|--------|------|------|--------|------|--------|
| | | **BLOOM** | (↑) | (↑(%)) | **GPT-J** | (↑) | (↑ (%)) | **LLaMA** | (↑) | (↑ (%)) | ↑ | ↑ (%) |
| QA | AdverQA | 18.3 | 4.8 | 35.6 | 9.8 | 6.8 | 226.7 | 23.3 | 20.1 | 628.1 | 10.6 | 296.8 |
| | OpenQA | 26.7 | 0.8 | 3.1 | 20.6 | 3.6 | 21.2 | 15.6 | 2.3 | 17.3 | 2.2 | 13.9 |
| | CloseQA | 7.9 | 1.5 | 23.4 | 8.6 | 1.7 | 24.6 | 14.1 | 3.3 | 30.6 | 2.2 | 26.2 |
| CLS | News | 95.9 | 3.1 | 3.3 | 5.8 | 5.8 | - | 10.2 | 9.1 | 827.3 | 6 | 415.3 |
| | Movie | 92.8 | 1.9 | 2.1 | 52.9 | 1.8 | 3.5 | 81.5 | 2.8 | 3.6 | 2.2 | 3.1 |
| | QASC | 99.9 | 0.5 | 0.5 | 56.6 | 2.6 | 4.8 | 70.4 | 8.8 | 14.3 | 4 | 6.5 |
| GEN | Topics | 35.1 | 5.6 | 19 | 21.9 | 4.4 | 25.1 | 18.7 | 4.4 | 30.8 | 4.8 | 25 |
| | Summary | 50.1 | 4 | 8.7 | 16.8 | 3.7 | 28.2 | 10.9 | 4.3 | 65.2 | 4 | 34 |
| | Explan | 7.1 | 1.4 | 24.6 | 10.9 | 2.4 | 28.2 | 8.2 | 1.3 | 18.8 | 1.7 | 23.9 |

Table 14: (↑) denotes the absolute performance increase achieved using MAPO-optimized prompts versus a frozen LLM, while (↑(%)) highlights the relative performance boost. Symbols ↑ and ↑(%) represent the average absolute and relative enhancements across all three LLMs, respectively. These enhancements pertain to specific downstream tasks, with "CLS" signifying classification and "GEN" indicating generation tasks. M, M-0.2, M-0.5, and M-0.8 correspond to using MAPO with temperature settings of [0,0.5], 0, 0.2, 0.5, and 0.8, respectively.

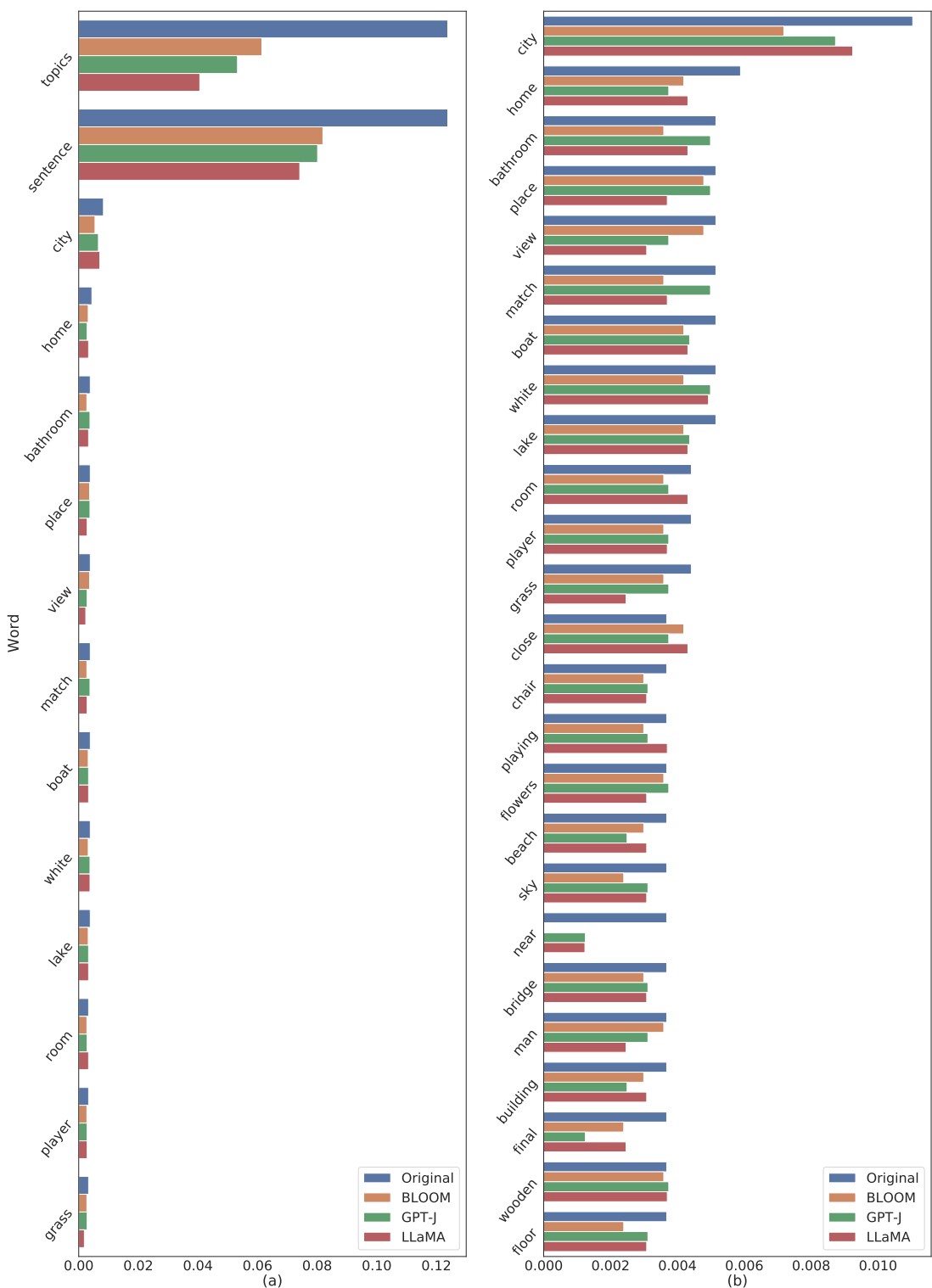

Figure 7: The distribution of three most frequent words, which extracted from the original prompt, in the optimized prompt among different LLMs in the generation task. (a) retaining frequent words in the instruction, (b) removing frequent words in the instruction.

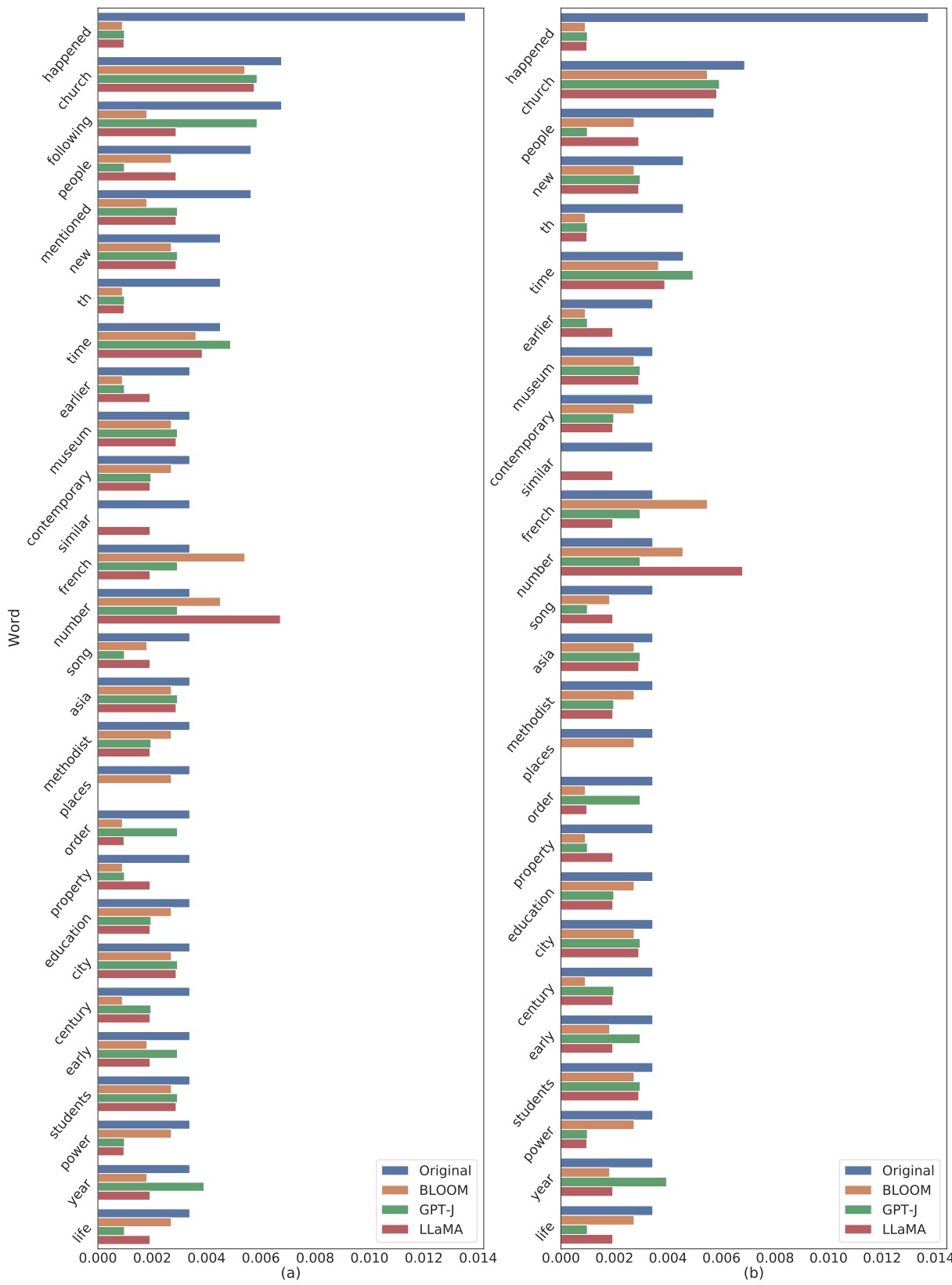

Figure 8: The distribution of three most frequent words, which extracted from the original prompt, in the optimized prompt among different LLMs in the QA task. (a) retaining frequent words in the instruction, (b) removing frequent words in the instruction.

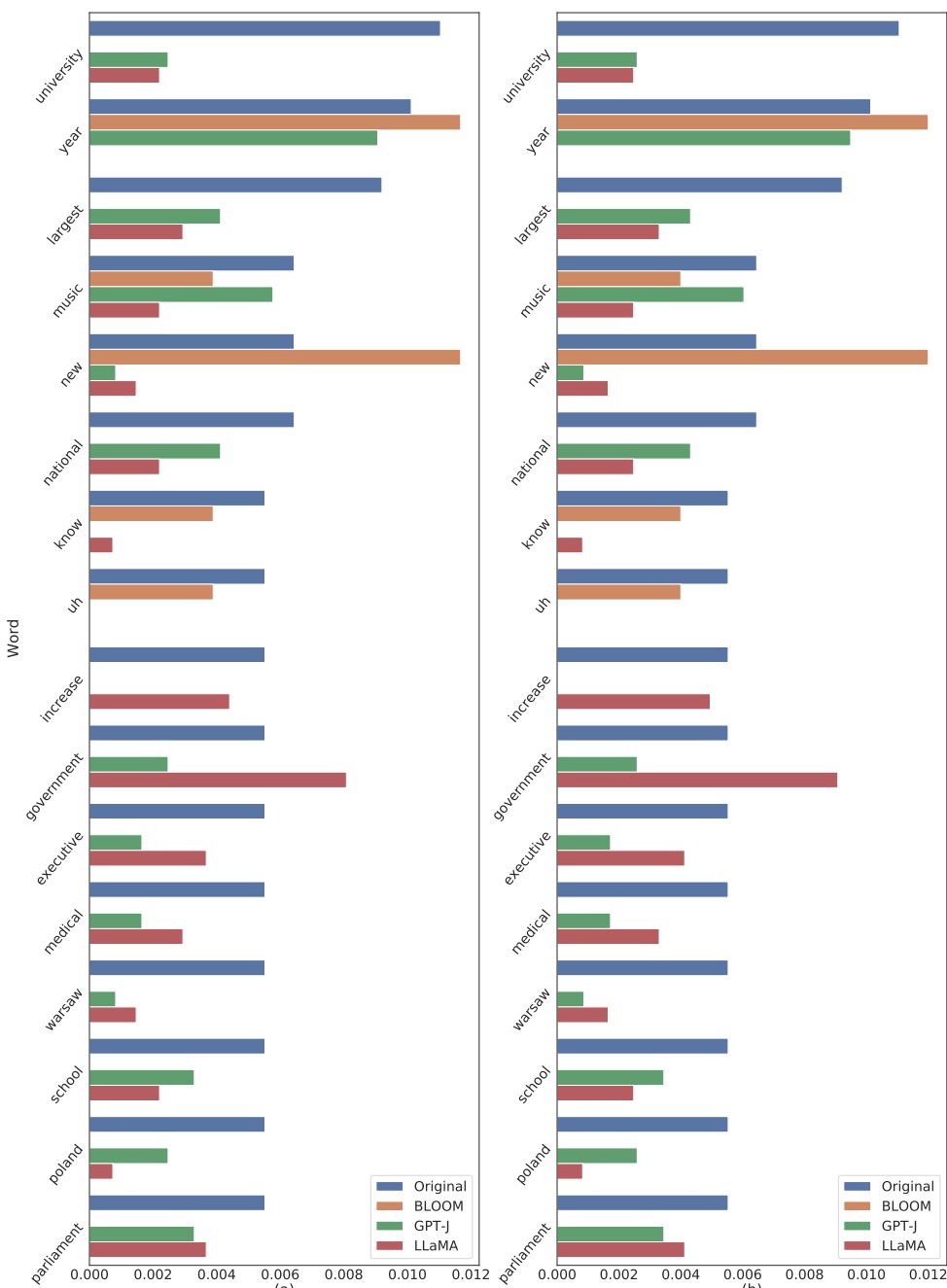

Figure 9: The distribution of three most frequent words, which extracted from the original prompt, in the optimized prompt among different LLMs in the classification task. (a) retaining frequent words in the instruction, (b) removing frequent words in the instruction.

## Table 15 (BLOOM and GPT-J)

**MAPO (M)**

| Task | Dataset | BLOOM | M | ↑ | ↑ | ↑(%) | ↑(%) | GPT-J | M | (↑) | ↑ | (↑(%)) | ↑(%) |
|---|---|---|---|---|---|---|---|---|---|---|---|---|---|
| Coref. | xwinograd | 60.1 | 60.6 | 0.5 | 0.5 | 0.9 | 0.9 | - | - | - | - | - | - |
| | BoolQ | 67.9 | 68.2 | 0.3 | | 0.4 | | 67.2 | 67.9 | 0.7 | 0.3 | 1 | 0.5 |
| | CB | 77.6 | 78.1 | 0.5 | | 0.6 | | 83.9 | 84.2 | 0.3 | | 0.4 | |
| | COPA | 74.0 | 75.0 | 1.0 | | 1.4 | | 84 | 84.2 | 0.2 | | 0.2 | |
| NLU | MultiRC | 59.7 | 60.4 | 0.7 | 0.6 | 1.2 | 0.9 | 63.8 | 64.1 | 0.3 | | 0.5 | |
| | ReCoRD | 69.8 | 70.2 | 0.4 | | 0.6 | | 74.4 | 74.7 | 0.3 | | 0.4 | |
| | WiC | 61.4 | 62.0 | 0.6 | | 1.0 | | 61 | 61.3 | 0.3 | | 0.5 | |
| | WSC | 64.4 | 65.1 | 0.7 | | 1.1 | | 77.9 | 78.1 | 0.2 | | 0.3 | |
| | ANLI R1 | 31.5 | 32.1 | 0.6 | | 1.9 | | 37.8 | 38.2 | 0.4 | 0.3 | 1.1 | 0.7 |
| NLI | ANLI R2 | 35.1 | 35.4 | 0.3 | 0.5 | 0.9 | 1.3 | 37.9 | 38.3 | 0.4 | | 1.1 | |
| | ANLI R3 | 37.1 | 37.8 | 0.7 | | 1.9 | | 40.9 | 41.1 | 0.2 | | 0.5 | |
| | StoryCloze | 79.0 | 79.5 | 0.5 | | 0.6 | | 87.8 | 87.9 | 0.1 | | 0.1 | |
| CLS | Amazon | 65.2 | 67.7 | 2.5 | 2.3 | 3.8 | 3.3 | 68.2 | 69.4 | 1.2 | 1.2 | 1.8 | 1.6 |
| | DBPedia | 70.5 | 72.5 | 2.0 | | 2.8 | | 83.9 | 85.1 | 1.2 | | 1.4 | |
| | DROP | 67.9 | 69.9 | 2.0 | | 2.9 | | 51.6 | 52.8 | 1.2 | 1 | 2.3 | 3.2 |
| QA | NQ | 15.1 | 16.1 | 1.0 | 1.8 | 6.6 | 5.6 | 19.6 | 20.8 | 1.2 | | 6.1 | |
| | RealTimeQA | 29.0 | 31.5 | 2.5 | | 8.6 | | 36 | 37.2 | 1.2 | | 3.3 | |
| | WebQs | 34.8 | 36.3 | 1.5 | | 4.3 | | 44.1 | 44.6 | 0.5 | | 1.1 | |

**MAPO (M-0)**

| Task | Dataset | BLOOM | M-0 | ↑ | ↑ | ↑(%) | ↑(%) | GPT-J | M-0 | (↑) | ↑ | (↑(%)) | ↑(%) |
|---|---|---|---|---|---|---|---|---|---|---|---|---|---|
| Coref. | xwinograd | 60.1 | 60.5 | 0.4 | 0.4 | 0.7 | 0.7 | - | - | - | - | - | - |
| | BoolQ | 67.9 | 68.0 | 0.1 | | 0.1 | | 67.2 | 67.8 | 0.6 | 0.3 | 0.9 | 0.4 |
| | CB | 77.6 | 78.0 | 0.4 | | 0.5 | | 83.9 | 84.1 | 0.2 | | 0.2 | |
| | COPA | 74.0 | 74.8 | 0.8 | | 1.1 | | 84 | 84.2 | 0.2 | | 0.2 | |
| NLU | MultiRC | 59.7 | 60.4 | 0.7 | 0.5 | 1.2 | 0.7 | 63.8 | 64.1 | 0.3 | | 0.5 | |
| | ReCoRD | 69.8 | 70.2 | 0.4 | | 0.6 | | 74.4 | 74.7 | 0.3 | | 0.4 | |
| | WiC | 61.4 | 61.8 | 0.4 | | 0.7 | | 61 | 61.3 | 0.3 | | 0.5 | |
| | WSC | 64.4 | 65.0 | 0.6 | | 0.9 | | 77.9 | 78.1 | 0.2 | | 0.3 | |
| | ANLI R1 | 31.5 | 32.0 | 0.5 | | 1.6 | | 37.8 | 38.2 | 0.4 | 0.2 | 1.1 | 0.6 |
| NLI | ANLI R2 | 35.1 | 35.4 | 0.3 | 0.5 | 0.9 | 1.2 | 37.9 | 38.1 | 0.2 | | 0.5 | |
| | ANLI R3 | 37.1 | 37.8 | 0.7 | | 1.9 | | 40.9 | 41.1 | 0.2 | | 0.5 | |
| | StoryCloze | 79.0 | 79.4 | 0.4 | | 0.5 | | 87.8 | 87.9 | 0.1 | | 0.1 | |
| CLS | Amazon | 65.2 | 67.3 | 2.1 | 1.8 | 3.2 | 2.7 | 68.2 | 69 | 0.8 | 1 | 1.2 | 1.3 |
| | DBPedia | 70.5 | 72.0 | 1.5 | | 2.1 | | 83.9 | 85.1 | 1.2 | | 1.4 | |
| | DROP | 67.9 | 69.5 | 1.6 | | 2.4 | | 51.6 | 52.3 | 0.7 | 0.7 | 1.4 | 2.3 |
| QA | NQ | 15.1 | 15.8 | 0.7 | 1.5 | 4.6 | 4.6 | 19.6 | 20.4 | 0.8 | | 4.1 | |
| | RealTimeQA | 29.0 | 31.2 | 2.2 | | 7.6 | | 36 | 37 | 1 | | 2.8 | |
| | WebQs | 34.8 | 36.1 | 1.3 | | 3.7 | | 44.1 | 44.5 | 0.4 | | 0.9 | |

**MAPO (M-0.2)**

| Task | Dataset | BLOOM | M-0.2 | ↑ | ↑ | ↑(%) | ↑(%) | GPT-J | M-0.2 | (↑) | ↑ | (↑(%)) | ↑(%) |
|---|---|---|---|---|---|---|---|---|---|---|---|---|---|
| Coref. | xwinograd | 60.1 | 60.6 | 0.5 | 0.5 | 0.8 | 0.8 | - | - | - | - | - | - |
| | BoolQ | 67.9 | 68.1 | 0.2 | | 0.3 | | 67.2 | 67.9 | 0.7 | 0.3 | 1 | 0.4 |
| | CB | 77.6 | 78.1 | 0.5 | | 0.6 | | 83.9 | 84.2 | 0.3 | | 0.4 | |
| | COPA | 74.0 | 74.6 | 0.6 | | 0.8 | | 84 | 84.2 | 0.2 | | 0.2 | |
| NLU | MultiRC | 59.7 | 60.3 | 0.6 | 0.5 | 1.0 | 0.7 | 63.8 | 64 | 0.2 | | 0.3 | |
| | ReCoRD | 69.8 | 70.2 | 0.4 | | 0.6 | | 74.4 | 74.6 | 0.2 | | 0.3 | |
| | WiC | 61.4 | 61.7 | 0.3 | | 0.5 | | 61 | 61.3 | 0.3 | | 0.5 | |
| | WSC | 64.4 | 65.1 | 0.7 | | 1.1 | | 77.9 | 78.1 | 0.2 | | 0.3 | |
| | ANLI R1 | 31.5 | 32.2 | 0.7 | | 2.2 | | 37.8 | 38.2 | 0.4 | 0.3 | 1.1 | 0.7 |
| NLI | ANLI R2 | 35.1 | 35.5 | 0.4 | 0.5 | 1.1 | 1.3 | 37.9 | 38.3 | 0.4 | | 1.1 | |
| | ANLI R3 | 37.1 | 37.6 | 0.5 | | 1.3 | | 40.9 | 41.1 | 0.2 | | 0.5 | |
| | StoryCloze | 79.0 | 79.5 | 0.5 | | 0.6 | | 87.8 | 87.9 | 0.1 | | 0.1 | |
| CLS | Amazon | 65.2 | 67.1 | 1.9 | 1.6 | 2.9 | 2.4 | 68.2 | 69.1 | 0.9 | 1 | 1.3 | 1.3 |
| | DBPedia | 70.5 | 71.8 | 1.3 | | 1.8 | | 83.9 | 85 | 1.1 | | 1.3 | |
| | DROP | 67.9 | 69.3 | 1.4 | | 2.1 | | 51.6 | 52.6 | 1 | 0.7 | 1.9 | 2.1 |
| QA | NQ | 15.1 | 15.6 | 0.5 | 1.2 | 3.3 | 3.8 | 19.6 | 20.3 | 0.7 | | 3.6 | |
| | RealTimeQA | 29.0 | 31.0 | 2.0 | | 6.9 | | 36 | 36.7 | 0.7 | | 1.9 | |
| | WebQs | 34.8 | 35.8 | 1.0 | | 2.9 | | 44.1 | 44.5 | 0.4 | | 0.9 | |

**MAPO (M-0.5)**

| Task | Dataset | BLOOM | M-0.5 | ↑ | ↑ | ↑(%) | ↑(%) | GPT-J | M-0.5 | (↑) | ↑ | (↑(%)) | ↑(%) |
|---|---|---|---|---|---|---|---|---|---|---|---|---|---|
| Coref. | xwinograd | 60.1 | 60.4 | 0.3 | 0.3 | 0.5 | 0.5 | - | - | - | - | - | - |
| | BoolQ | 67.9 | 68.0 | 0.1 | | 0.1 | | 67.2 | 67.6 | 0.4 | 0.2 | 0.6 | 0.2 |
| | CB | 77.6 | 77.8 | 0.2 | | 0.3 | | 83.9 | 84.1 | 0.2 | | 0.2 | |
| | COPA | 74.0 | 74.4 | 0.4 | | 0.4 | | 84 | 84.2 | 0.2 | | 0.2 | |
| NLU | MultiRC | 59.7 | 60.3 | 0.6 | 0.3 | 1.0 | 0.5 | 63.8 | 63.9 | 0.1 | | 0.2 | |
| | ReCoRD | 69.8 | 70.1 | 0.3 | | 0.4 | | 74.4 | 74.5 | 0.1 | | 0.1 | |
| | WiC | 61.4 | 61.6 | 0.2 | | 0.3 | | 61 | 61.2 | 0.2 | | 0.3 | |
| | WSC | 64.4 | 64.8 | 0.4 | | 0.6 | | 77.9 | 78 | 0.1 | | 0.1 | |
| | ANLI R1 | 31.5 | 32.0 | 0.5 | | 1.6 | | 37.8 | 38.2 | 0.4 | 0.2 | 1.1 | 0.5 |
| NLI | ANLI R2 | 35.1 | 35.3 | 0.2 | 0.3 | 0.6 | 0.9 | 37.9 | 38.1 | 0.2 | | 0.5 | |
| | ANLI R3 | 37.1 | 37.5 | 0.4 | | 1.1 | | 40.9 | 41 | 0.1 | | 0.2 | |
| | StoryCloze | 79.0 | 79.3 | 0.3 | | 0.4 | | 87.8 | 87.9 | 0.1 | | 0.1 | |
| CLS | Amazon | 65.2 | 66.7 | 1.5 | 1.2 | 2.3 | 1.8 | 68.2 | 68.9 | 0.7 | 0.5 | 1 | 0.8 |
| | DBPedia | 70.5 | 71.4 | 0.9 | | 1.3 | | 83.9 | 84.3 | 0.4 | | 0.5 | |
| | DROP | 67.9 | 68.5 | 0.6 | | 0.9 | | 51.6 | 52.2 | 0.6 | 0.5 | 1.2 | 1.7 |
| QA | NQ | 15.1 | 15.5 | 0.4 | 0.8 | 2.6 | 2.7 | 19.6 | 20.2 | 0.6 | | 3.1 | |
| | RealTimeQA | 29.0 | 30.6 | 1.6 | | 5.5 | | 36 | 36.6 | 0.6 | | 1.7 | |
| | WebQs | 34.8 | 35.4 | 0.6 | | 1.7 | | 44.1 | 44.5 | 0.4 | | 0.7 | |

**MAPO (M-0.8)**

| Task | Dataset | BLOOM | M-0.8 | ↑ | ↑ | ↑(%) | ↑(%) | GPT-J | M-0.8 | (↑) | ↑ | (↑(%)) | ↑(%) |
|---|---|---|---|---|---|---|---|---|---|---|---|---|---|
| Coref. | xwinograd | 60.1 | 60.2 | 0.1 | 0.1 | 0.2 | 0.2 | - | - | - | - | - | - |
| | BoolQ | 67.9 | 68.0 | 0.1 | | 0.1 | | 67.2 | 67.5 | 0.3 | 0.2 | 0.4 | 0.2 |
| | CB | 77.6 | 77.8 | 0.2 | | 0.3 | | 83.9 | 84.1 | 0.2 | | 0.2 | |
| | COPA | 74.0 | 74.3 | 0.3 | | 0.4 | | 84 | 84.2 | 0.2 | | 0.2 | |
| NLU | MultiRC | 59.7 | 60.3 | 0.6 | 0.3 | 1.0 | 0.4 | 63.8 | 63.9 | 0.1 | | 0.2 | |
| | ReCoRD | 69.8 | 70.1 | 0.3 | | 0.4 | | 74.4 | 74.5 | 0.1 | | 0.1 | |
| | WiC | 61.4 | 61.6 | 0.2 | | 0.3 | | 61 | 61.2 | 0.2 | | 0.3 | |
| | WSC | 64.4 | 64.7 | 0.3 | | 0.5 | | 77.9 | 78 | 0.1 | | 0.1 | |
| | ANLI R1 | 31.5 | 31.8 | 0.3 | | 1.0 | | 37.8 | 38.1 | 0.3 | 0.2 | 0.8 | 0.4 |
| NLI | ANLI R2 | 35.1 | 35.3 | 0.2 | 0.3 | 0.6 | 0.8 | 37.9 | 38 | 0.1 | | 0.3 | |
| | ANLI R3 | 37.1 | 37.5 | 0.4 | | 1.1 | | 40.9 | 41 | 0.1 | | 0.2 | |
| | StoryCloze | 79.0 | 79.3 | 0.3 | | 0.4 | | 87.8 | 87.9 | 0.1 | | 0.1 | |
| CLS | Amazon | 65.2 | 66.5 | 1.3 | 1.1 | 2.0 | 1.6 | 68.2 | 68.8 | 0.6 | 0.4 | 0.9 | 0.7 |
| | DBPedia | 70.5 | 71.3 | 0.8 | | 1.1 | | 83.9 | 84.2 | 0.3 | | 0.4 | |
| | DROP | 67.9 | 68.3 | 0.4 | | 0.6 | | 51.6 | 52.1 | 0.5 | 0.4 | 1 | 1.5 |
| QA | NQ | 15.1 | 15.5 | 0.4 | 0.7 | 2.6 | 2.5 | 19.6 | 20.1 | 0.5 | | 2.6 | |
| | RealTimeQA | 29.0 | 30.5 | 1.5 | | 5.2 | | 36 | 36.6 | 0.6 | | 1.7 | |
| | WebQs | 34.8 | 35.3 | 0.5 | | 1.4 | | 44.1 | 44.3 | 0.2 | | 0.5 | |

Table 15: Zero-shot domain transfer performance based on BLOOM and GPT-J with original and MAPO-Optimized prompts. CLS: Classification, M: MAPO. The (↑ ) and ↑ represent the absolute improvement scores of MAPO-optimized prompts compared with original prompts in each dataset and task, respectively. The (↑(%) ) and ↑(%) represent the relative improvement percentages of MAPO-optimized prompts compared with original prompts in each dataset and task, respectively.

## Table 16 (LLaMA)

**MAPO (M)**

| Task | Dataset | LLaMA | M | (↑) | ↑ | (↑(%)) | ↑(%) |
|---|---|---|---|---|---|---|---|
| RS | BoolQ | 76.5 | 76.7 | 0.2 | 0.3 | 0.3 | 0.5 |
| | PIQA | 79.8 | 80 | 0.2 | | 0.3 | |
| | SIQA | 48.9 | 49 | 0.1 | | 0.2 | |
| | HellaSwag | 76.1 | 76.5 | 0.4 | | 0.5 | |
| | WinoGrande | 70.1 | 70.5 | 0.4 | | 0.6 | |
| | ARC-e | 72.8 | 73.2 | 0.4 | | 0.5 | |
| | ARC-c | 47.6 | 47.8 | 0.2 | | 0.4 | |
| | OBQA | 57.2 | 57.9 | 0.7 | | 1.2 | |
| QA | NQ | 16.8 | 18.1 | 1.3 | 1.1 | 7.7 | 4.7 |
| | RACE | 50 | 50.8 | 0.8 | | 1.6 | |

**MAPO (M-0)**

| Task | Dataset | LLaMA | M-0 | (↑) | ↑ | (↑(%)) | ↑(%) |
|---|---|---|---|---|---|---|---|
| RS | BoolQ | 76.5 | 76.7 | 0.2 | 0.3 | 0.3 | 0.4 |
| | PIQA | 79.8 | 80 | 0.2 | | 0.3 | |
| | SIQA | 48.9 | 49 | 0.1 | | 0.2 | |
| | HellaSwag | 76.1 | 76.4 | 0.3 | | 0.4 | |
| | WinoGrande | 70.1 | 70.4 | 0.3 | | 0.4 | |
| | ARC-e | 72.8 | 73.2 | | | 0.5 | |
| | ARC-c | 47.6 | 47.8 | 0.2 | | 0.4 | |
| | OBQA | 57.2 | 57.8 | 0.6 | | 1 | |
| QA | NQ | 16.8 | 17.8 | 1 | 0.8 | 6 | 3.6 |
| | RACE | 50 | 50.6 | 0.6 | | 1.2 | |

**MAPO (M-0.2)**

| Task | Dataset | LLaMA | M-0.2 | (↑) | ↑ | (↑(%)) | ↑(%) |
|---|---|---|---|---|---|---|---|
| RS | BoolQ | 76.5 | 76.7 | 0.2 | 0.3 | 0.3 | 0.4 |
| | PIQA | 79.8 | 80 | 0.2 | | 0.3 | |
| | SIQA | 48.9 | 49 | 0.1 | | 0.2 | |
| | HellaSwag | 76.1 | 76.4 | 0.3 | | 0.4 | |
| | WinoGrande | 70.1 | 70.4 | 0.3 | | 0.4 | |
| | ARC-e | 72.8 | 73.1 | 0.3 | | 0.4 | |
| | ARC-c | 47.6 | 47.8 | 0.2 | | 0.4 | |
| | OBQA | 57.2 | 57.7 | 0.5 | | 0.9 | |
| QA | NQ | 16.8 | 18 | 1.2 | 1 | 7.1 | 4.3 |
| | RACE | 50 | 50.7 | 0.7 | | 1.4 | |

**MAPO (M-0.5)**

| Task | Dataset | LLaMA | M-0.5 | (↑) | ↑ | (↑(%)) | ↑(%) |
|---|---|---|---|---|---|---|---|
| RS | BoolQ | 76.5 | 76.7 | 0.2 | 0.2 | 0.3 | 0.3 |
| | PIQA | 79.8 | 79.9 | 0.1 | | 0.1 | |
| | SIQA | 48.9 | 49 | 0.1 | | 0.2 | |
| | HellaSwag | 76.1 | 76.4 | 0.3 | | 0.4 | |
| | WinoGrande | 70.1 | 70.4 | 0.3 | | 0.4 | |
| | ARC-e | 72.8 | 73 | 0.2 | | 0.3 | |
| | ARC-c | 47.6 | 47.7 | 0.1 | | 0.2 | |
| | OBQA | 57.2 | 57.6 | 0.4 | | 0.7 | |
| QA | NQ | 16.8 | 17.7 | 0.9 | 0.7 | 5.4 | 3.2 |
| | RACE | 50 | 50.5 | 0.5 | | 1 | |

**MAPO (M-0.8)**

| Task | Dataset | LLaMA | M-0.8 | (↑) | ↑ | (↑(%)) | ↑(%) |
|---|---|---|---|---|---|---|---|
| RS | BoolQ | 76.5 | 76.6 | 0.1 | 0.2 | 0.1 | 0.2 |
| | PIQA | 79.8 | 79.9 | 0.1 | | 0.1 | |
| | SIQA | 48.9 | 49 | 0.1 | | 0.2 | |
| | HellaSwag | 76.1 | 76.3 | 0.2 | | 0.3 | |
| | WinoGrande | 70.1 | 70.4 | 0.3 | | 0.4 | |
| | ARC-e | 72.8 | 73 | 0.2 | | 0.3 | |
| | ARC-c | 47.6 | 47.6 | 0 | | 0 | |
| | OBQA | 57.2 | 57.5 | 0.3 | | 0.5 | |
| QA | NQ | 16.8 | 17.6 | 0.8 | 0.5 | 4.8 | 2.7 |
| | RACE | 50 | 50.3 | 0.3 | | 0.6 | |

Table 16: Domain transfer performance with a frozen LLM for inference based on LLaMA with original and MAPO-Optimized prompts. CLS: Classification, M: MAPO. The (↑ ) and ↑ represent the absolute improvement scores of MAPO-optimized prompts compared with original prompts in each dataset and task, respectively. The (↑(%) ) and ↑(%) represent the relative improvement percentages of MAPO-optimized prompts compared with original prompts in each dataset and task, respectively.

| Tasks | Dataset | 20% | | 40% | | 60% | | 80% | | 100% | |
|---|---|---|---|---|---|---|---|---|---|---|---|
| **(BLOOM)** | | +SFT | +MAPO | +SFT | +MAPO | +SFT | +MAPO | +SFT | +MAPO | +SFT | +MAPO |
| QA | AdverQA | 14.6 | 17.8 | 16.9 | 18.6 | 17.5 | 18.9 | 17.7 | 19.2 | **18.3** | **19.5** |
| | OpenQA | 22.1 | 25.4 | 24.3 | 26.1 | 24.9 | 26.3 | 25.1 | 26.5 | **26.7** | **27.2** |
| | CloseQA | 6.6 | 8.0 | 7.7 | 8.6 | 7.3 | 8.8 | 7.8 | 9.1 | **7.8** | **9.4** |
| Class | News | 93.6 | 96.8 | 95.8 | 97.4 | 95.5 | 97.6 | 96.2 | 98.0 | 95.5 | **98.7** |
| | Movie | 87.1 | 89.8 | 88.3 | 90.9 | 88.9 | 91.3 | 90.7 | 92.5 | **92.6** | **93.3** |
| | QASC | 96.0 | 98.0 | 97.6 | 98.6 | 97.1 | 98.7 | 97.4 | 98.8 | **99.9** | **99.9** |
| Gen | Topics | 32.7 | 35.0 | 33.6 | 35.7 | 34.6 | 36.0 | 34.7 | 36.4 | **34.8** | **36.2** |
| | Summary | 45.8 | 47.9 | 47.1 | 48.9 | 47.3 | 49.3 | 48.5 | 49.7 | **48.8** | **50.2** |
| | Explan | 6.2 | 7.5 | 7.3 | 8.1 | 7.1 | 8.3 | 6.3 | 8.6 | **6.8** | **8.9** |
| Average | - | 45.0 | 47.4 | 46.5 | 48.1 | 46.7 | 48.4 | 47.2 | 48.8 | **47.9** | **49.3** |
| ↑ | - | - | 2.4 | - | 1.6 | - | 1.7 | - | 1.6 | - | 1.4 |
| ↑(%) | - | - | 5.3 | - | 3.4 | - | 3.6 | - | 3.4 | - | 2.9 |
| **Tasks** | **Dataset** | **20%** | | **40%** | | **60%** | | **80%** | | **100%** | |
| **(GPT-J)** | | +SFT | +MAPO | +SFT | +MAPO | +SFT | +MAPO | +SFT | +MAPO | +SFT | +MAPO |
| QA | AdverQA | 6.8 | 9.2 | 7.7 | 9.3 | 7.9 | 9.9 | **9.4** | **11.0** | 9.9 | 11.0 |
| | OpenQA | 17.3 | 19.0 | 17.3 | 19.1 | 18.2 | 19.5 | **20.3** | **21.0** | 19.8 | 21.1 |
| | CloseQA | 7.1 | 8.4 | 7.3 | 8.6 | 8.2 | 9.4 | **8.4** | **10.2** | 9.6 | 10.3 |
| Class | News | 2.4 | 4.6 | 2.8 | 4.8 | 3.9 | 5.3 | **5.5** | **6.3** | 5.5 | 6.4 |
| | Movie | 49.2 | 51.8 | 50.7 | 52.2 | 51.7 | 52.8 | **52.7** | **53.9** | 51.8 | 53.9 |
| | QASC | 50.3 | 53.8 | 52.8 | 54.4 | 53.6 | 55.2 | **56.3** | **56.8** | 55.2 | 56.9 |
| Gen | Topics | 19.5 | 21.3 | 20.0 | 21.6 | 20.8 | 22.2 | **21.6** | **23.4** | 21.8 | 23.6 |
| | Summary | 13.8 | 16.0 | 14.7 | 16.3 | 15.7 | 16.8 | **16.7** | **17.8** | 17.1 | 18.1 |
| | Explan | 8.5 | 11.2 | 9.9 | 11.4 | 10.1 | 11.9 | **10.7** | **12.9** | 11.8 | 13.2 |
| Average | - | 19.4 | 21.7 | 20.4 | 22.0 | 21.1 | 22.6 | **22.4** | **23.7** | 22.5 | 23.8 |
| ↑ | - | - | 2.3 | - | 1.6 | - | 1.5 | - | 1.3 | - | 1.3 |
| ↑(%) | - | - | 11.9 | - | 7.8 | - | 7.1 | - | 5.8 | - | 5.8 |
| **Tasks** | **Dataset** | **20%** | | **40%** | | **60%** | | **80%** | | **100%** | |
| **(LLaMA)** | | +SFT | +MAPO | +SFT | +MAPO | +SFT | +MAPO | +SFT | +MAPO | +SFT | +MAPO |
| QA | AdverQA | 19.6 | 22.2 | 22.0 | 23.2 | **23.2** | **25.1** | 24.5 | 25.4 | 24.4 | 26.0 |
| | OpenQA | 12.8 | 14.3 | 13.5 | 15.1 | **15.4** | **16.1** | 14.9 | 16.3 | 15.9 | 16.6 |
| | CloseQA | 11.5 | 13.0 | 12.2 | 13.8 | **13.9** | **14.8** | 14.5 | 15.2 | 14.8 | 15.4 |
| Class | News | 8.8 | 10.2 | 9.8 | 11.5 | **10.1** | **12.4** | 11.0 | 12.5 | 10.8 | 12.7 |
| | Movie | 77.5 | 79.8 | 79.0 | 80.2 | **81.3** | **82.5** | 80.6 | 82.7 | 81.5 | 83.1 |
| | QASC | 67.3 | 70.5 | 68.3 | 71.4 | **70.2** | **72.8** | 71.2 | 73.3 | 71.6 | 73.4 |
| Gen | Topics | 16.1 | 17.5 | 17.2 | 18.2 | **18.6** | **19.5** | 18.3 | 19.8 | 18.0 | 19.9 |
| | Summary | 8.2 | 10.1 | 9.4 | 11.3 | **10.7** | **12.2** | 10.5 | 12.4 | 11.6 | 12.8 |
| | Explan | 5.6 | 7.3 | 7.6 | 8.2 | **8.2** | **9.1** | 8.8 | 9.5 | 8.8 | 10.0 |
| Average | - | 25.3 | 27.2 | 26.6 | 28.1 | **28.0** | **29.4** | 28.3 | 29.7 | 28.6 | 30.0 |
| ↑ | - | - | 1.9 | - | 1.5 | - | 1.4 | - | 1.4 | - | 1.4 |
| ↑(%) | - | - | 7.5 | - | 5.6 | - | 5.0 | - | 4.9 | - | 4.9 |

Table 17: Performance of different proportion of warm-up dataset in various downstream tasks by three LLMs. Q: QA, C: classification, G:generation. SFT means using SFT-optimized prompts without RL.

| Task | Dataset | Best Performance | Best Epoch | Epoch | | | | | |
|------|---------|------------------|------------|---|---|----|----|----|----|
| - | - | - | - | 1 | 5 | 10 | 15 | 20 | 50 |
| QA | AdverQA | 18.3 | 18 | 14.8 | 14.3 | 15.4 | 18.2 | 18.3 | 18.2 |
| | OpenQA | 26.7 | 14 | 26.0 | 26.0 | 26.1 | 26.7 | 26.7 | 26.6 |
| | CloseQA | 7.8 | 19 | 6.8 | 7.0 | 7.0 | 7.5 | 7.8 | 7.8 |
| CLS | News | 95.5 | 20 | 93.2 | 93.7 | 94.3 | 95.3 | 95.5 | 95.7 |
| | Movie | 92.6 | 15 | 91.4 | 91.7 | 91.3 | 92.6 | 92.4 | 92.5 |
| | QASC | 99.9 | 7 | 99.6 | 99.8 | 99.9 | 99.9 | 99.9 | 99.9 |
| GEN | Topics | 34.8 | 19 | 30.3 | 31.5 | 33.8 | 34.2 | 34.8 | 34.8 |
| | Summary | 48.8 | 18 | 46.1 | 46.4 | 47.5 | 48.4 | 48.8 | 48.9 |
| | Explan | 6.8 | 15 | 5.9 | 6.3 | 6.4 | 6.8 | 6.5 | 6.7 |

| Task | Dataset | Best Performance | Best Epoch | Epoch | | | | | |
|------|---------|------------------|------------|---|---|----|----|----|----|
| - | - | - | - | 1 | 5 | 10 | 15 | 20 | 50 |
| QA | AdverQA | 9.4 | 20 | 4.4 | 5.3 | 6.7 | 8.8 | 9.4 | 9.6 |
| | OpenQA | 20.3 | 15 | 17.5 | 18.1 | 18.8 | 20.3 | 20.2 | 20.0 |
| | CloseQA | 8.4 | 20 | 7.0 | 7.3 | 7.8 | 8.2 | 8.4 | 8.6 |
| CLS | News | 5.5 | 20 | 1.4 | 2.6 | 3.1 | 4.8 | 5.5 | 5.9 |
| | Movie | 52.7 | 15 | 51.1 | 51.6 | 52.6 | 52.7 | 52.4 | 52.7 |
| | QASC | 56.3 | 19 | 54.3 | 54.6 | 54.5 | 55.4 | 56.3 | 56.3 |
| GEN | Topics | 21.6 | 18 | 17.9 | 20.2 | 20.7 | 21.3 | 21.6 | 21.6 |
| | Summary | 16.7 | 20 | 13.2 | 14.5 | 15.2 | 16.3 | 16.7 | 17.1 |
| | Explan | 10.7 | 19 | 8.8 | 9.2 | 9.9 | 9.6 | 10.7 | 11.0 |

| Task | Dataset | Best Performance | Best Epoch | Epoch | | | | | |
|------|---------|------------------|------------|---|---|----|----|----|----|
| - | - | - | - | 1 | 5 | 10 | 15 | 20 | 50 |
| QA | AdverQA | 23.2 | 20 | 5.8 | 9.3 | 13.3 | 18.7 | 23.2 | 29.5 |
| | OpenQA | 15.4 | 15 | 13.6 | 14.1 | 14.6 | 15.4 | 15.3 | 16.8 |
| | CloseQA | 13.9 | 20 | 10.8 | 11.3 | 12.9 | 13.5 | 13.9 | 14.5 |
| CLS | News | 10.1 | 13 | 3.4 | 5.7 | 9.3 | 10.1 | 10.1 | 10.1 |
| | Movie | 81.3 | 19 | 79.2 | 79.6 | 80.5 | 81.1 | 81.3 | 81.2 |
| | QASC | 70.2 | 20 | 62.8 | 63.7 | 65.9 | 69.2 | 70.2 | 71.9 |
| GEN | Topics | 18.6 | 20 | 15.5 | 16.1 | 17.7 | 18.4 | 18.6 | 19.3 |
| | Summary | 10.7 | 20 | 6.6 | 7.1 | 8.6 | 9.4 | 10.7 | 11.5 |
| | Explan | 8.2 | 14 | 7.3 | 7.5 | 7.5 | 8.2 | 8.2 | 8.3 |

Table 18: Performance of different number of epochs when training SFT. Best Performance means the best performance within 20 epochs. Best Epoch means the epoch corresponding to the best performance. We list the performance in the 1,5,10,15,20,50 epochs. We bold the performance metrics where a longer training epoch (epoch=50) results in a decline in performance.

| Task | Prompts |
|---|---|
| SST-2 | **Original**: The sentence is "khouri manages , with terrific flair, to keep the extremes of screwball farce and blood-curdling family intensity on one continuum. ".
**BLOOM**: The sentence is "khouri manages , with terrific flair, to keep the extremes of screwball farce and blood-curdling family intensity on one continuum. ".
**GPT-J**: The sentence is "With tremendous artistry, Khouri expertly maintains a fluid continuum between the uproarious comedy of screwball farce and the blood-curdling intensity of family dynamics.".
**LLaMA**: The sentence is "With remarkable skill, Khouri successfully maintains a seamless connection between the wild hilarity of screwball farce and the spine-chilling intensity of family dynamics.". |
| Yelp | **Original**: The sentence is "The food is good. Unfortunately the service is very hit or miss. The main issue seems to be with the kitchen, the waiters and waitresses are often very apologetic for the long waits and it's pretty obvious that some of them avoid the tables after taking the initial order to avoid hearing complaints.".
**BLOOM**: The sentence is "The quality of the food is commendable. However, the service provided is inconsistent. The primary concern appears to be with the kitchen, as the waitstaff frequently apologize for lengthy waits, and it's apparent that some of them intentionally avoid tables after taking initial orders to prevent complaints.".
**GPT-J**: The sentence is "The food is good. Unfortunately the service is very hit or miss. The main issue seems to be with the kitchen, the waiters and waitresses are often very apologetic for the long waits and it's pretty obvious that some of them avoid the tables after taking the initial order to avoid hearing complaints.".
**LLaMA**: The sentence is "The quality of the food is commendable. However, the service provided is inconsistent. The primary concern appears to be with the kitchen, as the waitstaff frequently apologize for lengthy waits, and it's apparent that some of them intentionally avoid tables after taking initial orders to prevent complaints.". |
| MR | **Original**: The sentence is "a respectable venture on its own terms , lacking the broader vision that has seen certain trek films . . . cross over to a more mainstream audience .".
**BLOOM**: The sentence is "An admirable endeavor in its own right, although it lacks the wider perspective that propelled certain Star Trek films to cross over into a more mainstream audience.".
**GPT-J**: The sentence is "a respectable venture on its own terms , lacking the broader vision that has seen certain trek films . . . cross over to a more mainstream audience .".
**LLaMA**: The sentence is "A praiseworthy pursuit on its own grounds, though it falls short of the broader vision that allowed certain Star Trek films to captivate a more mainstream audience.". |
| CR | **Original**: The sentence is "the fact that the 0 key is the space key for text input is a bit confusing , as many phones use the key instead".
**BLOOM**: The sentence is "It can be a little confusing that the space key for text input is the 0 key, considering that many phones utilize a different key for that purpose.".
**GPT-J**: The sentence is "the fact that the 0 key is the space key for text input is a bit confusing , as many phones use the key instead".
**LLaMA**: The sentence is "It might cause confusion that the space key for text input is assigned to the 0 key, whereas many phones utilize a different key for the same purpose.". |
| RTE | **Original**: The sentence 1 is "More than 150 dolphins, marine turtles and beaked whales have been washed up dead on beaches in Africa.". The sentence 2 is "Dead dolphins, turtles and whales have been found on African beaches.".
**BLOOM**: The sentence 1 is "African beaches have become a somber site with the washing up of over 150 deceased dolphins, marine turtles, and beaked whales.". The sentence 2 is "African beaches have become a distressing sight as dead dolphins, turtles, and whales have been uncovered.".
**GPT-J**: The sentence 1 is "The number of deceased dolphins, marine turtles, and beaked whales found on African beaches exceeds 150.". The sentence 2 is "There have been reports of deceased dolphins, turtles, and whales on African beaches.".
**LLaMA**: The sentence 1 is "More than 150 dolphins, marine turtles and beaked whales have been washed up dead on beaches in Africa.". The sentence 2 is "Dead dolphins, turtles and whales have been found on African beaches.". |
| QNLI | **Original**: The sentence 1 is "What came into force after the new constitution was herald?". The sentence 2 is "As of that day, the new constitution heralding the Second Republic came into force.".
**BLOOM**: The sentence 1 is "What started being in effect as a result of the heralding of the new constitution?". The sentence 2 is "The new constitution heralding the Second Republic started being in effect as of that day.".
**GPT-J**: The sentence 1 is "What commenced after the new constitution was heralded?". The sentence 2 is "The new constitution heralding the Second Republic came into force as of that day.".
**LLaMA**: The sentence 1 is "What became valid subsequent to the heralding of the new constitution?". The sentence 2 is " The new constitution heralding the Second Republic started being in force on that day.". |
| SNLI | **Original**: The sentence 1 is "A baby at the end of a slip and slide at a party". The sentence 2 is "The baby is wet".
**BLOOM**: The sentence 1 is "At a party, a baby reaches the conclusion of their slip and slide adventure.". The sentence 2 is "The little one is dripping wet.".
**GPT-J**: The sentence 1 is "A baby sliding to the end of a slip and slide at a celebration.". The sentence 2 is "The baby's clothes are moistened.".
**LLaMA**: The sentence 1 is "At the end of a festive occasion, a baby arrives at the final stretch of a slip and slide.". The sentence 2 is "The baby is entirely drenched.". |
| MNLI | **Original**: The sentence 1 is "Conversely, an increase in government saving adds to the supply of resources available for investment and may put downward pressure on interest rates.". The sentence 2 is "Interest rates should increase to increase saving.".
**BLOOM**: The sentence 1 is "Conversely, an increase in government saving adds to the supply of resources available for investment and may put downward pressure on interest rates.". The sentence 2 is "Interest rates should increase to increase saving.".
**GPT-J**: The sentence 1 is "Conversely, when the government saves more, it adds to the pool of resources for investment and may exert downward pressure on interest rates.". The sentence 2 is "The augmentation of interest rates is crucial for increasing saving.".
**LLaMA**: The sentence 1 is "On the flip side, an increase in government saving enhances the resources accessible for investment and may exert downward pressure on interest rates.". The sentence 2 is "The escalation of interest rates is essential for boosting savings.". |
| MRPC | **Original**: The sentence 1 is "Revenue in the first quarter of the year dropped 15 percent from the same period a year earlier.". The sentence 2 is "With the scandal hanging over Stewart's company, revenue the first quarter of the year dropped 15 percent from the same period a year earlier.".
**BLOOM**: The sentence 1 is "The first quarter of the year saw a 15 percent drop in revenue compared to the corresponding period a year earlier.". The sentence 2 is "The scandal's impact on Stewart's company led to a 15 percent drop in revenue during the first quarter, as compared to the same period in the prior year."
**GPT-J**: The sentence 1 is "There was a 15 percent decline in revenue during the first quarter of the year compared to the same period a year ago.". The sentence 2 is "Due to the scandal looming over Stewart's company, revenue in the first quarter of the year witnessed a 15 percent decline compared to the same period last year.".
**LLaMA**: The sentence 1 is "There was a 15 percent reduction in revenue during the first quarter of the year compared to the same period in the prior year.". The sentence 2 is "With the scandal overshadowing Stewart's company, revenue for the first quarter of the year dipped by 15 percent compared to the corresponding period in the previous year.". |

Table 19: Original prompts and MAPO-optimized prompts for different LLMs in more downstream tasks. Each prompt is start with the corresponding instruction as shown in Table 13. We omit instructions in the following Table due to space limits.