# OpenReview forum: "MAPO: Boosting Large Language Model Performance with Model-Adaptive Prompt Optimization"
_EMNLP/2023/Conference — EMNLP 2023 Findings_

### Official Review · Reviewer_DfN7 · 2023-08-04

**Typos Grammar Style And Presentation Improvements:** In Figure 3, better prompt should be …
**Soundness:** 3

**Excitement:**

3: Ambivalent: It has merits (e.g., it reports state-of-the-art results, the idea is nice), but there are key weaknesses (e.g., it describes incremental work), and it can significantly benefit from another round of revision. However, I won't object to accepting it if my co-reviewers champion it.

**Paper Topic And Main Contributions:**

The paper presents a compelling argument that different LLMs perform better with specific prompts. While previous prompt engineering research primarily focuses on optimizing prompts for specific tasks, this paper introduces the idea that prompts should be adapted to specific LLMs. The authors propose a model-adaptive prompt optimizer (MAPO) that tailors prompts for each LLM in downstream tasks. The experiments demonstrate that the refined prompts can lead to significant improvements.

**Questions For The Authors:**

See Reasons To Reject.

I don't see how the idea of adapting prompts to different models relates to the work on optimizing prompts for multi-modal large models (line 61-64).

**Reasons To Accept:**

1. The observation that different LLMs benefit from distinct prompts is interesting.
2. The paper demonstrates that the proposed model-adaptive prompt optimizer significantly improves the performance on downstream tasks.

**Reasons To Reject:**

1. While the idea of adapting prompts to different LLMs is interesting, the proposed method does not align entirely with my initial expectation when reading the introduction. The method does not seem to be explicitly designed for model-specific adaptation. Both the supervised fine-tuning and reinforcement learning methods are optimizing the prompts for a target LLM, which is also the case for most of the previous prompt tuning methods. I was expecting to see something like MAML (model-agnostic meta learning).
2. It is unclear how the similarity is computed for searching the optimal prompt (section 3.1). The paper mentions “the match degree between the output of each candidate prompt and that of the original prompt”. These words to mean the outputs of two prompts from the same, target LLM. If this is the case, why is it helpful to optimize the model to produce similar outputs to the original prompts? It would make more sense if the metric is defined as the similarity between the output of the candidate prompt by the target LLM (e.g., LLaMA) and the output of the original prompt by a stronger LLM (e.g., GPT-3.5), thus the model receives meaningful supervision. I hope this can be clarified.
3. The proposed method fine-tunes a LLM to provide better prompts for a LLM on a downstream task, leveraging supervised learning and reinforcement learning. A more straightforward approach would be directly fine-tuning the model itself on the downstream task. From my perspective, I'm not entirely convinced of the advantages of the proposed method compared to this simpler approach.

**Reproducibility:**

4: Could mostly reproduce the results, but there may be some variation because of sample variance or minor variations in their interpretation of the protocol or method.

**Reviewer Confidence:**

3: Pretty sure, but there's a chance I missed something. Although I have a good feel for this area in general, I did not carefully check the paper's details, e.g., the math, experimental design, or novelty.

---

> ### Author Rebuttal · Authors · 2023-08-29
>
> Thank you very much for your valuable suggestions and comments on our manuscript. Below are our responses to each of the points of criticism, questions, and typos you've raised.
>
>     R1: While the idea of adapting prompts to different LLMs is interesting, the proposed method does not align entirely with my initial expectation when reading the introduction. The method does not seem to be explicitly designed for model-specific adaptation. Both the supervised fine-tuning and reinforcement learning methods are optimizing the prompts for a target LLM, which is also the case for most of the previous prompt tuning methods. I was expecting to see something like MAML (model-agnostic meta learning).
>
> 1. Thank you for your insightful feedback. We understand that our approach, MAPO, may not have aligned with your initial expectations based on the introduction, particularly with regards to model-specific adaptation. Allow us to clarify our position.
> 2. The overarching methods of SFT and RL are indeed used to optimize prompts for a target LLM. However, the innovation of MAPO lies in its intent to finely optimize these prompts specifically for each LLM. **Unlike MAML, which seeks to find a general initialization for various tasks or models, MAPO focuses on the special characteristics of each model to improve its performance, thereby enhancing its performance on a given task. We believe that different LLMs likely require different prompts to achieve optimal performance for the same downstream task.**
> 3. Our experiments consistently demonstrate that **MAPO effectively improves upon the prompt optimization achieved solely through SFT, irrespective of how effective SFT might be on its own**. This shows that MAPO is better at customizing the optimization for each specific model compared to older methods.
>
> We hope this clarifies the distinctive aspects of MAPO and addresses your concerns.
>
>
>
>     R2: It is unclear how the similarity is computed for searching the optimal prompt (section 3.1). The paper mentions “the match degree between the output of each candidate prompt and that of the original prompt”. These words to mean the outputs of two prompts from the same, target LLM. If this is the case, why is it helpful to optimize the model to produce similar outputs to the original prompts? It would make more sense if the metric is defined as the similarity between the output of the candidate prompt by the target LLM (e.g., LLaMA) and the output of the original prompt by a stronger LLM (e.g., GPT-3.5), thus the model receives meaningful supervision. I hope this can be clarified.
>
>     i) why is it helpful to optimize the model to produce similar outputs to the original prompts?
>
> 1. Thank you for pointing out the error, and we appreciate the opportunity to clarify this aspect of our methodology. In our warm-up dataset establishment process, **we aim to find the most effective candidate prompts that will produce outputs closely aligned with the ground-truth outputs, not with the outputs generated by the original prompt as initially stated.**
>
> 2. Specifically, we input both the original prompt $P$ and each candidate prompt into a target LLM for inference. We then obtain corresponding outputs and measure the match degree between each output and the ground-truth output using specified evaluation metrics. For example, if the match degree between the output generated by a candidate prompt 1 and the ground-truth output is 0.8, and the match degree between the output generated by the original prompt and the ground-truth output is 0.6, we consider candidate prompt 1 to be superior to the original prompt.
>
> 3. In the construction of our prompt pair ranking, we would place candidate prompt 1 to the right of the original prompt. If among all the candidate prompts, candidate prompt $k$ produces an output with the highest match degree of 0.9 with the ground-truth output and this match degree also surpasses that of the original prompt, we regard this pair of original prompt $P$ and candidate prompt $k$ as a prompt pair, referred to as $\{P, P_o\}$, where $P_o$ is candidate prompt $k$.
>
> We hope this clarifies the methodology we use for determining optimal prompts.
>
>
>     ii) It would make more sense if the metric is defined as the similarity between the output of the candidate prompt by the target LLM (e.g., LLaMA) and the output of the original prompt by a stronger LLM (e.g., GPT-3.5), thus the model receives meaningful supervision.
>
>
> Thank you for your comment. However, our objective and methodology differ from what you suggest. Let's clarify the key differences:
>
> 1. Our Objective: Our goal is to find the optimal prompt that can generate an output as close as possible to the ground-truth output for a given target LLM. We can't directly reverse-engineer the prompt from the ground-truth output, and it's likely that different LLMs would need different optimal prompts to generate this ground-truth output. Therefore, we compute the "match degree" between the outputs generated by both the candidate and the original prompts against this ground-truth output. **The objective is for the original prompt to generate an output increasingly closer to the ground-truth through optimization.**
>
> 2. Your Suggestion: You propose using the output generated by a stronger LLM as a basis for comparison with the outputs generated by the candidate and original prompts of the target LLM. While a stronger LLM may produce better outputs, these are not guaranteed to be optimal. When a ground-truth output is not available, using a stronger LLM's output as a stand-in make sense.
>
> **The core difference between our method and your suggestion lies in the definition of the "gold standard" against which the "match degree" is calculated.** We use the ground-truth output as this standard, aiming for prompts that can generate outputs as close as possible to this truth. Your suggestion would use the output from a stronger, yet different, LLM as this standard. **When a ground-truth output is not available, using a stronger LLM's output as a stand-in make sense.**
>
> We hope this clears up any misunderstandings.
>
>
>
>
>     R3: The proposed method fine-tunes a LLM to provide better prompts for a LLM on a downstream task, leveraging supervised learning and reinforcement learning. A more straightforward approach would be directly fine-tuning the model itself on the downstream task. From my perspective, I'm not entirely convinced of the advantages of the proposed method compared to this simpler approach.
>
>
> Thank you for your question. Our research aims to demonstrate the specific advantages of MAPO over simply applying SFT (that is directly fine-tuning the model itself on the downstream task).
>
> 1. Firstly, we report the results of +SFT in Tables 3, 4, and 5 to compare the results of +SFT and +RL (MAPO). **We can observe that the performance improvements gained solely from using SFT are less than those achieved by our proposed MAPO method**, both on similar tasks (Table 3) and general NLP tasks (Tables 4 and 5). This clearly indicates the effectiveness of MAPO in optimizing model-adaptive prompts. The blank section indicates that the content is the same as the cell above. Due to the limitations of Markdown syntax, merging cells is not possible.
>
>
>
> Table 3: The few-shot performance of SFT, MAPO with SOTA prompt optimizing baselines in downstream tasks. F: Finetuning, C: Continous prompt, D: Discrete prompt.
> |       |              | SST-2    | YelpP.   | MR       | CR       | RTE      | QNLI     | SNLI     | MNLI     | MRPC     |
> |-------|--------------|----------|----------|----------|----------|----------|----------|----------|----------|----------|
> | F     | Finetuning   | 80.6     | 88.7     | 67.4     | 73.3     | 58.6     | 60.2     | 54.6     | 47.8     | 77.4     |
> | C     | Softprompt   | 73.8     | 88.6     | 74.1     | 75.9     | 54.7     | 49.7     | 36.1     | 33.2     | 51.6     |
> |       | Black-Box    | 89.1     | 93.2     | 86.6     | 87.4     | 52.6     | 48.8     | 46.6     | 42.9     | 61.6     |
> |       | Autoprompt   | 75.0     | 79.8     | 62.0     | 57.5     | -        | -        | -        | -        | -        |
> | D     | Manual       | 82.8     | 83.0     | 80.9     | 79.6     | 51.6     | 50.8     | 31.1     | 51.7     | 67.4     |
> |       | In-Context   | 85.9     | 89.6     | 80.6     | 85.5     | 60.4     | 53.8     | 47.1     | 53.4     | 45.8     |
> |       | Instructions | 89.0     | 84.4     | 85.2     | 80.8     | -        | -        | -        | -        | -        |
> |       | GrIPS        | 87.1     | 88.2     | 86.1     | 80.0     | -        | -        | -        | -        | -        |
> |       | RLprompt     | 92.5     | 95.1     | 87.1     | 89.5     | -        | -        | -        | -        | -        |
> |       | TEMPERA      | 91.9     | 92.6     | 88.0     | 91.1     | 60.3     | 57.4     | 56.4     | 45.2     | 74.0     |
> |       | AMA          | 95.7     | -        | -        | -        | 75.1     | -        | -        | -        | -        |
> | **D** | **SFT**      | **94.9** | **92.0** | **88.5** | **87.6** | **74.3** | **62.5** | **58.8** | **54.6** | **78.5** |
> | **D** | **MAPO**     | **96.1** | **93.5** | **90.2** | **88.9** | **75.3** | **63.1** | **60.0** | **55.7** | **79.3** |
>
>
> Table 4: Zero-shot domain transfer performance based on BLOOM and GPT-J with original, SFT-optimized and MAPO-Optimized prompts. CLS: Classification, M: MAPO. The (↑(%) ) and ↑(%) represent the increase degree of MAPO-optimized prompts compared with original prompts in each dataset and task, respectively (The same below).
> | Task   | Dataset    | BLOOM | +SFT | (↑(%)) | **↑(%)** | +MAPO | (↑(%) | **↑(%)** | GPT-J | +SFT | (↑(%)) | **↑(%)** | +MAPO | (↑(%)) | **↑(%)** |
> |--------|------------|-------|------|--------|----------|-------|-------|----------|-------|------|--------|----------|-------|--------|----------|
> | Coref. | xwinograd  | 60.1  | 60.2 | 0.2    | **0.2**  | 60.6  | 0.9   | **0.9**  | -     | -    | -      | **-**    | -     | -      | **-**    |
> | NLU    | BoolQ      | 67.9  | 68.0 | 0.1    | **0.4**  | 68.2  | 0.4   | **0.9**  | 67.2  | 67.4 | 0.3    | **0.2**  | 67.9  | 1.0    | **0.5**  |
> |        | CB         | 77.6  | 77.8 | 0.3    |          | 78.1  | 0.6   |          | 83.9  | 84.1 | 0.2    |          | 84.2  | 0.4    |          |
> |        | COPA       | 74.0  | 74.3 | 0.4    |          | 75.0  | 1.4   |          | 84.0  | 84.2 | 0.2    |          | 84.2  | 0.2    |          |
> |        | MultiRC    | 59.7  | 60.3 | 1.0    |          | 60.4  | 1.2   |          | 63.8  | 63.9 | 0.2    |          | 64.1  | 0.5    |          |
> |        | ReCoRD     | 69.8  | 70.1 | 0.4    |          | 70.2  | 0.6   |          | 74.4  | 74.5 | 0.1    |          | 74.7  | 0.4    |          |
> |        | WiC        | 61.4  | 61.6 | 0.3    |          | 62.0  | 1.0   |          | 61.0  | 61.1 | 0.2    |          | 61.3  | 0.5    |          |
> |        | WSC        | 64.4  | 64.7 | 0.5    |          | 65.1  | 1.1   |          | 77.9  | 78.0 | 0.1    |          | 78.1  | 0.3    |          |
> | NLI    | ANLIR1     | 31.5  | 31.7 | 0.6    | **0.6**  | 32.1  | 1.9   | **1.3**  | 37.8  | 38.0 | 0.5    | **0.3**  | 38.2  | 1.1    | **0.7**  |
> |        | ANLIR2     | 35.1  | 35.2 | 0.3    |          | 35.4  | 0.9   |          | 37.9  | 38.0 | 0.3    |          | 38.3  | 1.1    |          |
> |        | ANLIR3     | 37.1  | 37.5 | 1.1    |          | 37.8  | 1.9   |          | 40.9  | 41.0 | 0.2    |          | 41.1  | 0.5    |          |
> |        | StoryCloze | 79.0  | 79.2 | 0.3    |          | 79.5  | 0.6   |          | 87.8  | 87.9 | 0.1    |          | 87.9  | 0.1    |          |
> | CLS    | Amazon     | 65.2  | 66.4 | 1.8    | **1.4**  | 67.7  | 3.8   | **3.3**  | 68.2  | 68.7 | 0.7    | **0.6**  | 69.4  | 1.8    | **1.6**  |
> |        | DBPedia    | 70.5  | 71.2 | 1.0    |          | 72.5  | 2.8   |          | 83.9  | 84.2 | 0.4    |          | 85.1  | 1.4    |          |
> | QA     | DROP       | 67.9  | 68.2 | 0.4    | **1.9**  | 69.9  | 2.9   | **5.6**  | 51.6  | 51.9 | 0.6    | **1.3**  | 52.8  | 2.3    | **3.2**  |
> |        | NQ         | 15.1  | 15.4 | 2.0    |          | 16.1  | 6.6   |          | 19.6  | 20.1 | 2.6    |          | 20.8  | 6.1    |          |
> |        | RealTimeQA | 29.0  | 30.2 | 4.1    |          | 31.5  | 8.6   |          | 36.0  | 36.5 | 1.4    |          | 37.2  | 3.3    |          |
> |        | WebQs      | 34.8  | 35.1 | 0.9    |          | 36.3  | 4.3   |          | 44.1  | 44.3 | 0.5    |          | 44.6  | 1.1    |          |
>
> Table 5: Zero-shot domain transfer performance based on LLaMA with original, SFT-optimized and MAPO-Optimized prompts. RS: Commonsense Reasoning.
> | Task | Dataset    | LLaMA | +SFT | (↑(%)) | **↑(%)** | +MAPO | (↑(%)) | **↑(%)** |
> |------|------------|-------|------|--------|----------|-------|--------|----------|
> | RS   | BoolQ      | 76.5  | 76.6 | 0.1    | **0.1**  | 76.7  | 0.3    | **0.5**  |
> |      | PIQA       | 79.8  | 79.9 | 0.1    |          | 80.0  | 0.3    |          |
> |      | SIQA       | 48.9  | 48.9 | 0.0    |          | 49.0  | 0.2    |          |
> |      | HellaSwag  | 76.1  | 76.2 | 0.1    |          | 76.5  | 0.5    |          |
> |      | WinoGrande | 70.1  | 70.2 | 0.1    |          | 70.5  | 0.6    |          |
> |      | ARC-e      | 72.8  | 72.9 | 0.1    |          | 73.2  | 0.6    |          |
> |      | ARC-c      | 47.6  | 47.6 | 0.0    |          | 47.8  | 0.4    |          |
> |      | OBQA       | 57.2  | 57.4 | 0.3    |          | 57.9  | 1.2    |          |
> | QA   | NQ         | 16.8  | 17.2 | 2.4    | **1.4**  | 18.1  | 7.7    | **4.7**  |
> |      | RACE       | 50.0  | 50.2 | 0.4    |          | 50.8  | 1.6    |          |
>
>
>
> 2. Secondly, we use various proportions of the warm-up dataset, specifically 20%, 40%, 60%, 80%, and 100%, to progressively increase the SFT training data and then introduce RL to it. For each subset, such as the 20% data subset, we perform both SFT and SFT+RL (that is MAPO), with the SFT+RL approach building upon the same SFT subset. Performance metrics for each subset on the target tasks are carefully recorded. We report the results in the following tables. **Our findings consistently show that RL adds value to the performance beyond what is achieved by SFT alone across all proportions of the dataset. This affirms the effectiveness of RL irrespective of the SFT dataset size.** However, as the proportion of the warm-up dataset increases, the margin of improvement from adding RL begins to decline. The blank section indicates that the content is the same as the cell above. Due to the limitations of Markdown syntax, merging cells is not possible.
>
>
>
> Table12: Performance of different proportion of warm-up dataset in various downstream tasks by three LLMs. Q: QA, C: classification, G:generation. +SFT means using SFT-optimized prompts without RL.
>
> #### LLaMA
> | Tasks   | Dataset |  20% |         |  40% |         |  **60%** |           |  80% |       | 100% |       |
> |---------|:-------:|:----:|:-------:|:----:|:-------:|:--------:|:---------:|:----:|:-----:|:----:|:-----:|
> |         |         | +SFT |  +MAPO  | +SFT |  +MAPO  | **+SFT** | **+MAPO** | +SFT | +MAPO | +SFT | +MAPO |
> | QA      | AdverQA | 19.6 |   22.2  |  22  |   23.2  | **23.2** |  **25.1** | 24.5 |  25.4 | 24.4 |   26  |
> |         |  OpenQA | 12.8 |   14.3  | 13.5 |   15.1  | **15.4** |  **16.1** | 14.9 |  16.3 | 15.9 |  16.6 |
> |         | CloseQA | 11.5 |    13   | 12.2 |   13.8  | **13.9** |  **14.8** | 14.5 |  15.2 | 14.8 |  15.4 |
> | Class   |   News  |  8.8 |   10.2  |  9.8 |   11.5  | **10.1** |  **12.4** |  11  |  12.5 | 10.8 |  12.7 |
> |         |  Movie  | 77.5 |   79.8  |  79  |   80.2  | **81.3** |  **82.5** | 80.6 |  82.7 | 81.5 |  83.1 |
> |         |   QASC  | 67.3 |   70.5  | 68.3 |   71.4  | **70.2** |  **72.8** | 71.2 |  73.3 | 71.6 |  73.4 |
> | Gen     |  Topics | 16.1 |   17.5  | 17.2 |   18.2  | **18.6** |  **19.5** | 18.3 |  19.8 |  18  |  19.9 |
> |         | Summary |  8.2 |   10.1  |  9.4 |   11.3  | **10.7** |  **12.2** | 10.5 |  12.4 | 11.6 |  12.8 |
> |         |  Explan |  5.6 |   7.3   |  7.6 |   8.2   |  **8.2** |  **9.1**  |  8.8 |  9.5  |  8.8 |   10  |
> | Average |    -    | 25.3 |   27.2  | 26.6 |   28.1  |  **28**  |  **29.4** | 28.3 |  29.7 | 28.6 |   30  |
> | ↑       |    -    |   -  | **1.9** |   -  | **1.5** |     -    |  **1.4**  |   -  |  1.4  |   -  |  1.4  |
> | ↑(%)    |    -    |   -  | **7.5** |   -  | **5.6** |     -    |   **5**   |   -  |  4.9  |   -  |  4.9  |
>
> #### BLOOM
> | Tasks   | Dataset |  20% |         |  40% |         |  60% |         |  80% |         | **100%** |          |
> |---------|:-------:|:----:|:-------:|:----:|:-------:|:----:|:-------:|:----:|:-------:|:--------:|:--------:|
> |         |         |  +SFT |   +MAPO  |  +SFT |   +MAPO  |  +SFT |   +MAPO  |  +SFT |   +MAPO  |  **+SFT** | **+MAPO** |
> | QA      | AdverQA | 14.6 |   17.8  | 16.9 |   18.6  | 17.5 |   18.9  | 17.7 |   19.2  | **18.3** | **19.5** |
> |         |  OpenQA | 22.1 |   25.4  | 24.3 |   26.1  | 24.9 |   26.3  | 25.1 |   26.5  | **26.7** | **27.2** |
> |         | CloseQA |  6.6 |    8    |  7.7 |   8.6   |  7.3 |   8.8   |  7.8 |   9.1   |  **7.8** |  **9.4** |
> | Class   |   News  | 93.6 |   96.8  | 95.8 |   97.4  | 95.5 |   97.6  | 96.2 |    98   | **95.5** | **98.7** |
> |         |  Movie  | 87.1 |   89.8  | 88.3 |   90.9  | 88.9 |   91.3  | 90.7 |   92.5  | **92.6** | **93.3** |
> |         |   QASC  |  96  |    98   | 97.6 |   98.6  | 97.1 |   98.7  | 97.4 |   98.8  | **99.9** | **99.9** |
> | Gen     |  Topics | 32.7 |    35   | 33.6 |   35.7  | 34.6 |    36   | 34.7 |   36.4  | **34.8** | **36.2** |
> |         | Summary | 45.8 |   47.9  | 47.1 |   48.9  | 47.3 |   49.3  | 48.5 |   49.7  | **48.8** | **50.2** |
> |         |  Explan |  6.2 |   7.5   |  7.3 |   8.1   |  7.1 |   8.3   |  6.3 |   8.6   |  **6.8** |  **8.9** |
> | Average |    -    |  45  |   47.4  | 46.5 |   48.1  | 46.7 |   48.4  | 47.2 |   48.8  | **47.9** | **49.3** |
> | ↑       |    -    |   -  | **2.4** |   -  | **1.6** |   -  | **1.7** |   -  | **1.6** |     -    |  **1.4** |
> | ↑(%)    |    -    |   -  | **5.3** |   -  | **3.4** |   -  | **3.6** |   -  | **3.4** |     -    |  **2.9** |
>
>
> #### GPT-J
> | Tasks   | Dataset |  20% |          |  40% |         |  60% |         |  **80%** |          | 100% |         |
> |---------|:-------:|:----:|:--------:|:----:|:-------:|:----:|:-------:|:--------:|:--------:|:----:|:-------:|
> |         |         |  +SFT |   +MAPO   |  +SFT |   +MAPO  |  +SFT |   +MAPO  |  **+SFT** | **+MAPO** |  +SFT |   +MAPO  |
> | QA      | AdverQA |  6.8 |    9.2   |  7.7 |   9.3   |  7.9 |   9.9   |  **9.4** |  **11**  |  9.9 |    11   |
> |         |  OpenQA | 17.3 |    19    | 17.3 |   19.1  | 18.2 |   19.5  | **20.3** |  **21**  | 19.8 |   21.1  |
> |         | CloseQA |  7.1 |    8.4   |  7.3 |   8.6   |  8.2 |   9.4   |  **8.4** | **10.2** |  9.6 |   10.3  |
> | Class   |   News  |  2.4 |    4.6   |  2.8 |   4.8   |  3.9 |   5.3   |  **5.5** |  **6.3** |  5.5 |   6.4   |
> |         |  Movie  | 49.2 |   51.8   | 50.7 |   52.2  | 51.7 |   52.8  | **52.7** | **53.9** | 51.8 |   53.9  |
> |         |   QASC  | 50.3 |   53.8   | 52.8 |   54.4  | 53.6 |   55.2  | **56.3** | **56.8** | 55.2 |   56.9  |
> | Gen     |  Topics | 19.5 |   21.3   |  20  |   21.6  | 20.8 |   22.2  | **21.6** | **23.4** | 21.8 |   23.6  |
> |         | Summary | 13.8 |    16    | 14.7 |   16.3  | 15.7 |   16.8  | **16.7** | **17.8** | 17.1 |   18.1  |
> |         |  Explan |  8.5 |   11.2   |  9.9 |   11.4  | 10.1 |   11.9  | **10.7** | **12.9** | 11.8 |   13.2  |
> | Average |    -    | 19.4 |   21.7   | 20.4 |    22   | 21.1 |   22.6  | **22.4** | **23.7** | 22.5 |   23.8  |
> | ↑       |    -    |   -  |  **2.3** |   -  | **1.6** |   -  | **1.5** |     -    |  **1.3** |   -  | **1.3** |
> | ↑(%)    |    -    |   -  | **11.9** |   -  | **7.8** |   -  | **7.1** |     -    |  **5.8** |   -  | **5.8** |
>
>
>
> 3. Thirdly, we have tried different number of epochs to see if extended training time consistently improves SFT performance before submitting the manuscript. **Extending the training time does not consistently lead to performance improvements for SFT. In some instances, the performance even declines.** It is important to note that we save the best-performing models in real-time during training, as the peak performance does not necessarily occur at the final epoch. To further validate that extended training time is not the reason SFT underperforms compared to MAPO, we also conduct additional experiments that increasing the number of epochs (epoch=50) for SFT in three LLMs (LLaMA, BLOOM, GPT-J). **The findings indicate that merely increasing the number of epochs does not significantly improve the SFT performance in all tasks.**  The results are shown in the following tables.
> Based on the comparative results between SFT and MAPO across different proportions of datasets, theoretically, MAPO is capable of further improving prompt optimization on top of whatever performance SFT achieves.
>
> Table13: Performance of different number of epochs when training SFT. Best Performance means the best performance within 20 epochs. Best Epoch means the epoch corresponding to the best performance。 We list the performance in the 1,5,10, 15,20,50 epochs. We bold the performance metrics where a longer training epoch (epoch=50) results in a decline in performance.
>
> #### BLOOM
> | Task | Dataset | Best Performance | Best Epoch | Epoch |      |      |      |      |          |
> |------|---------|------------------|------------|:-----:|:----:|:----:|:----:|:----:|:--------:|
> | -    | -       | -                | -          | 1     | 5    | 10   | 15   | 20   | 50       |
> | QA   | AdverQA | 18.3             | 18         | 14.8  | 14.3 | 15.4 | 18.2 | 18.3 | **18.2** |
> |      | OpenQA  | 26.7             | 14         | 26    | 26   | 26.1 | 26.7 | 26.7 | **26.6** |
> |      | CloseQA | 7.8              | 19         | 6.8   | 7    | 7    | 7.5  | 7.8  | **7.8**  |
> | CLS  | News    | 95.5             | 20         | 93.2  | 93.7 | 94.3 | 95.3 | 95.5 | 95.7     |
> |      | Movie   | 92.6             | 15         | 91.4  | 91.7 | 91.3 | 92.6 | 92.4 | 92.5     |
> |      | QASC    | 99.9             | 7          | 99.6  | 99.8 | 99.9 | 99.9 | 99.9 | **99.9** |
> | GEN  | Topics  | 34.8             | 19         | 30.3  | 31.5 | 33.8 | 34.2 | 34.8 | **34.8** |
> |      | Summary | 48.8             | 18         | 46.1  | 46.4 | 47.5 | 48.4 | 48.8 | 48.9     |
> |      | Explan  | 6.8              | 15         | 5.9   | 6.3  | 6.4  | 6.8  | 6.5  | 6.7      |
>
>
> #### GPT-J
> | Task | Dataset | Best Performance | Best Epoch | Epoch |      |      |      |      |          |
> |------|---------|------------------|------------|:-----:|:----:|:----:|:----:|:----:|:--------:|
> | -    | -       | -                | -          | 1     | 5    | 10   | 15   | 20   | 50       |
> | QA   | AdverQA | 9.4              | 20         | 4.4   | 5.3  | 6.7  | 8.8  | 9.4  | 9.6      |
> |      | OpenQA  | 20.3             | 15         | 17.5  | 18.1 | 18.8 | 20.3 | 20.2 | **20**   |
> |      | CloseQA | 8.4              | 20         | 7     | 7.3  | 7.8  | 8.2  | 8.4  | 8.6      |
> | CLS  | News    | 5.5              | 20         | 1.4   | 2.6  | 3.1  | 4.8  | 5.5  | 5.9      |
> |      | Movie   | 52.7             | 15         | 51.1  | 51.6 | 52.6 | 52.7 | 52.4 | 52.7     |
> |      | QASC    | 56.3             | 19         | 54.3  | 54.6 | 54.5 | 55.4 | 56.3 | **56.3** |
> | GEN  | Topics  | 21.6             | 18         | 17.9  | 20.2 | 20.7 | 21.3 | 21.6 | **21.6** |
> |      | Summary | 16.7             | 20         | 13.2  | 14.5 | 15.2 | 16.3 | 16.7 | 17.1     |
> |      | Explan  | 10.7             | 19         | 8.8   | 9.2  | 9.9  | 9.6  | 10.7 | 11       |
>
>
> #### LLaMA
> | Task | Dataset | Best Performance | Best Epoch | Epoch |      |      |      |      |          |
> |------|---------|------------------|------------|:-----:|:----:|:----:|:----:|:----:|:--------:|
> | -    | -       | -                | -          | 1     | 5    | 10   | 15   | 20   | 50       |
> | QA   | AdverQA | 23.2             | 20         | 5.8   | 9.3  | 13.3 | 18.7 | 23.2 | 29.5     |
> |      | OpenQA  | 15.4             | 15         | 13.6  | 14.1 | 14.6 | 15.4 | 15.3 | 16.8     |
> |      | CloseQA | 13.9             | 20         | 10.8  | 11.3 | 12.9 | 13.5 | 13.9 | 14.5     |
> | CLS  | News    | 10.1             | 13         | 3.4   | 5.7  | 9.3  | 10.1 | 10.1 | **10.1** |
> |      | Movie   | 81.3             | 19         | 79.2  | 79.6 | 80.5 | 81.1 | 81.3 | **81.2** |
> |      | QASC    | 70.2             | 20         | 62.8  | 63.7 | 65.9 | 69.2 | 70.2 | 71.9     |
> | GEN  | Topics  | 18.6             | 20         | 15.5  | 16.1 | 17.7 | 18.4 | 18.6 | 19.3     |
> |      | Summary | 10.7             | 20         | 6.6   | 7.1  | 8.6  | 9.4  | 10.7 | 11.5     |
> |      | Explan  | 8.2              | 14         | 7.3   | 7.5  | 7.5  | 8.2  | 8.2  | 8.3      |
>
>
>
> 4. We also investigate the separate effect of PPO and RRHF in Table 9 in Appendix, which demonstrate the important roles played by both PPO and RRHF in enhancing the performance of MAPO. We put the detailed results as follows. We also add the results of SFT-optimized prompts for comparison.
>
> Table 9: The separate effect of PPO and RRHF, which demonstrate the important roles played by both PPO and RRHF in enhancing the performance of MAPO.
> | Tasks   | Dataset | +SFT |       |       | +PPO |         |         | +RRHF |         |         | +MAPO |         |         |
> |---------|---------|:-------------:|:-----:|:-----:|:-------------:|:-------:|:-------:|:--------------:|:-------:|:-------:|:--------------:|:-------:|:-------:|
> |         |         | BLOOM         | GPT-J | LLaMA | BLOOM         |  GPT-J  |  LLaMA  | BLOOM          |  GPT-J  |  LLaMA  | BLOOM          | GPT-J   | LLaMA   |
> | QA      | AdverQA | 18.3          | 9.4   | 23.2  | 18.8          | 9.6     | 24.3    | 19.2           | 9.9     | 24.8    | 19.5           | 11.0    | 25.1    |
> |         | OpenQA  | 26.7          | 20.3  | 15.4  | 26.9          | 20.7    | 16.0    | 27.0           | 20.9    | 15.8    | 27.2           | 21.0    | 16.1    |
> |         | CloseQA | 7.8           | 8.4   | 13.9  | 8.3           | 8.8     | 14.4    | 9.0            | 9.8     | 14.6    | 9.4            | 10.2    | 14.8    |
> | Class   | News    | 95.5          | 5.5   | 10.1  | 96.9          | 6.1     | 12.1    | 98.3           | 6.2     | 12.2    | 98.7           | 6.3     | 12.4    |
> |         | Movie   | 92.6          | 52.7  | 81.3  | 93.1          | 53.1    | 82.1    | 93.0           | 53.6    | 82.2    | 93.3           | 53.9    | 82.5    |
> |         | QASC    | 99.9          | 56.3  | 70.2  | 99.9          | 56.7    | 72.4    | 99.9           | 56.5    | 71.2    | 99.9           | 56.8    | 72.8    |
> | Gen     | Topics  | 34.8          | 21.6  | 18.6  | 35.9          | 23.1    | 18.9    | 35.2           | 23.2    | 19.1    | 36.2           | 23.4    | 19.5    |
> |         | Summary | 48.8          | 16.7  | 10.7  | 50.1          | 17.2    | 11.7    | 49.4           | 17.7    | 12.0    | 50.2           | 17.8    | 12.2    |
> |         | Explan  | 6.8           | 10.7  | 8.2   | 8.3           | 11.5    | 8.4     | 7.2            | 12.4    | 8.7     | 8.9            | 12.9    | 9.1     |
> | Average | -       | 47.9          | 22.4  | 28.0  | 48.7          | 23.0    | 28.9    | 48.7           | 23.4    | 29.0    | 49.3           | 23.7    | 29.4    |
> | ↑       | -       | -             | -     | -     | **0.8**       | **0.6** | **0.9** | **0.8**        | **1.7** | **1.0** | **1.4**        | **1.3** | **1.4** |
> | ↑(%)    | -       | -             | -     | -     | **1.7**       | **2.7** | **3.2** | **1.7**        | **4.5** | **3.6** | **2.9**        | **5.8** | **5.0** |
>
>
> In essence, while direct fine-tuning of the model for downstream tasks is a straightforward and established approach, our MAPO technique offers a systematic and consistently improved prompt optimization, making it a valuable complement rather than a replacement. We will add these analysis in the revised version of manuscript.
>
>
>
>     Q:  I don't see how the idea of adapting prompts to different models relates to the work on optimizing prompts for multi-modal large models (line 61-64).
>
> 1. Thank you for pointing this out. **The idea of adapting prompts to different LLMs is intrinsically tied to the concept of "model-preferred prompts" or "model-specific prompts,"** which suggests that each LLM, given its architecture and training data, might have a unique or preferred way it wants to receive input to perform optimally. **The work by Hao et al. (2022) on multi-modal LLMs underscores this concept,** emphasizing that there's a need for a systematic method to automatically align user intentions with the specific prompt preferences of each model.
> 2. Inspired by the concept of "model-preferred prompts" introduced by Hao et al., we begin to investigate the idea that different LLMs require unique prompts to achieve optimal performance in the same downstream task. To address this, we propose the MAPO method, specifically designed to optimize individual prompts for different LLMs within the same downstream application. This allows each LLM to operate at its fullest potential, aligning with the notion that a one-size-fits-all approach to prompts may not be universally effective.
>
> We hope this addresses your concerns.
>
>     Typos Grammar Style And Presentation Improvements: In Figure 3, better prompt should be P_o.
>
> Thanks for your suggestion. We have modified the Figure 3 in https://p.sda1.dev/12/35aebb7ebe7acb6a433e25c2954ebbce/framework.png, and we will revise it in the final version.
>
> Thank you very much for your valuable suggestions and comments on our manuscript. We hope that our responses have addressed your concerns. We would be most grateful if you could consider **raising your score** in light of these clarifications. Thank you once again for your time and consideration!

---

### Official Review · Reviewer_Wer2 · 2023-08-04

**Soundness:** 3

**Excitement:**

3: Ambivalent: It has merits (e.g., it reports state-of-the-art results, the idea is nice), but there are key weaknesses (e.g., it describes incremental work), and it can significantly benefit from another round of revision. However, I won't object to accepting it if my co-reviewers champion it.

**Paper Topic And Main Contributions:**

This paper first quantifies the performance variations across large language models (LLMs) given the same set of prompts. The large variance in performance motivates the research problem of this paper, i.e., adaptively optimize the prompt for each LLM given a specific downstream task. The proposed RL-based method, model-adaptive prompt optimizer (MAPO), resembles the pipeline of RLHF (first supervised fine-tuning, then train a reward model, finally reinforce the policy with PPO), and is empirically shown to have led to gains in three types of downstream tasks by large margins.

**Questions For The Authors:**

- Have you studied the necessity of RL in your proposed method?

**Reasons To Accept:**

- This paper provides quantitative initiative for adapting prompt for an LLM.
- The proposed method yields promising improvements.

**Reasons To Reject:**

- The proposed method may not be useful for low-resource tasks, as it is infeasible to get enough data for both the supervised fine-tuning and the reinforcement learning stages.
- The result does not demonstrate why RL is necessary. A fair comparison against MAPO should be continuing training SFT model for longer and presumably on larger warmup dataset. The results will be strengthened a lot if the authors can show MAPO outperforms a stronger SFT model.

**Reproducibility:**

3: Could reproduce the results with some difficulty. The settings of parameters are underspecified or subjectively determined; the training/evaluation data are not widely available.

**Reviewer Confidence:**

3: Pretty sure, but there's a chance I missed something. Although I have a good feel for this area in general, I did not carefully check the paper's details, e.g., the math, experimental design, or novelty.

**Typos Grammar Style And Presentation Improvements:**

- L148 I had trouble understanding what “match degree” means when I first read it. Perhaps explain it before using this term.

---

> ### Author Rebuttal · Authors · 2023-08-29
>
> Thank you very much for your valuable suggestions and comments on our manuscript. Below are our responses to each of the points of criticism, questions, and typos you've raised.
>
>     R1: The proposed method may not be useful for low-resource tasks, as it is infeasible to get enough data for both the supervised fine-tuning and the reinforcement learning stages.
>
> 1. Thanks for your question. In response to the concern that our proposed method may not be effective for low-resource tasks due to the need for substantial data for both supervised fine-tuning and reinforcement learning, we'd like to draw attention to our comprehensive set of experiments. Our ablation study illustrated in Figure 4 shows that increasing the size of the warm-up dataset generally enhances MAPO’s performance. **Importantly, even with considerable reductions in the dataset size, the drop in performance is modest. The ablation study results demonstrate that the method performs well even when the size of the warm-up dataset varies. This indicates that even a smaller dataset could be effective, thereby making the method suitable for low-resource tasks. (Note: Upon further review, we find that the LLMs should be arranged from left to right as GPT-J, BLOOM, and LLaMA, instead of BLOOM, GPT-J, and LLaMA as currently shown. We apologize for this oversight and will correct this error in the final version of the manuscript.).** The blank section indicates that the content is the same as the cell above. Due to the limitations of Markdown syntax, merging cells is not possible.
>
> 2. However, increasing the warm-up dataset size often requires more training time and resources, and the boost in performance doesn't always align proportionally with the increment in data size. **Consequently, we conduct experiments to identify the optimal proportion of the warm-up dataset, seeking to achieve the best results within resource constraints as shown in the Table 11**. In our evaluations, we utilize various proportions of the warm-up dataset: 20%, 40%, 60%, 80%, and 100%. Our results show that for LLaMA, performance improvements begin to level off at 60%. For GPT-J, this slowdown happens at 80%. BLOOM, on the other hand, doesn't present a distinct slowdown in performance improvements. **Based on these insights, we opt to use 60% of the dataset for LLaMA's MAPO implementation (encompassing both SFT and RL phases), 100% for BLOOM, and 80% for GPT-J**. Therefore, our method consistently targets optimal results while considering resource constraints, proving its suitability for low-resource tasks.
>
> 3. We also examine the trade-offs between performance and dataset size as shown in the following tables. Taking LLaMA as a specific example, when we reduce the dataset from 100% to 80%, 60%, and 20%, the average absolute performance declines are just 0.3, 0.6, 1.9, and 2.8 respectively. The relative performance declines were only 1%, 2%, 6.7%, and 10.2%, even as the dataset sizes are reduced by 55,822, 111,645, 167,468, and 223,291, which correspond to relative reductions of 25%, 50%, 66.7%, 150%, and 400%. **This consistent and minimal decline in performance across such dataset reductions is not specific to LLaMA**; we observed a similar trend across all other tested models, confirming the robustness of our approach in low-resource scenarios. **The detailed performance metrics, particularly using the example of LLaMA, show that even with substantial reductions in dataset size, the decline in performance is relatively minimal. This supports the method's robustness in scenarios where data is limited.**
>
> 4. Moreover, we conduct experiments on various general NLP tasks in a few-shot setting with just 10% of the data, and we still observe improvements. This underscores the strong generalizability of our method even in conditions of data scarcity. The results from the few-shot experiments further strengthen the argument for the method's effectiveness in low-resource settings. **These experiments show that the method still produces improvements even when just 10% of the data is used.**
>
> In summary, our experimental findings collectively suggest that our method is not just feasible but also effective in low-resource situations, covering both the supervised fine-tuning and reinforcement learning stages. We will incorporate these findings into our revised manuscript to fully address concerns about our method's applicability in low-resource settings.
>
>
> Table11: Performance of different proportion of warm-up dataset in various downstream tasks by three LLMs. Q: QA, C: classification, G:generation. ↓means the number of performance decline. ↓(%) means the percentage of performance decline. D-↓ means the number of data reduction. D-↓(%) means the percentage of data reduction.
>
> #### LLaMA
> | Tasks      | Dataset |        20% |        40% |        60% |       80% |   100% |
> |------------|---------|-----------:|-----------:|-----------:|----------:|-------:|
> | QA         | AdverQA |       22.2 |       23.2 |       25.1 |      25.4 |     26 |
> |            | OpenQA  |       14.3 |       15.1 |       16.1 |      16.3 |   16.6 |
> |            | CloseQA |         13 |       13.8 |       14.8 |      15.2 |   15.4 |
> | Class      | News    |       10.2 |       11.5 |       12.4 |      12.5 |   12.7 |
> |            | Movie   |       79.8 |       80.2 |       82.5 |      82.7 |   83.1 |
> |            | QASC    |       70.5 |       71.4 |       72.8 |      73.3 |   73.4 |
> | Gen        | Topics  |       17.5 |       18.2 |       19.5 |      19.8 |   19.9 |
> |            | Summary |       10.1 |       11.3 |       12.2 |      12.4 |   12.8 |
> |            | Explan  |        7.3 |        8.2 |        9.1 |       9.5 |     10 |
> | Average    | -       |       27.2 |       28.1 |       29.4 |      29.7 |     30 |
> | ↓      | -   |    **2.8** |    **1.9** |    **0.6** |   **0.3** | -  |
> |↓(%)   | -   |   **10.2** |    **6.7** |      **2** |     **1** | -  |
> | Dataset    | -       |      55822 |     111645 |     167468 |    223291 | 279113 |
> | D-↓    | -   | **223291** | **167468** | **111645** | **55822** | -  |
> | D-↓(%) | -   |    **400** |    **150** |   **66.7** |    **25** | -  |
>
>
>
> #### BLOOM
> | Tasks      | Dataset |        20% |        40% |        60% |       80% |   100% |
> |------------|---------|-----------:|-----------:|-----------:|----------:|-------:|
> | QA         | AdverQA |       17.8 |       18.6 |       18.9 |      19.2 |   19.5 |
> |            | OpenQA  |       25.4 |       26.1 |       26.3 |      26.5 |   27.2 |
> |            | CloseQA |          8 |        8.6 |        8.8 |       9.1 |    9.4 |
> | Class      | News    |       96.8 |       97.4 |       97.6 |        98 |   98.7 |
> |            | Movie   |       89.8 |       90.9 |       91.3 |      92.5 |   93.3 |
> |            | QASC    |         98 |       98.6 |       98.7 |      98.8 |   99.9 |
> | Gen        | Topics  |         35 |       35.7 |         36 |      36.4 |   36.2 |
> |            | Summary |       47.9 |       48.9 |       49.3 |      49.7 |   50.2 |
> |            | Explan  |        7.5 |        8.1 |        8.3 |       8.6 |    8.9 |
> | Average    | -       |       47.4 |       48.1 |       48.4 |      48.8 |   49.3 |
> | ↓      | -   |    **1.9** |    **1.2** |    **0.9** |   **0.5** | -  |
> | ↓(%)   | -   |      **4** |    **2.4** |    **1.9** |     **1** | -  |
> | Dataset    | -       |      55822 |     111645 |     167468 |    223291 | 279113 |
> | D-↓    | -   | **223291** | **167468** | **111645** | **55822** | -  |
> | D-↓(%) | -   |    **400** |    **150** |   **66.7** |    **25** | -  |
>
>
>
> #### GPT-J
> | Tasks      | Dataset |        20% |        40% |        60% |       80% |   100% |
> |------------|---------|-----------:|-----------:|-----------:|----------:|-------:|
> | QA         | AdverQA |        9.2 |        9.3 |        9.9 |        11 |     11 |
> |            | OpenQA  |         19 |       19.1 |       19.5 |        21 |   21.1 |
> |            | CloseQA |        8.4 |        8.6 |        9.4 |      10.2 |   10.3 |
> | Class      | News    |        4.6 |        4.8 |        5.3 |       6.3 |    6.4 |
> |            | Movie   |       51.8 |       52.2 |       52.8 |      53.9 |   53.9 |
> |            | QASC    |       53.8 |       54.4 |       55.2 |      56.8 |   56.9 |
> | Gen        | Topics  |       21.3 |       21.6 |       22.2 |      23.4 |   23.6 |
> |            | Summary |         16 |       16.3 |       16.8 |      17.8 |   18.1 |
> |            | Explan  |       11.2 |       11.4 |       11.9 |      12.9 |   13.2 |
> | Average    | -       |       21.7 |         22 |       22.6 |      23.7 |   23.8 |
> | ↓      | -  |    **2.1** |    **1.9** |    **1.3** |   **0.1** | -  |
> | ↓(%)  | -  |    **9.8** |    **8.5** |    **5.7** |   **0.6** | -  |
> | Dataset    | -       |      55822 |     111645 |     167468 |    223291 | 279113 |
> | D-↓    | -   | **223291** | **167468** | **111645** | **55822** | -  |
> | D-↓(%) | -   |    **400** |    **150** |   **66.7** |    **25** | -  |
>
>
>
>
>     R2: The result does not demonstrate why RL is necessary. A fair comparison against MAPO should be continuing training SFT model for longer and presumably on larger warmup dataset. The results will be strengthened a lot if the authors can show MAPO outperforms a stronger SFT model.
>
> 1. Thanks for your question. In response to the query about why RL is necessary and how it compares to simply extending SFT with a larger warm-up dataset, we would like to highlight our targeted experiments. We use various proportions of the warm-up dataset, specifically 20%, 40%, 60%, 80%, and 100%, to progressively increase the SFT training data and then introduce RL to it. For each subset, such as the 20% data subset, we perform both SFT and SFT+RL (that is MAPO), with the SFT+RL approach building upon the same SFT subset. Performance metrics for each subset on the target tasks are carefully recorded. We report the results in Table 12. The blank section indicates that the content is the same as the cell above. Due to the limitations of Markdown syntax, merging cells is not possible.
>
>
> 2. **Our findings consistently show that RL adds value to the performance beyond what is achieved by SFT alone across all proportions of the dataset. This affirms the effectiveness of RL irrespective of the SFT dataset size.** However, as the proportion of the warm-up dataset increases, the margin of improvement from adding RL begins to decline. **While one could hypothesize that adding RL to a very large SFT dataset might not result in as significant an improvement as it would for a smaller dataset, this observation actually underscores our method's suitability for low-resource scenarios.**
>
> 3. Therefore, our experimental evidence establishes that RL plays a significant role in enhancing the model's performance and shows that MAPO, which includes both SFT and RL, consistently outperforms a model relying solely on an extended SFT training regime. This strengthens our argument and answers the concern about the necessity of incorporating RL into the method. We will include these additional findings in our revised manuscript to comprehensively address questions about the efficacy and necessity of RL in our approach.
>
>
> Table12: Performance of different proportion of warm-up dataset in various downstream tasks by three LLMs. Q: QA, C: classification, G:generation. SFT means using SFT-optimized prompts without RL.
>
> #### LLaMA
> | Tasks   | Dataset |  20% |         |  40% |         |  **60%** |           |  80% |       | 100% |       |
> |---------|:-------:|:----:|:-------:|:----:|:-------:|:--------:|:---------:|:----:|:-----:|:----:|:-----:|
> |         |         | +SFT |  +MAPO  | +SFT |  +MAPO  | **+SFT** | **+MAPO** | +SFT | +MAPO | +SFT | +MAPO |
> | QA      | AdverQA | 19.6 |   22.2  |  22  |   23.2  | **23.2** |  **25.1** | 24.5 |  25.4 | 24.4 |   26  |
> |         |  OpenQA | 12.8 |   14.3  | 13.5 |   15.1  | **15.4** |  **16.1** | 14.9 |  16.3 | 15.9 |  16.6 |
> |         | CloseQA | 11.5 |    13   | 12.2 |   13.8  | **13.9** |  **14.8** | 14.5 |  15.2 | 14.8 |  15.4 |
> | Class   |   News  |  8.8 |   10.2  |  9.8 |   11.5  | **10.1** |  **12.4** |  11  |  12.5 | 10.8 |  12.7 |
> |         |  Movie  | 77.5 |   79.8  |  79  |   80.2  | **81.3** |  **82.5** | 80.6 |  82.7 | 81.5 |  83.1 |
> |         |   QASC  | 67.3 |   70.5  | 68.3 |   71.4  | **70.2** |  **72.8** | 71.2 |  73.3 | 71.6 |  73.4 |
> | Gen     |  Topics | 16.1 |   17.5  | 17.2 |   18.2  | **18.6** |  **19.5** | 18.3 |  19.8 |  18  |  19.9 |
> |         | Summary |  8.2 |   10.1  |  9.4 |   11.3  | **10.7** |  **12.2** | 10.5 |  12.4 | 11.6 |  12.8 |
> |         |  Explan |  5.6 |   7.3   |  7.6 |   8.2   |  **8.2** |  **9.1**  |  8.8 |  9.5  |  8.8 |   10  |
> | Average |    -    | 25.3 |   27.2  | 26.6 |   28.1  |  **28**  |  **29.4** | 28.3 |  29.7 | 28.6 |   30  |
> | ↑       |    -    |   -  | **1.9** |   -  | **1.5** |     -    |  **1.4**  |   -  |  1.4  |   -  |  1.4  |
> | ↑(%)    |    -    |   -  | **7.5** |   -  | **5.6** |     -    |   **5**   |   -  |  4.9  |   -  |  4.9  |
>
> #### BLOOM
> | Tasks   | Dataset |  20% |         |  40% |         |  60% |         |  80% |         | **100%** |          |
> |---------|:-------:|:----:|:-------:|:----:|:-------:|:----:|:-------:|:----:|:-------:|:--------:|:--------:|
> |         |         |  +SFT |   +MAPO  |  +SFT |   +MAPO  |  +SFT |   +MAPO  |  +SFT |   +MAPO  |  **+SFT** | **+MAPO** |
> | QA      | AdverQA | 14.6 |   17.8  | 16.9 |   18.6  | 17.5 |   18.9  | 17.7 |   19.2  | **18.3** | **19.5** |
> |         |  OpenQA | 22.1 |   25.4  | 24.3 |   26.1  | 24.9 |   26.3  | 25.1 |   26.5  | **26.7** | **27.2** |
> |         | CloseQA |  6.6 |    8    |  7.7 |   8.6   |  7.3 |   8.8   |  7.8 |   9.1   |  **7.8** |  **9.4** |
> | Class   |   News  | 93.6 |   96.8  | 95.8 |   97.4  | 95.5 |   97.6  | 96.2 |    98   | **95.5** | **98.7** |
> |         |  Movie  | 87.1 |   89.8  | 88.3 |   90.9  | 88.9 |   91.3  | 90.7 |   92.5  | **92.6** | **93.3** |
> |         |   QASC  |  96  |    98   | 97.6 |   98.6  | 97.1 |   98.7  | 97.4 |   98.8  | **99.9** | **99.9** |
> | Gen     |  Topics | 32.7 |    35   | 33.6 |   35.7  | 34.6 |    36   | 34.7 |   36.4  | **34.8** | **36.2** |
> |         | Summary | 45.8 |   47.9  | 47.1 |   48.9  | 47.3 |   49.3  | 48.5 |   49.7  | **48.8** | **50.2** |
> |         |  Explan |  6.2 |   7.5   |  7.3 |   8.1   |  7.1 |   8.3   |  6.3 |   8.6   |  **6.8** |  **8.9** |
> | Average |    -    |  45  |   47.4  | 46.5 |   48.1  | 46.7 |   48.4  | 47.2 |   48.8  | **47.9** | **49.3** |
> | ↑       |    -    |   -  | **2.4** |   -  | **1.6** |   -  | **1.7** |   -  | **1.6** |     -    |  **1.4** |
> | ↑(%)    |    -    |   -  | **5.3** |   -  | **3.4** |   -  | **3.6** |   -  | **3.4** |     -    |  **2.9** |
>
>
> #### GPT-J
> | Tasks   | Dataset |  20% |          |  40% |         |  60% |         |  **80%** |          | 100% |         |
> |---------|:-------:|:----:|:--------:|:----:|:-------:|:----:|:-------:|:--------:|:--------:|:----:|:-------:|
> |         |         |  +SFT |   +MAPO   |  +SFT |   +MAPO  |  +SFT |   +MAPO  |  **+SFT** | **+MAPO** |  +SFT |   +MAPO  |
> | QA      | AdverQA |  6.8 |    9.2   |  7.7 |   9.3   |  7.9 |   9.9   |  **9.4** |  **11**  |  9.9 |    11   |
> |         |  OpenQA | 17.3 |    19    | 17.3 |   19.1  | 18.2 |   19.5  | **20.3** |  **21**  | 19.8 |   21.1  |
> |         | CloseQA |  7.1 |    8.4   |  7.3 |   8.6   |  8.2 |   9.4   |  **8.4** | **10.2** |  9.6 |   10.3  |
> | Class   |   News  |  2.4 |    4.6   |  2.8 |   4.8   |  3.9 |   5.3   |  **5.5** |  **6.3** |  5.5 |   6.4   |
> |         |  Movie  | 49.2 |   51.8   | 50.7 |   52.2  | 51.7 |   52.8  | **52.7** | **53.9** | 51.8 |   53.9  |
> |         |   QASC  | 50.3 |   53.8   | 52.8 |   54.4  | 53.6 |   55.2  | **56.3** | **56.8** | 55.2 |   56.9  |
> | Gen     |  Topics | 19.5 |   21.3   |  20  |   21.6  | 20.8 |   22.2  | **21.6** | **23.4** | 21.8 |   23.6  |
> |         | Summary | 13.8 |    16    | 14.7 |   16.3  | 15.7 |   16.8  | **16.7** | **17.8** | 17.1 |   18.1  |
> |         |  Explan |  8.5 |   11.2   |  9.9 |   11.4  | 10.1 |   11.9  | **10.7** | **12.9** | 11.8 |   13.2  |
> | Average |    -    | 19.4 |   21.7   | 20.4 |    22   | 21.1 |   22.6  | **22.4** | **23.7** | 22.5 |   23.8  |
> | ↑       |    -    |   -  |  **2.3** |   -  | **1.6** |   -  | **1.5** |     -    |  **1.3** |   -  | **1.3** |
> | ↑(%)    |    -    |   -  | **11.9** |   -  | **7.8** |   -  | **7.1** |     -    |  **5.8** |   -  | **5.8** |
>
>
> 4. Moreover, we have tried different number of epochs to see if extended training time consistently improves SFT performance before submitting the manuscript as shown in Table 13. **Extending the training time does not consistently lead to performance improvements for SFT. In some instances, the performance even declines.** It is important to note that we save the best-performing models in real-time during training, as the peak performance does not necessarily occur at the final epoch. To further validate that extended training time is not the reason SFT underperforms compared to MAPO, we also conduct additional experiments that increasing the number of epochs (epoch=50) for SFT in three LLMs (LLaMA, BLOOM, GPT-J). **The findings indicate that merely increasing the number of epochs does not significantly improve the SFT performance in all tasks.** The results are shown in Table 13.
>
>
> Table13: Performance of different number of epochs when training SFT. Best Performance means the best performance within 20 epochs. Best Epoch means the epoch corresponding to the best performance. We list the performance in the 1,5,10,15,20,50 epochs. We bold the performance metrics where a longer training epoch (epoch=50) results in a decline in performance.
>
> #### BLOOM
> | Task | Dataset | Best Performance | Best Epoch | Epoch |      |      |      |      |          |
> |------|---------|------------------|------------|:-----:|:----:|:----:|:----:|:----:|:--------:|
> | -    | -       | -                | -          | 1     | 5    | 10   | 15   | 20   | 50       |
> | QA   | AdverQA | 18.3             | 18         | 14.8  | 14.3 | 15.4 | 18.2 | 18.3 | **18.2** |
> |      | OpenQA  | 26.7             | 14         | 26    | 26   | 26.1 | 26.7 | 26.7 | **26.6** |
> |      | CloseQA | 7.8              | 19         | 6.8   | 7    | 7    | 7.5  | 7.8  | **7.8**  |
> | CLS  | News    | 95.5             | 20         | 93.2  | 93.7 | 94.3 | 95.3 | 95.5 | 95.7     |
> |      | Movie   | 92.6             | 15         | 91.4  | 91.7 | 91.3 | 92.6 | 92.4 | 92.5     |
> |      | QASC    | 99.9             | 7          | 99.6  | 99.8 | 99.9 | 99.9 | 99.9 | **99.9** |
> | GEN  | Topics  | 34.8             | 19         | 30.3  | 31.5 | 33.8 | 34.2 | 34.8 | **34.8** |
> |      | Summary | 48.8             | 18         | 46.1  | 46.4 | 47.5 | 48.4 | 48.8 | 48.9     |
> |      | Explan  | 6.8              | 15         | 5.9   | 6.3  | 6.4  | 6.8  | 6.5  | 6.7      |
>
>
> #### GPT-J
> | Task | Dataset | Best Performance | Best Epoch | Epoch |      |      |      |      |          |
> |------|---------|------------------|------------|:-----:|:----:|:----:|:----:|:----:|:--------:|
> | -    | -       | -                | -          | 1     | 5    | 10   | 15   | 20   | 50       |
> | QA   | AdverQA | 9.4              | 20         | 4.4   | 5.3  | 6.7  | 8.8  | 9.4  | 9.6      |
> |      | OpenQA  | 20.3             | 15         | 17.5  | 18.1 | 18.8 | 20.3 | 20.2 | **20**   |
> |      | CloseQA | 8.4              | 20         | 7     | 7.3  | 7.8  | 8.2  | 8.4  | 8.6      |
> | CLS  | News    | 5.5              | 20         | 1.4   | 2.6  | 3.1  | 4.8  | 5.5  | 5.9      |
> |      | Movie   | 52.7             | 15         | 51.1  | 51.6 | 52.6 | 52.7 | 52.4 | 52.7     |
> |      | QASC    | 56.3             | 19         | 54.3  | 54.6 | 54.5 | 55.4 | 56.3 | **56.3** |
> | GEN  | Topics  | 21.6             | 18         | 17.9  | 20.2 | 20.7 | 21.3 | 21.6 | **21.6** |
> |      | Summary | 16.7             | 20         | 13.2  | 14.5 | 15.2 | 16.3 | 16.7 | 17.1     |
> |      | Explan  | 10.7             | 19         | 8.8   | 9.2  | 9.9  | 9.6  | 10.7 | 11       |
>
>
> #### LLaMA
> | Task | Dataset | Best Performance | Best Epoch | Epoch |      |      |      |      |          |
> |------|---------|------------------|------------|:-----:|:----:|:----:|:----:|:----:|:--------:|
> | -    | -       | -                | -          | 1     | 5    | 10   | 15   | 20   | 50       |
> | QA   | AdverQA | 23.2             | 20         | 5.8   | 9.3  | 13.3 | 18.7 | 23.2 | 29.5     |
> |      | OpenQA  | 15.4             | 15         | 13.6  | 14.1 | 14.6 | 15.4 | 15.3 | 16.8     |
> |      | CloseQA | 13.9             | 20         | 10.8  | 11.3 | 12.9 | 13.5 | 13.9 | 14.5     |
> | CLS  | News    | 10.1             | 13         | 3.4   | 5.7  | 9.3  | 10.1 | 10.1 | **10.1** |
> |      | Movie   | 81.3             | 19         | 79.2  | 79.6 | 80.5 | 81.1 | 81.3 | **81.2** |
> |      | QASC    | 70.2             | 20         | 62.8  | 63.7 | 65.9 | 69.2 | 70.2 | 71.9     |
> | GEN  | Topics  | 18.6             | 20         | 15.5  | 16.1 | 17.7 | 18.4 | 18.6 | 19.3     |
> |      | Summary | 10.7             | 20         | 6.6   | 7.1  | 8.6  | 9.4  | 10.7 | 11.5     |
> |      | Explan  | 8.2              | 14         | 7.3   | 7.5  | 7.5  | 8.2  | 8.2  | 8.3      |
>
>
> Based on the comparative results between SFT and MAPO across different proportions of datasets, theoretically, MAPO is capable of further improving prompt optimization on top of whatever performance SFT achieves.
>
>
>     Q:  Have you studied the necessity of RL in your proposed method?
>
> 1. Thanks for your suggestion, we report the results of SFT in Tables 3, 4, and 5 to compare the results of SFT and RL (i.e. MAPO). **We can observe that the performance improvements gained solely from using SFT are less than half of those achieved by our proposed MAPO method, both on similar tasks (Table 3) and general NLP tasks (Tables 4 and 5).** This clearly indicates the effectiveness of MAPO in optimizing model-adaptive prompts. The blank section indicates that the content is the same as the cell above. Due to the limitations of Markdown syntax, merging cells is not possible.
>
>
>
> Table 3: The few-shot performance of SFT, MAPO with SOTA prompt optimizing baselines in downstream tasks. F: Finetuning, C: Continous prompt, D: Discrete prompt.
> |       |              | SST-2    | YelpP.   | MR       | CR       | RTE      | QNLI     | SNLI     | MNLI     | MRPC     |
> |-------|--------------|----------|----------|----------|----------|----------|----------|----------|----------|----------|
> | F     | Finetuning   | 80.6     | 88.7     | 67.4     | 73.3     | 58.6     | 60.2     | 54.6     | 47.8     | 77.4     |
> | C     | Softprompt   | 73.8     | 88.6     | 74.1     | 75.9     | 54.7     | 49.7     | 36.1     | 33.2     | 51.6     |
> |       | Black-Box    | 89.1     | 93.2     | 86.6     | 87.4     | 52.6     | 48.8     | 46.6     | 42.9     | 61.6     |
> |       | Autoprompt   | 75.0     | 79.8     | 62.0     | 57.5     | -        | -        | -        | -        | -        |
> | D     | Manual       | 82.8     | 83.0     | 80.9     | 79.6     | 51.6     | 50.8     | 31.1     | 51.7     | 67.4     |
> |       | In-Context   | 85.9     | 89.6     | 80.6     | 85.5     | 60.4     | 53.8     | 47.1     | 53.4     | 45.8     |
> |       | Instructions | 89.0     | 84.4     | 85.2     | 80.8     | -        | -        | -        | -        | -        |
> |       | GrIPS        | 87.1     | 88.2     | 86.1     | 80.0     | -        | -        | -        | -        | -        |
> |       | RLprompt     | 92.5     | 95.1     | 87.1     | 89.5     | -        | -        | -        | -        | -        |
> |       | TEMPERA      | 91.9     | 92.6     | 88.0     | 91.1     | 60.3     | 57.4     | 56.4     | 45.2     | 74.0     |
> |       | AMA          | 95.7     | -        | -        | -        | 75.1     | -        | -        | -        | -        |
> | **D** | **SFT**      | **94.9** | **92.0** | **88.5** | **87.6** | **74.3** | **62.5** | **58.8** | **54.6** | **78.5** |
> | **D** | **MAPO**     | **96.1** | **93.5** | **90.2** | **88.9** | **75.3** | **63.1** | **60.0** | **55.7** | **79.3** |
>
>
> Table 4: Zero-shot domain transfer performance based on BLOOM and GPT-J with original, SFT-optimized and MAPO-Optimized prompts. CLS: Classification, M: MAPO. The (↑(%) ) and ↑(%) represent the increase degree of MAPO-optimized prompts compared with original prompts in each dataset and task, respectively (The same below).
> | Task   | Dataset    | BLOOM | +SFT | (↑(%)) | **↑(%)** | +MAPO | (↑(%) | **↑(%)** | GPT-J | +SFT | (↑(%)) | **↑(%)** | +MAPO | (↑(%)) | **↑(%)** |
> |--------|------------|-------|------|--------|----------|-------|-------|----------|-------|------|--------|----------|-------|--------|----------|
> | Coref. | xwinograd  | 60.1  | 60.2 | 0.2    | **0.2**  | 60.6  | 0.9   | **0.9**  | -     | -    | -      | **-**    | -     | -      | **-**    |
> | NLU    | BoolQ      | 67.9  | 68.0 | 0.1    | **0.4**  | 68.2  | 0.4   | **0.9**  | 67.2  | 67.4 | 0.3    | **0.2**  | 67.9  | 1.0    | **0.5**  |
> |        | CB         | 77.6  | 77.8 | 0.3    |          | 78.1  | 0.6   |          | 83.9  | 84.1 | 0.2    |          | 84.2  | 0.4    |          |
> |        | COPA       | 74.0  | 74.3 | 0.4    |          | 75.0  | 1.4   |          | 84.0  | 84.2 | 0.2    |          | 84.2  | 0.2    |          |
> |        | MultiRC    | 59.7  | 60.3 | 1.0    |          | 60.4  | 1.2   |          | 63.8  | 63.9 | 0.2    |          | 64.1  | 0.5    |          |
> |        | ReCoRD     | 69.8  | 70.1 | 0.4    |          | 70.2  | 0.6   |          | 74.4  | 74.5 | 0.1    |          | 74.7  | 0.4    |          |
> |        | WiC        | 61.4  | 61.6 | 0.3    |          | 62.0  | 1.0   |          | 61.0  | 61.1 | 0.2    |          | 61.3  | 0.5    |          |
> |        | WSC        | 64.4  | 64.7 | 0.5    |          | 65.1  | 1.1   |          | 77.9  | 78.0 | 0.1    |          | 78.1  | 0.3    |          |
> | NLI    | ANLIR1     | 31.5  | 31.7 | 0.6    | **0.6**  | 32.1  | 1.9   | **1.3**  | 37.8  | 38.0 | 0.5    | **0.3**  | 38.2  | 1.1    | **0.7**  |
> |        | ANLIR2     | 35.1  | 35.2 | 0.3    |          | 35.4  | 0.9   |          | 37.9  | 38.0 | 0.3    |          | 38.3  | 1.1    |          |
> |        | ANLIR3     | 37.1  | 37.5 | 1.1    |          | 37.8  | 1.9   |          | 40.9  | 41.0 | 0.2    |          | 41.1  | 0.5    |          |
> |        | StoryCloze | 79.0  | 79.2 | 0.3    |          | 79.5  | 0.6   |          | 87.8  | 87.9 | 0.1    |          | 87.9  | 0.1    |          |
> | CLS    | Amazon     | 65.2  | 66.4 | 1.8    | **1.4**  | 67.7  | 3.8   | **3.3**  | 68.2  | 68.7 | 0.7    | **0.6**  | 69.4  | 1.8    | **1.6**  |
> |        | DBPedia    | 70.5  | 71.2 | 1.0    |          | 72.5  | 2.8   |          | 83.9  | 84.2 | 0.4    |          | 85.1  | 1.4    |          |
> | QA     | DROP       | 67.9  | 68.2 | 0.4    | **1.9**  | 69.9  | 2.9   | **5.6**  | 51.6  | 51.9 | 0.6    | **1.3**  | 52.8  | 2.3    | **3.2**  |
> |        | NQ         | 15.1  | 15.4 | 2.0    |          | 16.1  | 6.6   |          | 19.6  | 20.1 | 2.6    |          | 20.8  | 6.1    |          |
> |        | RealTimeQA | 29.0  | 30.2 | 4.1    |          | 31.5  | 8.6   |          | 36.0  | 36.5 | 1.4    |          | 37.2  | 3.3    |          |
> |        | WebQs      | 34.8  | 35.1 | 0.9    |          | 36.3  | 4.3   |          | 44.1  | 44.3 | 0.5    |          | 44.6  | 1.1    |          |
>
> Table 5: Zero-shot domain transfer performance based on LLaMA with original, SFT-optimized and MAPO-Optimized prompts. RS: Commonsense Reasoning.
> | Task | Dataset    | LLaMA | +SFT | (↑(%)) | **↑(%)** | +MAPO | (↑(%)) | **↑(%)** |
> |------|------------|-------|------|--------|----------|-------|--------|----------|
> | RS   | BoolQ      | 76.5  | 76.6 | 0.1    | **0.1**  | 76.7  | 0.3    | **0.5**  |
> |      | PIQA       | 79.8  | 79.9 | 0.1    |          | 80.0  | 0.3    |          |
> |      | SIQA       | 48.9  | 48.9 | 0.0    |          | 49.0  | 0.2    |          |
> |      | HellaSwag  | 76.1  | 76.2 | 0.1    |          | 76.5  | 0.5    |          |
> |      | WinoGrande | 70.1  | 70.2 | 0.1    |          | 70.5  | 0.6    |          |
> |      | ARC-e      | 72.8  | 72.9 | 0.1    |          | 73.2  | 0.6    |          |
> |      | ARC-c      | 47.6  | 47.6 | 0.0    |          | 47.8  | 0.4    |          |
> |      | OBQA       | 57.2  | 57.4 | 0.3    |          | 57.9  | 1.2    |          |
> | QA   | NQ         | 16.8  | 17.2 | 2.4    | **1.4**  | 18.1  | 7.7    | **4.7**  |
> |      | RACE       | 50.0  | 50.2 | 0.4    |          | 50.8  | 1.6    |          |
>
>
> 2. We also investigate the separate effect of PPO and RRHF in Table 9 in Appendix, which demonstrate the important roles played by both PPO and RRHF in enhancing the performance of MAPO. We put the detailed results as follows. We also add the results of SFT-optimized prompts for comparison.
>
> Table 9: The separate effect of PPO and RRHF, which demonstrate the important roles played by both PPO and RRHF in enhancing the performance of MAPO.
> | Tasks   | Dataset | +SFT |       |       | +PPO |         |         | +RRHF |         |         | +MAPO |         |         |
> |---------|---------|:-------------:|:-----:|:-----:|:-------------:|:-------:|:-------:|:--------------:|:-------:|:-------:|:--------------:|:-------:|:-------:|
> |         |         | BLOOM         | GPT-J | LLaMA | BLOOM         |  GPT-J  |  LLaMA  | BLOOM          |  GPT-J  |  LLaMA  | BLOOM          | GPT-J   | LLaMA   |
> | QA      | AdverQA | 18.3          | 9.4   | 23.2  | 18.8          | 9.6     | 24.3    | 19.2           | 9.9     | 24.8    | 19.5           | 11.0    | 25.1    |
> |         | OpenQA  | 26.7          | 20.3  | 15.4  | 26.9          | 20.7    | 16.0    | 27.0           | 20.9    | 15.8    | 27.2           | 21.0    | 16.1    |
> |         | CloseQA | 7.8           | 8.4   | 13.9  | 8.3           | 8.8     | 14.4    | 9.0            | 9.8     | 14.6    | 9.4            | 10.2    | 14.8    |
> | Class   | News    | 95.5          | 5.5   | 10.1  | 96.9          | 6.1     | 12.1    | 98.3           | 6.2     | 12.2    | 98.7           | 6.3     | 12.4    |
> |         | Movie   | 92.6          | 52.7  | 81.3  | 93.1          | 53.1    | 82.1    | 93.0           | 53.6    | 82.2    | 93.3           | 53.9    | 82.5    |
> |         | QASC    | 99.9          | 56.3  | 70.2  | 99.9          | 56.7    | 72.4    | 99.9           | 56.5    | 71.2    | 99.9           | 56.8    | 72.8    |
> | Gen     | Topics  | 34.8          | 21.6  | 18.6  | 35.9          | 23.1    | 18.9    | 35.2           | 23.2    | 19.1    | 36.2           | 23.4    | 19.5    |
> |         | Summary | 48.8          | 16.7  | 10.7  | 50.1          | 17.2    | 11.7    | 49.4           | 17.7    | 12.0    | 50.2           | 17.8    | 12.2    |
> |         | Explan  | 6.8           | 10.7  | 8.2   | 8.3           | 11.5    | 8.4     | 7.2            | 12.4    | 8.7     | 8.9            | 12.9    | 9.1     |
> | Average | -       | 47.9          | 22.4  | 28.0  | 48.7          | 23.0    | 28.9    | 48.7           | 23.4    | 29.0    | 49.3           | 23.7    | 29.4    |
> | ↑       | -       | -             | -     | -     | **0.8**       | **0.6** | **0.9** | **0.8**        | **1.7** | **1.0** | **1.4**        | **1.3** | **1.4** |
> | ↑(%)    | -       | -             | -     | -     | **1.7**       | **2.7** | **3.2** | **1.7**        | **4.5** | **3.6** | **2.9**        | **5.8** | **5.0** |
>
>
>     Typos Grammar Style And Presentation Improvements: L148 I had trouble understanding what “match degree” means when I first read it. Perhaps explain it before using this term.
>
>
> We apologize for any confusion the term "match degree" may have caused. In our methodology, **"match degree" actually refers to the similarity between the outputs generated using a candidate prompt and the ground truth output, not the similarity between the output generated by each candidate prompt and by the original prompt.** This similarity is quantified using specific evaluation metrics tailored to the task at hand. For instance, in Question-Answering tasks, we use the F1 score to evaluate how closely the answers match. In classification tasks, we use accuracy, and for generative tasks, we employ the ROUGE-L score. **The purpose is to identify the candidate prompt that produces an output most similar to the ground truth output**, effectively helping us find the 'optimal' prompt for downstream tasks. We'll make sure to define this term early on in our revised manuscript for better clarity.
>
> Thank you very much for your valuable suggestions and comments on our manuscript. We hope that our responses have addressed your concerns. We would be most grateful if you could consider **raising your score** in light of these clarifications. Thank you once again for your time and consideration!

---

### Official Review · Reviewer_9pBE · 2023-08-05

**Soundness:** 4

**Excitement:**

3: Ambivalent: It has merits (e.g., it reports state-of-the-art results, the idea is nice), but there are key weaknesses (e.g., it describes incremental work), and it can significantly benefit from another round of revision. However, I won't object to accepting it if my co-reviewers champion it.

**Paper Topic And Main Contributions:**

This paper considers the problem of prompt optimization, which aims to optimize a given prompt to obtain better output from an LLM. The proposed method, MAPO, mainly consists of an SFT stage and an RL stage. The RL stage is composed of three main components: PPO, SFT approximation and maintaining generalization.

Experiments are conducted on various tasks and datasets. Significant gains are obtained by adding SFT; further gains are obtained by the proposed MAPO.

**Questions For The Authors:**

* Question A: In Line 198, there is "{$P_1, P_2, ..., P, P_{k-1}, P_k$}" and the authors stated that "Prompts to the left of P exhibit poorer inference results, while prompts to the right demonstrate better results.". I was wondering whether the number of better prompts is always exactly two, i.e., $P_{k-1}$ and $P_k$.
* Question B: What does it mean by putting a hat on "LLM"? In Line 203, the authors stated that it is the LLM utilized in the SFT process. So is it a SFT-LLM?
* Question C: In Line 413, the paragraph titled "The effect of PPO and RRHF" implies that RRHF, Ranking Responses from Human Feedback, is utilized in this paper. But I did not find clues about "human feedback" in this paper. I understand that the proposed method borrows an idea from RRHF, but either this title is not appropriate or I miss something.
* Question D: In Tables 4 and 5, what are the results for +SFT?

**Reasons To Accept:**

* This paper presents a complicated process to optimize the prompts for a specific LLM. Results show that the proposed MAPO achieves superior performance on some benchmarks.

**Reasons To Reject:**

* The proposed method seems to be highly computationally expensive, given the whole process of SFT and RL. It would be better if the authors could provide the computational cost of MAPO.
* It would be better if the authors could report the results of +SFT in Tables 3, 4, and 5. In Table 2, +SFT significantly outperforms the naive baselines, which raises the concern that whether the improvements in Tables 3, 4, and 5 are from +SFT or MAPO.
* In practice, people normally incorporate randomness (e.g., temperature) during the generation process of LLM. It would be better if such randomness could be considered in this paper.

**Reproducibility:**

4: Could mostly reproduce the results, but there may be some variation because of sample variance or minor variations in their interpretation of the protocol or method.

**Reviewer Confidence:**

3: Pretty sure, but there's a chance I missed something. Although I have a good feel for this area in general, I did not carefully check the paper's details, e.g., the math, experimental design, or novelty.

**Typos Grammar Style And Presentation Improvements:**

* Typos:
  * Line 103: "the same"
  * Line 104: "datasets"

* This paper is not quite easy to follow in its current state and requires further proofreading.
* The presentation of Section 3, as well as that of Figure 3, is a little messy.
* The font size in almost all tables and figures is too small.
* It would be better if the authors could report absolute improvement scores rather than relative improvement percentages. Or maybe report both of them.

---

> ### Author Rebuttal · Authors · 2023-08-29
>
> Thank you very much for your valuable suggestions and comments on our manuscript. Below are our responses to each of the points of criticism, questions, and typos you've raised.
>
>     R1: The proposed method seems to be highly computationally expensive, given the whole process of SFT and RL. It would be better if the authors could provide the computational cost of MAPO.
>
> Thanks for your question. While the training phase is computationally intensive, the generation phase is relatively lightweight. Specifically, once the prompt optimizing model MAPO is trained, the prompt generation process simply involves a feed forward propagation to generate the optimal prompt instead of further optimization through SFT and RL, thus significantly reducing the computational complexity.
>
>
> We list the computational complexity during the training and inference phase:
>
> 1.Training Phase
>
> 1.1 Warm-up Dataset Establishment
>
> 1.1.1 Generating Candidate Prompts: We use GPT-3.5 to generate 1000 candidate prompts for each original prompt.
>
> Time Complexity: **O(N * M)**, where N is the number of original prompts, and M is the number of candidates per prompt.
>
> Space Complexity: **O(N * M)**, for storing all generated candidate prompts.
>
> 1.1.2 Searching for Optimal Prompt: This involves comparison among candidate prompts, thus both time and space complexity are O(N*M).
>
> 1.2 Prompt Optimizer Construction
>
> 1.2.1 Supervised Fine-tuning (SFT):
>
> Time Complexity: **O(E * B * T)**, where E is the number of epochs, B is the batch size, and T is the number of model parameters.
>
> Space Complexity: **O(T)**, mainly for model parameters and gradients.
>
> 1.2.2 Building the Reward Model:
>
> Both time and space complexities are **O(N * M * log M)**, mainly for sorting and storing.
>
> 1.2.3 Reinforcement Learning (RL):
>
> Time Complexity: **O(E’ * B’ * T)**, where E' is the number of epochs in the RL stage, B' is the RL batch size, and T is the number of model parameters.
>
> Space Complexity: **O(T)**.
>
> **The total time complexity for the training phase is: O(N * M) + O(E * B * T) + O(N * M * log M) + O(E’ * B’ * T)**;
>
> **The total space complexity for the training phase is: O(N * M) + O(T) + O(N * M * log M) + O(T) = O(N * M * log M) + O(T)**
>
>
> 2.Inference Phase
>
> 2.1 Generating Optimized Prompt: Given an original prompt, an optimized prompt is generated through MAPO.
>
> Time Complexity: **O(T)**, as it's just a feed forward propagation.
>
> Space Complexity: **O(1)**, almost no extra space required other than for LLM and input/output storage.
>
> **The total time complexity for the inference phase is:O(T);**
>
> **The total space complexity for the inference phase is:O(1)**
>
>     R2: It would be better if the authors could report the results of +SFT in Tables 3, 4, and 5. In Table 2, +SFT significantly outperforms the naive baselines, which raises the concern that whether the improvements in Tables 3, 4, and 5 are from +SFT or MAPO.
>
>
>
> Thanks for your suggestion, we report the results of +SFT in Tables 3, 4, and 5. We also report both absolute improvement scores and relative improvement percentages. The blank section indicates that the content is the same as the cell above. Due to the limitations of Markdown syntax, merging cells is not possible.
> **We can observe that the performance improvements gained solely from using SFT are less than half of those achieved by our proposed MAPO method, both on similar tasks (Table 3) and general NLP tasks (Tables 4 and 5).** This clearly indicates the effectiveness of MAPO in optimizing model-adaptive prompts.
>
>
>
> Table 3: The few-shot performance of SFT, MAPO with SOTA prompt optimizing baselines in downstream tasks. F: Finetuning, C: Continous prompt, D: Discrete prompt.
> |   |              | SST-2 | YelpP. | MR   | CR   | RTE  | QNLI | SNLI | MNLI | MRPC |
> |---|--------------|-------|--------|------|------|------|------|------|------|------|
> | F | Finetuning   | 80.6  | 88.7   | 67.4 | 73.3 | 58.6 | 60.2 | 54.6 | 47.8 | 77.4 |
> | C | Softprompt   | 73.8  | 88.6   | 74.1 | 75.9 | 54.7 | 49.7 | 36.1 | 33.2 | 51.6 |
> |   | Black-Box    | 89.1  | 93.2   | 86.6 | 87.4 | 52.6 | 48.8 | 46.6 | 42.9 | 61.6 |
> |   | Autoprompt   | 75    | 79.8   | 62   | 57.5 | -    | -    | -    | -    | -    |
> | D | Manual       | 82.8  | 83     | 80.9 | 79.6 | 51.6 | 50.8 | 31.1 | 51.7 | 67.4 |
> |   | In-Context   | 85.9  | 89.6   | 80.6 | 85.5 | 60.4 | 53.8 | 47.1 | 53.4 | 45.8 |
> |   | Instructions | 89    | 84.4   | 85.2 | 80.8 | -    | -    | -    | -    | -    |
> |   | GrIPS        | 87.1  | 88.2   | 86.1 | 80   | -    | -    | -    | -    | -    |
> |   | RLprompt     | 92.5  | 95.1   | 87.1 | 89.5 | -    | -    | -    | -    | -    |
> |   | TEMPERA      | 91.9  | 92.6   | 88   | 91.1 | 60.3 | 57.4 | 56.4 | 45.2 | 74   |
> |   | AMA          | 95.7  | -      | -    | -    | 75.1 | -    | -    | -    | -    |
> | **D** | **SFT**         | **94.9**  | **92**     | **88.5** | **87.6** | **74.3** | **62.5** | **58.8** | **54.6** | **78.5** |
> | D | MAPO         | 96.1  | 93.5   | 90.2 | 88.9 | 75.3 | 63.1 | 60   | 55.7 | 79.3 |
>
> Table 4: Zero-shot domain transfer performance based on BLOOM and GPT-J with original, SFT-optimized and MAPO-Optimized prompts. CLS: Classification, M: MAPO. The (↑(%) ) and ↑(%) represent the increase degree of MAPO-optimized prompts compared with original prompts in each dataset and task, respectively (The same below).
> | Task   | Dataset    | BLOOM | **+SFT** |  (**↑**(%))| **↑**(%) | +MAPO | (↑(%) | ↑(%) | GPT-J | **+SFT** | (**↑**(%))| **↑**(%) | +MAPO | (↑(%)) | ↑(%) |
> |--------|------------|-------|------|--------|------|-------|-------|------|-------|------|--------|------|-------|--------|------|
> | Coref. | xwinograd  | 60.1  | **60.2** | **0.2**    | **0.2**  | 60.6  | 0.9   | 0.9  | -     | **-**    | **-**      | **-**    | -     | -      | -    |
> | NLU    | BoolQ      | 67.9  | **68**   | **0.1**    | **0.4**  | 68.2  | 0.4   | 0.9  | 67.2  | **67.4** | **0.3**    | **0.2**  | 67.9  | 1      | 0.5  |
> |        | CB         | 77.6  | **77.8** | **0.3**    |      | 78.1  | 0.6   |      | 83.9  | **84.1** | **0.2**    |      | 84.2  | 0.4    |      |
> |        | COPA       | 74    | **74.3** | **0.4**    |      | 75    | 1.4   |      | 84    | **84.2** | **0.2**    |      | 84.2  | 0.2    |      |
> |        | MultiRC    | 59.7  | **60.3** | **1**      |      | 60.4  | 1.2   |      | 63.8  | **63.9** | **0.2**    |      | 64.1  | 0.5    |      |
> |        | ReCoRD     | 69.8  | **70.1** | **0.4**    |      | 70.2  | 0.6   |      | 74.4  | **74.5** | **0.1**    |      | 74.7  | 0.4    |      |
> |        | WiC        | 61.4  | **61.6** | **0.3**    |      | 62    | 1     |      | 61    | **61.1** | **0.2**    |      | 61.3  | 0.5    |      |
> |        | WSC        | 64.4  | **64.7** | **0.5**    |      | 65.1  | 1.1   |      | 77.9  | **78**   | **0.1**    |      | 78.1  | 0.3    |      |
> | NLI    | ANLIR1     | 31.5  | **31.7** | **0.6**    | **0.6**  | 32.1  | 1.9   | 1.3  | 37.8  | **38**   | **0.5**    | **0.3**  | 38.2  | 1.1    | 0.7  |
> |        | ANLIR2     | 35.1  | **35.2** | **0.3**    |      | 35.4  | 0.9   |      | 37.9  | **38**   | **0.3**    |      | 38.3  | 1.1    |      |
> |        | ANLIR3     | 37.1  | **37.5** | **1.1**    |      | 37.8  | 1.9   |      | 40.9  | **41**   | **0.2**    |      | 41.1  | 0.5    |      |
> |        | StoryCloze | 79    | **79.2** | **0.3**    |      | 79.5  | 0.6   |      | 87.8  | **87.9** | **0.1**    |      | 87.9  | 0.1    |      |
> | CLS    | Amazon     | 65.2  | **66.4** | **1.8**    | **1.4**  | 67.7  | 3.8   | 3.3  | 68.2  | **68.7** | **0.7**    | **0.6**  | 69.4  | 1.8    | 1.6  |
> |        | DBPedia    | 70.5  | **71.2** | **1**      |      | 72.5  | 2.8   |      | 83.9  | **84.2** | **0.4**    |      | 85.1  | 1.4    |      |
> | QA     | DROP       | 67.9  | **68.2** | **0.4**    | **1.9**  | 69.9  | 2.9   | 5.6  | 51.6  | **51.9** | **0.6**    | **1.3**  | 52.8  | 2.3    | 3.2  |
> |        | NQ         | 15.1  | **15.4** | **2**      |      | 16.1  | 6.6   |      | 19.6  | **20.1** | **2.6**    |      | 20.8  | 6.1    |      |
> |        | RealTimeQA | 29    | **30.2** | **4.1**    |      | 31.5  | 8.6   |      | 36    | **36.5** | **1.4**    |      | 37.2  | 3.3    |      |
> |        | WebQs      | 34.8  | **35.1** | **0.9**    |      | 36.3  | 4.3   |      | 44.1  | **44.3** | **0.5**    |      | 44.6  | 1.1    |      |
> |        |            |       |      |        |      |       |       |      |       |      |        |      |       |        |      |
>
>
> Table 5: Zero-shot domain transfer performance based on LLaMA with original, SFT-optimized and MAPO-Optimized prompts. RS: Commonsense Reasoning.
> | Task | Dataset    | LLaMA | +SFT | (↑(%)) |↑(%) | +MAPO | (↑(%)) |↑(%) |
> |------|------------|-------|------|--------|------|-------|--------|------|
> | RS   | BoolQ      | 76.5  | **76.6** | **0.1**    | **0.1**  | 76.7  | 0.3    | 0.5  |
> |      | PIQA       | 79.8  | **79.9** | **0.1**    |      | 80    | 0.3    |      |
> |      | SIQA       | 48.9  | **48.9** | **0**      |      | 49    | 0.2    |      |
> |      | HellaSwag  | 76.1  | **76.2** | **0.1**    |      | 76.5  | 0.5    |      |
> |      | WinoGrande | 70.1  | **70.2** | **0.1**    |      | 70.5  | 0.6    |      |
> |      | ARC-e      | 72.8  | **72.9** | **0.1**    |      | 73.2  | 0.6    |      |
> |      | ARC-c      | 47.6  | **47.6** | **0**      |      | 47.8  | 0.4    |      |
> |      | OBQA       | 57.2  | **57.4** | **0.3**    |      | 57.9  | 1.2    |      |
> | QA   | NQ         | 16.8  | **17.2** | **2.4**    | **1.4**  | 18.1  | 7.7    | 4.7  |
> |      | RACE       | 50    | **50.2** | **0.4**    |      | 50.8  | 1.6    |      |
>
>
>     R3: In practice, people normally incorporate randomness (e.g., temperature) during the generation process of LLM. It would be better if such randomness could be considered in this paper.
>
> 1. The "temperature" parameter during the generation phase dictates the sharpness of the output distribution. A lower temperature value, closer to zero, tends to make the LLM opt for the most probable output, thereby reducing randomness. **Given that our prompts do not require high creativity, we have set a lower temperature range [0-0.5] for generation, within which we aim to generate optimal prompts.**
> 2. To further investigate the impact of varying temperatures on the generated output, **we conduct an additional set of experiments to assess the performance of the MAPO method under different randomness settings (temperature=0,0.2,0.5,0.8) as shown in Table 2,3,4,5.** We also report both absolute improvement scores and relative improvement percentages. The blank section indicates that the content is the same as the cell above. Due to the limitations of Markdown syntax, merging cells is not possible.
> 3. Each experiment group runs 5 times. **Our findings reveal that a high-temperature setting (t=0.8) tends to produce inferior prompts that lead to less accurate outputs for a specific task. Lower temperature (t=0.2) or greedy settings (t=0) are likely to produce more accurate outputs that are closer to our optimal results. This suggests that in a task like prompt optimization, introducing a stable (low temperature) but slight degree of variability (non-zero temperature) yields the best results.**
>
>
> Table 2: (↑) represents the absolute increase in performance for the LLM on the left when using MAPO-optimized prompts compared to Zero-shot. (↑(%)) indicates the relative increase in performance for the LLM on the left when using MAPO-optimized prompts compared to Zero-shot. The symbol ↑ denotes the average absolute increase in performance across all three LLMs with MAPO-optimized prompts compared to Zero-shot. ↑(%) signifies the average relative increase for the same across all three LLMs. All these increases are specific to the downstream tasks listed in the corresponding rows. CLS: classification, GEN: generation.
>
> | Task | Dataset | M |     |        |       |     |        |       |      |        |     |       |
> |------|---------|----------------|-----|--------|-------|-----|--------|-------|------|--------|-----|-------|
> |      |         | BLOOM          | (↑) | (↑(%)) | GPT-J | (↑) | (↑ (%)) | LLaMA | (↑)  | (↑ (%)) | ↑   | ↑ (%)  |
> | QA   | AdverQA | 19.5           | 6   | 44.4   | 11    | 8   | 266.7  | 25.1  | 21.9 | 684.4  | **12**  | **331.8** |
> |      | OpenQA  | 27.2           | 1.3 | 5      | 21    | 4   | 23.5   | 16.1  | 2.8  | 21.1   | **2.7** | **16.5**  |
> |      | CloseQA | 9.4            | 3   | 46.9   | 10.2  | 3.3 | 47.8   | 14.8  | 4    | 37     | **3.4** | **43.9**  |
> | CLS  | News    | 98.7           | 5.9 | 6.4    | 6.3   | 6.3 | -      | 12.4  | 11.3 | 1027.3 | **7.8** | **516.9** |
> |      | Movie   | 93.3           | 2.4 | 2.6    | 53.9  | 2.8 | 5.5    | 82.5  | 3.8  | 4.8    | **3**   | **4.3**   |
> |      | QASC    | 99.9           | 0.5 | 0.5    | 56.8  | 2.8 | 5.2    | 72.8  | 11.2 | 18.2   | **4.8** | **8**     |
> | GEN  | Topics  | 36.2           | 6.7 | 22.7   | 23.4  | 5.9 | 33.7   | 19.5  | 5.2  | 36.4   | **5.9** | **30.9**  |
> |      | Summary | 50.2           | 4.1 | 8.9    | 17.8  | 4.7 | 35.9   | 12.2  | 5.6  | 84.8   | **4.8** | **43.2**  |
> |      | Explan  | 8.9            | 3.2 | 56.1   | 12.9  | 4.4 | 51.8   | 9.1   | 2.2  | 31.9   | **3.3** | **46.6**  |
> Note: M means using MAPO with temperature [0,0.5] in the original manuscript.
>
>
> | Task | Dataset | M-0 |     |        |       |     |        |       |      |        |      |       |
> |------|---------|------------------|-----|--------|-------|-----|--------|-------|------|--------|------|-------|
> |      |         | BLOOM            | (↑) | (↑(%)) | GPT-J | (↑) | (↑(%)) | LLaMA | (↑)  | (↑(%)) | ↑    | ↑(%)  |
> | QA   | AdverQA | 19.3             | 5.8 | 43     | 10.5  | 7.5 | 250    | 24.8  | 21.6 | 675    | **11.6** | **322.7** |
> |      | OpenQA  | 26.8             | 0.9 | 3.5    | 20.9  | 3.9 | 22.9   | 15.2  | 1.9  | 14.3   | **2.2**  | **13.6**  |
> |      | CloseQA | 8.9              | 2.5 | 39.1   | 9.8   | 2.9 | 42     | 14    | 3.2  | 29.6   | **2.9**  | **36.9**  |
> | CLS  | News    | 96.5             | 3.7 | 4      | 5.9   | 5.9 | -      | 11.9  | 10.8 | 981.8  | **6.8**  | **492.9** |
> |      | Movie   | 92.8             | 1.9 | 2.1    | 53.6  | 2.5 | 4.9    | 82    | 3.3  | 4.2    | **2.6**  | **3.7**   |
> |      | QASC    | 99.9             | 0.5 | 0.5    | 56.6  | 2.6 | 4.8    | 72.2  | 10.6 | 17.2   | **4.6**  | **7.5**   |
> | GEN  | Topics  | 35.9             | 6.4 | 21.7   | 23    | 5.5 | 31.4   | 19    | 4.7  | 32.9   | **5.5**  | **28.7**  |
> |      | Summary | 49.7             | 3.6 | 7.8    | 17    | 3.9 | 29.8   | 11.9  | 5.3  | 80.3   | **4.3**  | **39.3**  |
> |      | Explan  | 7.8              | 2.1 | 36.8   | 12.3  | 3.8 | 44.7   | 8.8   | 1.9  | 27.5   | **2.6**  | **36.3**  |
> Note: M-0 means using MAPO with temperature 0.
>
> | Task | Dataset | M-0.2 |     |        |       |     |        |       |      |        |      |       |
> |------|---------|--------------------|-----|--------|-------|-----|--------|-------|------|--------|------|-------|
> |      |         | BLOOM              | (↑) | (↑(%)) | GPT-J | (↑) | (↑(%)) | LLaMA | (↑)  | (↑(%)) | ↑    | ↑(%)  |
> | QA   | AdverQA | 19                 | 5.5 | 40.7   | 10.7  | 7.7 | 256.7  | 24.4  | 21.2 | 662.5  | **11.5** | **320**   |
> |      | OpenQA  | 26.9               | 1   | 3.9    | 20.1  | 3.1 | 18.2   | 15.6  | 2.3  | 17.3   | **2.1**  | **13.1**  |
> |      | CloseQA | 8.4                | 2   | 31.3   | 10    | 3.1 | 44.9   | 14.2  | 3.4  | 31.5   | **2.8**  | **35.9**  |
> | CLS  | News    | 96.7               | 3.9 | 4.2    | 6.1   | 6.1 | -      | 12.1  | 11   | 1000   | **7**    | **502.1** |
> |      | Movie   | 92.9               | 2   | 2.2    | 53.2  | 2.1 | 4.1    | 81.5  | 2.8  | 3.6    | **2.3**  | **3.3**   |
> |      | QASC    | 99.9               | 0.5 | 0.5    | 56.6  | 2.6 | 4.8    | 72.3  | 10.7 | 17.4   | **4.6**  | **7.6**   |
> | GEN  | Topics  | 36                 | 6.5 | 22     | 23.3  | 5.8 | 33.1   | 19.3  | 5    | 35     | **5.8**  | **30**    |
> |      | Summary | 50.2               | 4.1 | 8.9    | 16.6  | 3.5 | 26.7   | 12    | 5.4  | 81.8   | **4.3**  | **39.1**  |
> |      | Explan  | 8.2                | 2.5 | 43.9   | 12.5  | 4   | 47.1   | 9     | 2.1  | 30.4   | **2.9**  | **40.5**  |
> Note: M-0.2 means using MAPO with temperature 0.2.
>
> | Task | Dataset | M-0.5 |     |        |       |     |        |       |      |        |      |       |
> |------|---------|--------------------|-----|--------|-------|-----|--------|-------|------|--------|------|-------|
> |      |         | BLOOM              | (↑) | (↑(%)) | GPT-J | (↑) | (↑(%)) | LLaMA | (↑)  | (↑(%)) | ↑    | ↑(%)  |
> | QA   | AdverQA | 18.8               | 5.3 | 39.3   | 10    | 7   | 233.3  | 23.6  | 20.4 | 637.5  | **10.9** | **303.4** |
> |      | OpenQA  | 27.1               | 1.2 | 4.6    | 20.4  | 3.4 | 20     | 15.8  | 2.5  | 18.8   | **2.4**  | **14.5**  |
> |      | CloseQA | 8.3                | 1.9 | 29.7   | 8.5   | 1.6 | 23.2   | 14.2  | 3.4  | 31.5   | **2.3**  | **28.1**  |
> | CLS  | News    | 96.2               | 3.4 | 3.7    | 5.7   | 5.7 | -      | 10.3  | 9.2  | 836.4  | **6.1**  | **420.1** |
> |      | Movie   | 93.2               | 2.3 | 2.5    | 53    | 1.9 | 3.7    | 81.6  | 2.9  | 3.7    | **2.4**  | **3.3**   |
> |      | QASC    | 99.9               | 0.5 | 0.5    | 56.5  | 2.5 | 4.6    | 70.6  | 9    | 14.6   | **4**    | **6.6**   |
> | GEN  | Topics  | 35.5               | 6   | 20.3   | 21.9  | 4.4 | 25.1   | 18.9  | 4.6  | 32.2   | **5**    | **25.9**  |
> |      | Summary | 49.2               | 3.1 | 6.7    | 17    | 3.9 | 29.8   | 11.1  | 4.5  | 68.2   | **3.8**  | **34.9**  |
> |      | Explan  | 7.1                | 1.4 | 24.6   | 11.1  | 2.6 | 30.6   | 8.5   | 1.6  | 23.2   | **1.9**  | **26.1**  |
> Note: M-0.5 means using MAPO with temperature 0.5.
>
> | Task | Dataset | M-0.8 |     |        |       |     |        |       |      |        |      |       |
> |------|---------|--------------------|-----|--------|-------|-----|--------|-------|------|--------|------|-------|
> |      |         | BLOOM              | (↑) | (↑(%)) | GPT-J | (↑) | (↑(%)) | LLaMA | (↑)  | (↑(%)) | ↑    | ↑(%)  |
> | QA   | AdverQA | 18.3               | 4.8 | 35.6   | 9.8   | 6.8 | 226.7  | 23.3  | 20.1 | 628.1  | **10.6** | **296.8** |
> |      | OpenQA  | 26.7               | 0.8 | 3.1    | 20.6  | 3.6 | 21.2   | 15.6  | 2.3  | 17.3   | **2.2**  | **13.9**  |
> |      | CloseQA | 7.9                | 1.5 | 23.4   | 8.6   | 1.7 | 24.6   | 14.1  | 3.3  | 30.6   | **2.2**  | **26.2**  |
> | CLS  | News    | 95.9               | 3.1 | 3.3    | 5.8   | 5.8 | -      | 10.2  | 9.1  | 827.3  | **6**    | **415.3** |
> |      | Movie   | 92.8               | 1.9 | 2.1    | 52.9  | 1.8 | 3.5    | 81.5  | 2.8  | 3.6    | **2.2**  | **3.1**   |
> |      | QASC    | 99.9               | 0.5 | 0.5    | 56.6  | 2.6 | 4.8    | 70.4  | 8.8  | 14.3   | **4**    | **6.5**   |
> | GEN  | Topics  | 35.1               | 5.6 | 19     | 21.9  | 4.4 | 25.1   | 18.7  | 4.4  | 30.8   | **4.8**  | **25**    |
> |      | Summary | 50.1               | 4   | 8.7    | 16.8  | 3.7 | 28.2   | 10.9  | 4.3  | 65.2   | **4**   | **34**    |
> |      | Explan  | 7.1                | 1.4 | 24.6   | 10.9  | 2.4 | 28.2   | 8.2   | 1.3  | 18.8   | **1.7**  | **23.9**  |
> Note: M-0.8 means using MAPO with temperature 0.8.
>
> Table 3: The few-shot performance of MAPO with SOTA prompt optimizing baselines in downstream tasks. F: Finetuning, C: Continous prompt, D: Discrete prompt. MAPO means using MAPO with temperature [0,0.5]. MAPO-0 means using MAPO with temperature 0.
>
> |          | SST-2 | Yelp P. | MR   | CR   | RTE  | QNLI | SNLI | MNLI | MRPC | Average |
> |----------|-------|---------|------|------|------|------|------|------|------|---------|
> | MAPO     | 96.1  | 93.5    | 90.2 | 88.9 | 75.3 | 63.1 | 60   | 55.7 | 79.3 | **78**      |
> | MAPO-0   | 95.8  | 93.1    | 90.2 | 88.2 | 74.9 | 62.9 | 59.7 | 55.1 | 78.9 | **77.6**    |
> | MAPO-0.2 | 95.9  | 93.3    | 90.3 | 88   | 75.2 | 63   | 59.8 | 54.8 | 78.3 | **77.6**    |
> | MAPO-0.5 | 95.2  | 92.2    | 88.9 | 87.9 | 74.8 | 62.8 | 59.2 | 55   | 78.8 | **77.2**    |
> | MAPO-0.8 | 95.3  | 92.3    | 88.7 | 87.9 | 74.5 | 62.7 | 59.2 | 54.9 | 78.7 | **77.1**    |
>
> Table 4,5: Zero-shot domain transfer performance based on BLOOM, GPT-J and LLaMA with original and MAPO-Optimized prompts. CLS: Classification, M: MAPO. The (↑ ) and ↑represent the absolute improvement scores of MAPO-optimized prompts compared with original prompts in each dataset and task, respectively. The (↑(%) ) and ↑(%) represent the relative improvement percentages of MAPO-optimized prompts compared with original prompts in each dataset and task, respectively.
>
> #### BLOOM
> | Task   | Dataset    | BLOOM | M    | (↑) | ↑   | (↑(%)) | ↑(%) |
> |--------|------------|-------|------|-----|-----|--------|------|
> | Coref. | xwinograd  | 60.1  | 60.6 | 0.5 | **0.5** | 0.9    | **0.9**  |
> | NLU    | BoolQ      | 67.9  | 68.2 | 0.3 | **0.6** | 0.4    | **0.9**  |
> |        | CB         | 77.6  | 78.1 | 0.5 |     | 0.6    |      |
> |        | COPA       | 74    | 75   | 1   |     | 1.4    |      |
> |        | MultiRC    | 59.7  | 60.4 | 0.7 |     | 1.2    |      |
> |        | ReCoRD     | 69.8  | 70.2 | 0.4 |     | 0.6    |      |
> |        | WiC        | 61.4  | 62   | 0.6 |     | 1      |      |
> |        | WSC        | 64.4  | 65.1 | 0.7 |     | 1.1    |      |
> | NLI    | ANLI R1    | 31.5  | 32.1 | 0.6 | **0.5** | 1.9    | **1.3**  |
> |        | ANLI R2    | 35.1  | 35.4 | 0.3 |     | 0.9    |      |
> |        | ANLI R3    | 37.1  | 37.8 | 0.7 |     | 1.9    |      |
> |        | StoryCloze | 79    | 79.5 | 0.5 |     | 0.6    |      |
> | CLS    | Amazon     | 65.2  | 67.7 | 2.5 | **2.3** | 3.8    | **3.3**  |
> |        | DBPedia    | 70.5  | 72.5 | 2   |     | 2.8    |      |
> | QA     | DROP       | 67.9  | 69.9 | 2   | **1.8** | 2.9    | **5.6**  |
> |        | NQ         | 15.1  | 16.1 | 1   |     | 6.6    |      |
> |        | RealTimeQA | 29    | 31.5 | 2.5 |     | 8.6    |      |
> |        | WebQs      | 34.8  | 36.3 | 1.5 |     | 4.3    |      |
> Note: M means using MAPO with temperature [0,0.5] in the original manuscript.
>
> | Task   | Dataset    | BLOOM | M-0  | (↑) | ↑   | (↑(%)) | ↑(%) |
> |--------|------------|-------|------|-----|-----|--------|------|
> | Coref. | xwinograd  | 60.1  | 60.5 | 0.4 | **0.4** | 0.7    | **0.7**  |
> | NLU    | BoolQ      | 67.9  | 68   | 0.1 | **0.5** | 0.1    | **0.7**  |
> |        | CB         | 77.6  | 78   | 0.4 |     | 0.5    |      |
> |        | COPA       | 74    | 74.8 | 0.8 |     | 1.1    |      |
> |        | MultiRC    | 59.7  | 60.4 | 0.7 |     | 1.2    |      |
> |        | ReCoRD     | 69.8  | 70.2 | 0.4 |     | 0.6    |      |
> |        | WiC        | 61.4  | 61.8 | 0.4 |     | 0.7    |      |
> |        | WSC        | 64.4  | 65   | 0.6 |     | 0.9    |      |
> | NLI    | ANLI R1    | 31.5  | 32   | 0.5 | **0.5** | 1.6    | **1.2**  |
> |        | ANLI R2    | 35.1  | 35.4 | 0.3 |     | 0.9    |      |
> |        | ANLI R3    | 37.1  | 37.8 | 0.7 |     | 1.9    |      |
> |        | StoryCloze | 79    | 79.4 | 0.4 |     | 0.5    |      |
> | CLS    | Amazon     | 65.2  | 67.3 | 2.1 | **1.8** | 3.2    | **2.7**  |
> |        | DBPedia    | 70.5  | 72   | 1.5 |     | 2.1    |      |
> | QA     | DROP       | 67.9  | 69.5 | 1.6 | **1.5** | 2.4    | **4.6**  |
> |        | NQ         | 15.1  | 15.8 | 0.7 |     | 4.6    |      |
> |        | RealTimeQA | 29    | 31.2 | 2.2 |     | 7.6    |      |
> |        | WebQs      | 34.8  | 36.1 | 1.3 |     | 3.7    |      |
> Note: M-0 means using MAPO with temperature 0.
>
> | Task   | Dataset    | BLOOM | M-0.2 | (↑) | ↑   | (↑(%)) | ↑(%) |
> |--------|------------|-------|-------|-----|-----|--------|------|
> | Coref. | xwinograd  | 60.1  | 60.6  | 0.5 | **0.5** | 0.8    | **0.8**  |
> | NLU    | BoolQ      | 67.9  | 68.1  | 0.2 | **0.5** | 0.3    | **0.7**  |
> |        | CB         | 77.6  | 78.1  | 0.5 |     | 0.6    |      |
> |        | COPA       | 74    | 74.6  | 0.6 |     | 0.8    |      |
> |        | MultiRC    | 59.7  | 60.3  | 0.6 |     | 1      |      |
> |        | ReCoRD     | 69.8  | 70.2  | 0.4 |     | 0.6    |      |
> |        | WiC        | 61.4  | 61.7  | 0.3 |     | 0.5    |      |
> |        | WSC        | 64.4  | 65.1  | 0.7 |     | 1.1    |      |
> | NLI    | ANLI R1    | 31.5  | 32.2  | 0.7 | **0.5** | 2.2    | **1.3**  |
> |        | ANLI R2    | 35.1  | 35.5  | 0.4 |     | 1.1    |      |
> |        | ANLI R3    | 37.1  | 37.6  | 0.5 |     | 1.3    |      |
> |        | StoryCloze | 79    | 79.5  | 0.5 |     | 0.6    |      |
> | CLS    | Amazon     | 65.2  | 67.1  | 1.9 | **1.6** | 2.9    | **2.4**  |
> |        | DBPedia    | 70.5  | 71.8  | 1.3 |     | 1.8    |      |
> | QA     | DROP       | 67.9  | 69.3  | 1.4 | **1.2** | 2.1    | **3.8**  |
> |        | NQ         | 15.1  | 15.6  | 0.5 |     | 3.3    |      |
> |        | RealTimeQA | 29    | 31    | 2   |     | 6.9    |      |
> |        | WebQs      | 34.8  | 35.8  | 1   |     | 2.9    |      |
> Note: M-0.2 means using MAPO with temperature 0.2.
>
> | Task   | Dataset    | BLOOM | M-0.5 | (↑) | ↑   | (↑(%)) | ↑(%) |
> |--------|------------|-------|-------|-----|-----|--------|------|
> | Coref. | xwinograd  | 60.1  | 60.4  | 0.3 | **0.3** | 0.5    | **0.5**  |
> | NLU    | BoolQ      | 67.9  | 68    | 0.1 | **0.3** | 0.1    | **0.5**  |
> |        | CB         | 77.6  | 77.8  | 0.2 |     | 0.3    |      |
> |        | COPA       | 74    | 74.4  | 0.4 |     | 0.5    |      |
> |        | MultiRC    | 59.7  | 60.3  | 0.6 |     | 1      |      |
> |        | ReCoRD     | 69.8  | 70.1  | 0.3 |     | 0.4    |      |
> |        | WiC        | 61.4  | 61.6  | 0.2 |     | 0.3    |      |
> |        | WSC        | 64.4  | 64.8  | 0.4 |     | 0.6    |      |
> | NLI    | ANLI R1    | 31.5  | 32    | 0.5 | **0.3** | 1.6    | **0.9**  |
> |        | ANLI R2    | 35.1  | 35.3  | 0.2 |     | 0.6    |      |
> |        | ANLI R3    | 37.1  | 37.5  | 0.4 |     | 1.1    |      |
> |        | StoryCloze | 79    | 79.3  | 0.3 |     | 0.4    |      |
> | CLS    | Amazon     | 65.2  | 66.7  | 1.5 | **1.2** | 2.3    | **1.8**  |
> |        | DBPedia    | 70.5  | 71.4  | 0.9 |     | 1.3    |      |
> | QA     | DROP       | 67.9  | 68.5  | 0.6 | **0.8** | 0.9    | **2.7**  |
> |        | NQ         | 15.1  | 15.5  | 0.4 |     | 2.6    |      |
> |        | RealTimeQA | 29    | 30.6  | 1.6 |     | 5.5    |      |
> |        | WebQs      | 34.8  | 35.4  | 0.6 |     | 1.7    |      |
> Note: M-0.5 means using MAPO with temperature 0.5.
>
> | Task   | Dataset    | BLOOM | M-0.8 | (↑) | ↑   | (↑(%)) | ↑(%) |
> |--------|------------|-------|-------|-----|-----|--------|------|
> | Coref. | xwinograd  | 60.1  | 60.2  | 0.1 | **0.1** | 0.2    | **0.2**  |
> | NLU    | BoolQ      | 67.9  | 68    | 0.1 | **0.3** | 0.1    | **0.4**  |
> |        | CB         | 77.6  | 77.8  | 0.2 |     | 0.3    |      |
> |        | COPA       | 74    | 74.3  | 0.3 |     | 0.4    |      |
> |        | MultiRC    | 59.7  | 60.3  | 0.6 |     | 1      |      |
> |        | ReCoRD     | 69.8  | 70.1  | 0.3 |     | 0.4    |      |
> |        | WiC        | 61.4  | 61.6  | 0.2 |     | 0.3    |      |
> |        | WSC        | 64.4  | 64.7  | 0.3 |     | 0.5    |      |
> | NLI    | ANLI R1    | 31.5  | 31.8  | 0.3 | **0.3** | 1      | **0.8**  |
> |        | ANLI R2    | 35.1  | 35.3  | 0.2 |     | 0.6    |      |
> |        | ANLI R3    | 37.1  | 37.5  | 0.4 |     | 1.1    |      |
> |        | StoryCloze | 79    | 79.3  | 0.3 |     | 0.4    |      |
> | CLS    | Amazon     | 65.2  | 66.5  | 1.3 | **1.1** | 2      | **1.6**  |
> |        | DBPedia    | 70.5  | 71.3  | 0.8 |     | 1.1    |      |
> | QA     | DROP       | 67.9  | 68.3  | 0.4 | **0.7** | 0.6    | **2.5**  |
> |        | NQ         | 15.1  | 15.5  | 0.4 |     | 2.6    |      |
> |        | RealTimeQA | 29    | 30.5  | 1.5 |     | 5.2    |      |
> |        | WebQs      | 34.8  | 35.3  | 0.5 |     | 1.4    |      |
> Note: M-0.8 means using MAPO with temperature 0.8.
>
> #### GPT-J
> | Task   | Dataset    | GPT-J | M    | (↑) | ↑   | (↑(%)) | ↑(%) |
> |--------|------------|-------|------|-----|-----|--------|------|
> | Coref. | xwinograd  | -     | -    | -   | -   | -      | -    |
> | NLU    | BoolQ      | 67.2  | 67.9 | 0.7 | 0.3 | 1      | **0.5**  |
> |        | CB         | 83.9  | 84.2 | 0.3 |     | 0.4    |      |
> |        | COPA       | 84    | 84.2 | 0.2 |     | 0.2    |      |
> |        | MultiRC    | 63.8  | 64.1 | 0.3 |     | 0.5    |      |
> |        | ReCoRD     | 74.4  | 74.7 | 0.3 |     | 0.4    |      |
> |        | WiC        | 61    | 61.3 | 0.3 |     | 0.5    |      |
> |        | WSC        | 77.9  | 78.1 | 0.2 |     | 0.3    |      |
> | NLI    | ANLI R1    | 37.8  | 38.2 | 0.4 | 0.3 | 1.1    | **0.7**  |
> |        | ANLI R2    | 37.9  | 38.3 | 0.4 |     | 1.1    |      |
> |        | ANLI R3    | 40.9  | 41.1 | 0.2 |     | 0.5    |      |
> |        | StoryCloze | 87.8  | 87.9 | 0.1 |     | 0.1    |      |
> | CLS    | Amazon     | 68.2  | 69.4 | 1.2 | 1.2 | 1.8    | **1.6**  |
> |        | DBPedia    | 83.9  | 85.1 | 1.2 |     | 1.4    |      |
> | QA     | DROP       | 51.6  | 52.8 | 1.2 | 1   | 2.3    | **3.2**  |
> |        | NQ         | 19.6  | 20.8 | 1.2 |     | 6.1    |      |
> |        | RealTimeQA | 36    | 37.2 | 1.2 |     | 3.3    |      |
> |        | WebQs      | 44.1  | 44.6 | 0.5 |     | 1.1    |      |
> Note: M means using MAPO with temperature [0,0.5] in the original manuscript.
>
> | Task   | Dataset    | GPT-J | M-0  | (↑) | ↑   | (↑(%)) | ↑(%) |
> |--------|------------|-------|------|-----|-----|--------|------|
> | Coref. | xwinograd  | -     | -    | -   | -   | -      | -    |
> | NLU    | BoolQ      | 67.2  | 67.8 | 0.6 | 0.3 | 0.9    | **0.4**  |
> |        | CB         | 83.9  | 84.1 | 0.2 |     | 0.2    |      |
> |        | COPA       | 84    | 84.2 | 0.2 |     | 0.2    |      |
> |        | MultiRC    | 63.8  | 64.1 | 0.3 |     | 0.5    |      |
> |        | ReCoRD     | 74.4  | 74.7 | 0.3 |     | 0.4    |      |
> |        | WiC        | 61    | 61.3 | 0.3 |     | 0.5    |      |
> |        | WSC        | 77.9  | 78.1 | 0.2 |     | 0.3    |      |
> | NLI    | ANLI R1    | 37.8  | 38.2 | 0.4 | 0.2 | 1.1    | **0.6**  |
> |        | ANLI R2    | 37.9  | 38.1 | 0.2 |     | 0.5    |      |
> |        | ANLI R3    | 40.9  | 41.1 | 0.2 |     | 0.5    |      |
> |        | StoryCloze | 87.8  | 87.9 | 0.1 |     | 0.1    |      |
> | CLS    | Amazon     | 68.2  | 69   | 0.8 | 1   | 1.2    | **1.3**  |
> |        | DBPedia    | 83.9  | 85.1 | 1.2 |     | 1.4    |      |
> | QA     | DROP       | 51.6  | 52.3 | 0.7 | 0.7 | 1.4    | **2.3**  |
> |        | NQ         | 19.6  | 20.4 | 0.8 |     | 4.1    |      |
> |        | RealTimeQA | 36    | 37   | 1   |     | 2.8    |      |
> |        | WebQs      | 44.1  | 44.5 | 0.4 |     | 0.9    |      |
> Note: M-0 means using MAPO with temperature 0.
>
> | Task   | Dataset    | GPT-J | M-0.2 | (↑) | ↑   | (↑(%)) | ↑(%) |
> |--------|------------|-------|-------|-----|-----|--------|------|
> | Coref. | xwinograd  | -     | -     | -   | -   | -      | -    |
> | NLU    | BoolQ      | 67.2  | 67.9  | 0.7 | 0.3 | 1      | **0.4**  |
> |        | CB         | 83.9  | 84.2  | 0.3 |     | 0.4    |      |
> |        | COPA       | 84    | 84.2  | 0.2 |     | 0.2    |      |
> |        | MultiRC    | 63.8  | 64    | 0.2 |     | 0.3    |      |
> |        | ReCoRD     | 74.4  | 74.6  | 0.2 |     | 0.3    |      |
> |        | WiC        | 61    | 61.3  | 0.3 |     | 0.5    |      |
> |        | WSC        | 77.9  | 78.1  | 0.2 |     | 0.3    |      |
> | NLI    | ANLI R1    | 37.8  | 38.2  | 0.4 | 0.3 | 1.1    | **0.7**  |
> |        | ANLI R2    | 37.9  | 38.3  | 0.4 |     | 1.1    |      |
> |        | ANLI R3    | 40.9  | 41.1  | 0.2 |     | 0.5    |      |
> |        | StoryCloze | 87.8  | 87.9  | 0.1 |     | 0.1    |      |
> | CLS    | Amazon     | 68.2  | 69.1  | 0.9 | 1   | 1.3    | **1.3**  |
> |        | DBPedia    | 83.9  | 85    | 1.1 |     | 1.3    |      |
> | QA     | DROP       | 51.6  | 52.6  | 1   | 0.7 | 1.9    | **2.1**  |
> |        | NQ         | 19.6  | 20.3  | 0.7 |     | 3.6    |      |
> |        | RealTimeQA | 36    | 36.7  | 0.7 |     | 1.9    |      |
> |        | WebQs      | 44.1  | 44.5  | 0.4 |     | 0.9    |      |
> Note: M-0.2 means using MAPO with temperature 0.2.
>
>
> | Task   | Dataset    | GPT-J | M-0.5 | (↑) | ↑   | (↑(%)) | ↑(%) |
> |--------|------------|-------|-------|-----|-----|--------|------|
> | Coref. | xwinograd  | -     | -     | -   | -   | -      | -    |
> | NLU    | BoolQ      | 67.2  | 67.6  | 0.4 | 0.2 | 0.6    | **0.2**  |
> |        | CB         | 83.9  | 84.1  | 0.2 |     | 0.2    |      |
> |        | COPA       | 84    | 84.2  | 0.2 |     | 0.2    |      |
> |        | MultiRC    | 63.8  | 63.9  | 0.1 |     | 0.2    |      |
> |        | ReCoRD     | 74.4  | 74.5  | 0.1 |     | 0.1    |      |
> |        | WiC        | 61    | 61.2  | 0.2 |     | 0.3    |      |
> |        | WSC        | 77.9  | 78    | 0.1 |     | 0.1    |      |
> | NLI    | ANLI R1    | 37.8  | 38.2  | 0.4 | 0.2 | 1.1    | **0.5**  |
> |        | ANLI R2    | 37.9  | 38.1  | 0.2 |     | 0.5    |      |
> |        | ANLI R3    | 40.9  | 41    | 0.1 |     | 0.2    |      |
> |        | StoryCloze | 87.8  | 87.9  | 0.1 |     | 0.1    |      |
> | CLS    | Amazon     | 68.2  | 68.9  | 0.7 | 0.5 | 1      | **0.8**  |
> |        | DBPedia    | 83.9  | 84.3  | 0.4 |     | 0.5    |      |
> | QA     | DROP       | 51.6  | 52.2  | 0.6 | 0.5 | 1.2    | **1.7**  |
> |        | NQ         | 19.6  | 20.2  | 0.6 |     | 3.1    |      |
> |        | RealTimeQA | 36    | 36.6  | 0.6 |     | 1.7    |      |
> |        | WebQs      | 44.1  | 44.4  | 0.3 |     | 0.7    |      |
> Note: M-0.5 means using MAPO with temperature 0.5.
>
>
> | Task   | Dataset    | GPT-J | M-0.8 | (↑) | ↑   | (↑(%)) | ↑(%) |
> |--------|------------|-------|-------|-----|-----|--------|------|
> | Coref. | xwinograd  | -     | -     | -   | -   | -      | -    |
> | NLU    | BoolQ      | 67.2  | 67.5  | 0.3 | 0.2 | 0.4    | **0.2**  |
> |        | CB         | 83.9  | 84.1  | 0.2 |     | 0.2    |      |
> |        | COPA       | 84    | 84.2  | 0.2 |     | 0.2    |      |
> |        | MultiRC    | 63.8  | 63.9  | 0.1 |     | 0.2    |      |
> |        | ReCoRD     | 74.4  | 74.5  | 0.1 |     | 0.1    |      |
> |        | WiC        | 61    | 61.2  | 0.2 |     | 0.3    |      |
> |        | WSC        | 77.9  | 78    | 0.1 |     | 0.1    |      |
> | NLI    | ANLI R1    | 37.8  | 38.1  | 0.3 | 0.2 | 0.8    | **0.4**  |
> |        | ANLI R2    | 37.9  | 38    | 0.1 |     | 0.3    |      |
> |        | ANLI R3    | 40.9  | 41    | 0.1 |     | 0.2    |      |
> |        | StoryCloze | 87.8  | 87.9  | 0.1 |     | 0.1    |      |
> | CLS    | Amazon     | 68.2  | 68.8  | 0.6 | 0.4 | 0.9    | **0.7**  |
> |        | DBPedia    | 83.9  | 84.2  | 0.3 |     | 0.4    |      |
> | QA     | DROP       | 51.6  | 52.1  | 0.5 | 0.4 | 1      | **1.5**  |
> |        | NQ         | 19.6  | 20.1  | 0.5 |     | 2.6    |      |
> |        | RealTimeQA | 36    | 36.6  | 0.6 |     | 1.7    |      |
> |        | WebQs      | 44.1  | 44.3  | 0.2 |     | 0.5    |      |
> Note: M-0.8 means using MAPO with temperature 0.8.
>
>
> #### LLaMA
> | Task | Dataset    | LLaMA | M    | (↑) | ↑   | (↑(%)) | ↑(%) |
> |------|------------|-------|------|-----|-----|--------|------|
> | RS   | BoolQ      | 76.5  | 76.7 | 0.2 | 0.3 | 0.3    | **0.5**  |
> |      | PIQA       | 79.8  | 80   | 0.2 |     | 0.3    |      |
> |      | SIQA       | 48.9  | 49   | 0.1 |     | 0.2    |      |
> |      | HellaSwag  | 76.1  | 76.5 | 0.4 |     | 0.5    |      |
> |      | WinoGrande | 70.1  | 70.5 | 0.4 |     | 0.6    |      |
> |      | ARC-e      | 72.8  | 73.2 | 0.4 |     | 0.5    |      |
> |      | ARC-c      | 47.6  | 47.8 | 0.2 |     | 0.4    |      |
> |      | OBQA       | 57.2  | 57.9 | 0.7 |     | 1.2    |      |
> | QA   | NQ         | 16.8  | 18.1 | 1.3 | 1.1 | 7.7    | **4.7**  |
> |      | RACE       | 50    | 50.8 | 0.8 |     | 1.6    |      |
> Note: M means using MAPO with temperature [0,0.5] in the original manuscript.
>
>
> | Task | Dataset    | LLaMA | M-0  | (↑) | ↑   | (↑(%)) | ↑(%) |
> |------|------------|-------|------|-----|-----|--------|------|
> | RS   | BoolQ      | 76.5  | 76.7 | 0.2 | 0.3 | 0.3    | **0.4**  |
> |      | PIQA       | 79.8  | 80   | 0.2 |     | 0.3    |      |
> |      | SIQA       | 48.9  | 49   | 0.1 |     | 0.2    |      |
> |      | HellaSwag  | 76.1  | 76.4 | 0.3 |     | 0.4    |      |
> |      | WinoGrande | 70.1  | 70.4 | 0.3 |     | 0.4    |      |
> |      | ARC-e      | 72.8  | 73.2 | 0.4 |     | 0.5    |      |
> |      | ARC-c      | 47.6  | 47.8 | 0.2 |     | 0.4    |      |
> |      | OBQA       | 57.2  | 57.8 | 0.6 |     | 1      |      |
> | QA   | NQ         | 16.8  | 17.8 | 1   | 0.8 | 6      | **3.6**  |
> |      | RACE       | 50    | 50.6 | 0.6 |     | 1.2    |      |
> Note: M-0 means using MAPO with temperature 0.
>
>
> | Task | Dataset    | LLaMA | M-0.2 | (↑) | ↑   | (↑(%)) | ↑(%) |
> |------|------------|-------|-------|-----|-----|--------|------|
> | RS   | BoolQ      | 76.5  | 76.7  | 0.2 | 0.3 | 0.3    | **0.4**  |
> |      | PIQA       | 79.8  | 80    | 0.2 |     | 0.3    |      |
> |      | SIQA       | 48.9  | 49    | 0.1 |     | 0.2    |      |
> |      | HellaSwag  | 76.1  | 76.4  | 0.3 |     | 0.4    |      |
> |      | WinoGrande | 70.1  | 70.4  | 0.3 |     | 0.4    |      |
> |      | ARC-e      | 72.8  | 73.1  | 0.3 |     | 0.4    |      |
> |      | ARC-c      | 47.6  | 47.8  | 0.2 |     | 0.4    |      |
> |      | OBQA       | 57.2  | 57.7  | 0.5 |     | 0.9    |      |
> | QA   | NQ         | 16.8  | 18    | 1.2 | 1   | 7.1    | **4.3**  |
> |      | RACE       | 50    | 50.7  | 0.7 |     | 1.4    |      |
> Note: M-0.2 means using MAPO with temperature 0.2.
>
>
> | Task | Dataset    | LLaMA | M-0.5 | (↑) | ↑   | (↑(%)) | ↑(%) |
> |------|------------|-------|-------|-----|-----|--------|------|
> | RS   | BoolQ      | 76.5  | 76.7  | 0.2 | 0.2 | 0.3    | **0.3**  |
> |      | PIQA       | 79.8  | 79.9  | 0.1 |     | 0.1    |      |
> |      | SIQA       | 48.9  | 49    | 0.1 |     | 0.2    |      |
> |      | HellaSwag  | 76.1  | 76.4  | 0.3 |     | 0.4    |      |
> |      | WinoGrande | 70.1  | 70.4  | 0.3 |     | 0.4    |      |
> |      | ARC-e      | 72.8  | 73    | 0.2 |     | 0.3    |      |
> |      | ARC-c      | 47.6  | 47.7  | 0.1 |     | 0.2    |      |
> |      | OBQA       | 57.2  | 57.6  | 0.4 |     | 0.7    |      |
> | QA   | NQ         | 16.8  | 17.7  | 0.9 | 0.7 | 5.4    | **3.2**  |
> |      | RACE       | 50    | 50.5  | 0.5 |     | 1      |      |
> Note: M-0.5 means using MAPO with temperature 0.5.
>
>
> | Task | Dataset    | LLaMA | M-0.8 | (↑) | ↑   | (↑(%)) | ↑(%) |
> |------|------------|-------|-------|-----|-----|--------|------|
> | RS   | BoolQ      | 76.5  | 76.6  | 0.1 | 0.2 | 0.1    | **0.2**  |
> |      | PIQA       | 79.8  | 79.9  | 0.1 |     | 0.1    |      |
> |      | SIQA       | 48.9  | 49    | 0.1 |     | 0.2    |      |
> |      | HellaSwag  | 76.1  | 76.3  | 0.2 |     | 0.3    |      |
> |      | WinoGrande | 70.1  | 70.4  | 0.3 |     | 0.4    |      |
> |      | ARC-e      | 72.8  | 73    | 0.2 |     | 0.3    |      |
> |      | ARC-c      | 47.6  | 47.6  | 0   |     | 0      |      |
> |      | OBQA       | 57.2  | 57.5  | 0.3 |     | 0.5    |      |
> | QA   | NQ         | 16.8  | 17.6  | 0.8 | 0.5 | 4.8    | **2.7**  |
> |      | RACE       | 50    | 50.3  | 0.3 |     | 0.6    |      |
> Note: M-0.8 means using MAPO with temperature 0.8.
>
>
>     Question A: In Line 198, there is "{P_1,P_2,...,P,P_{K-1},P_{K}}" and the authors stated that "Prompts to the left of P exhibit poorer inference results, while prompts to the right demonstrate better results.". I was wondering whether the number of better prompts is always exactly two, i.e.,P_{k-1} and P_k.
>
> Thanks for your question. **No, the number of better prompts is not always exactly two** (i.e., $P_{k-1}$ and $P_k$) based on the method described in the paper. The sequence $\{P_1, P_2, \ldots, P, P_{k-1}, P_k\}$ is an example of a ranking sequence, but the number of prompts that perform better than the original prompt ($P$) could vary. The paper states that they generate 1,000 candidate prompts per original prompt, making a total of 1,001 prompts including the original one. These are then ranked based on their performance in the downstream task. The number of prompts that are better than the original prompt ($P$) could range from 0 to 1,000, depending on their actual performance. Therefore, prompts to the right of the original prompt ($P$) in the ranking sequence would demonstrate better results, but the number of such prompts is not specified to be exactly two.
>
>
>     Question B: What does it mean by putting a hat on "LLM"? In Line 203, the authors stated that it is the LLM utilized in the SFT process. So is it a SFT-LLM?
>
> Thanks for your question. The hat symbol (^) on "LLM" in the paper indicates a specific LLM that has been fine-tuned through the SFT process. Because different LLMs might require different optimized prompts, the hat on "LLM" is also used to signify a specific LLM whose prompts need to be optimized.
> **For example, if the paper is studying how to find an optimal prompt for an LLM like "BLOOM," then the ^LLM in that context would be the BLOOM model that has undergone the SFT process.**
>
>
>
>
>
>     Question C: In Line 413, the paragraph titled "The effect of PPO and RRHF" implies that RRHF, Ranking Responses from Human Feedback, is utilized in this paper. But I did not find clues about "human feedback" in this paper. I understand that the proposed method borrows an idea from RRHF, but either this title is not appropriate or I miss something.
>
> 1. Thank you for bringing up your concern regarding the use of the term "RRHF" (Ranking Responses from Human Feedback) in Line 413 under the section "The effect of PPO and RRHF." We apologize for the confusion. You are correct that the title may be misleading, as we do not utilize human feedback in our framework.
>
> 2. In our work, **we have indeed borrowed the idea from RRHF but adapted it to focus on "model feedback" instead of "human feedback."** Our reward model ranks prompts based on a specific LLM's preferences rather than human preferences. This is detailed in the section discussing the construction of the reward model.
>
> 3. To better align the title with the actual content of our paper, we have changed the title from "The effect of PPO and RRHF" to "The effect of PPO and **RRMF" (Ranking Responses from Model Feedback).**
>
> We hope this clears up any misunderstandings, and we appreciate your attention to detail.
>
>
>     Question D: In Tables 4 and 5, what are the results for +SFT?
>
> See responses in above-mentioned R2.
>
>
>
>
>     Typos: Line 103: "the same". Line 104: "datasets"
>
> Thanks for your suggestion. We will revise it in the final version.
>
>
>     This paper is not quite easy to follow in its current state and requires further proofreading.
>
> Thanks for your suggestion. We will proofread it in the final version.
>
>
>     The presentation of Section 3, as well as that of Figure 3, is a little messy.
>
> Thanks for your suggestion. We have modified the Figure 3 in https://p.sda1.dev/12/35aebb7ebe7acb6a433e25c2954ebbce/framework.png, and we will revise it in the final version.
>
>
>
>     The font size in almost all tables and figures is too small.
>
> Thanks for your suggestion. We will enlarge them in the final version.
>
>
>     It would be better if the authors could report absolute improvement scores rather than relative improvement percentages. Or maybe report both of them.
>
> Thanks for your suggestion. We report both absolute improvement scores and relative improvement percentages in Table 2,3,4,5. The blank section indicates that the content is the same as the cell above. Due to the limitations of Markdown syntax, merging cells is not possible.
>
>
> Table 2: (↑) under SFT-optimized represents the absolute increase in performance for the LLM on the left when using SFT-optimized prompts compared to Zero-shot. (↑(%)) under SFT-optimized indicates the relative increase in performance for the same. The symbol ↑ under SFT-optimized denotes the average absolute increase in performance across all three LLMs with SFT-optimized prompts compared to Zero-shot. ↑(%) under SFT-optimized signifies the average relative increase for the same across all three LLMs. (↑) under MAPO-optimized represents the absolute increase in performance for the LLM on the left when using MAPO-optimized prompts compared to Zero-shot. (↑(%)) under MAPO-optimized indicates the relative increase in performance for the same. The symbol ↑ under MAPO-optimized denotes the average absolute increase in performance across all three LLMs with MAPO-optimized prompts compared to Zero-shot. ↑(%) under MAPO-optimized signifies the average relative increase for the same across all three LLMs. All these increases are specific to the downstream tasks listed in the corresponding rows. CLS: classification, GEN: generation.
> | Task | Dataset | Original |       |       | +SFT  |         |            |       |         |            |       |         |            |          |           | +MAPO |         |            |       |         |            |       |          |            |         |           |
> |------|---------|----------|-------|-------|-------|---------|------------|-------|---------|------------|-------|---------|------------|----------|-----------|-------|---------|------------|-------|---------|------------|-------|----------|------------|---------|-----------|
> |      |         | BLOOM    | GPT-J | LLaMA | BLOOM | **(↑)** | **(↑(%))** | GPT-J | **(↑)** | **(↑(%))** | LLaMA | **(↑)** | **(↑(%))** | **↑**    | **↑(%)**  | BLOOM | **(↑)** | **(↑(%))** | GPT-J | **(↑)** | **(↑(%))** | LLaMA | **(↑)**  | **(↑(%))** | **↑**   | **↑(%)**  |
> | QA   | AdverQA | 13.5     | 3     | 3.2   | 18.3  | **4.8** | **35.6**   | 9.4   | **6.4** | **213.3**  | 23.2  | **20**  | **625**    | **10.4** | **291.3** | 19.5  | **6**   | **44.4**   | 11    | **8**   | **266.7**  | 25.1  | **21.9** | **684.4**  | **12**  | **331.8** |
> |      | OpenQA  | 25.9     | 17    | 13.3  | 26.7  | **0.8** | **3.1**    | 20.3  | **3.3** | **19.4**   | 15.4  | **2.1** | **15.8**   | **2.1**  | **12.8**  | 27.2  | **1.3** | **5**      | 21    | **4**   | **23.5**   | 16.1  | **2.8**  | **21.1**   | **2.7** | **16.5**  |
> |      | CloseQA | 6.4      | 6.9   | 10.8  | 7.8   | **1.4** | **21.9**   | 8.4   | **1.5** | **21.7**   | 13.9  | **3.1** | **28.7**   | **2**    | **24.1**  | 9.4   | **3**   | **46.9**   | 10.2  | **3.3** | **47.8**   | 14.8  | **4**    | **37**     | **3.4** | **43.9**  |
> | CLS  | News    | 92.8     | 0     | 1.1   | 95.5  | **2.7** | **2.9**    | 5.5   | **5.5** | **-**      | 10.1  | **9**   | **818.2**  | **5.7**  | **410.6** | 98.7  | **5.9** | **6.4**    | 6.3   | **6.3** | **-**      | 12.4  | **11.3** | **1027.3** | **7.8** | **516.9** |
> |      | Movie   | 90.9     | 51.1  | 78.7  | 92.6  | **1.7** | **1.9**    | 52.7  | **1.6** | **3.1**    | 81.3  | **2.6** | **3.3**    | **2**    | **2.8**   | 93.3  | **2.4** | **2.6**    | 53.9  | **2.8** | **5.5**    | 82.5  | **3.8**  | **4.8**    | **3**   | **4.3**   |
> |      | QASC    | 99.4     | 54    | 61.6  | 99.9  | **0.5** | **0.5**    | 56.3  | **2.3** | **4.3**    | 70.2  | **8.6** | **14**     | **3.8**  | **6.3**   | 99.9  | **0.5** | **0.5**    | 56.8  | **2.8** | **5.2**    | 72.8  | **11.2** | **18.2**   | **4.8** | **8**     |
> | GEN  | Topics  | 29.5     | 17.5  | 14.3  | 34.8  | **5.3** | **18**     | 21.6  | **4.1** | **23.4**   | 18.6  | **4.3** | **30.1**   | **4.6**  | **23.8**  | 36.2  | **6.7** | **22.7**   | 23.4  | **5.9** | **33.7**   | 19.5  | **5.2**  | **36.4**   | **5.9** | **30.9**  |
> |      | Summary | 46.1     | 13.1  | 6.6   | 48.8  | **2.7** | **5.9**    | 16.7  | **3.6** | **27.5**   | 10.7  | **4.1** | **62.1**   | **3.5**  | **31.8**  | 50.2  | **4.1** | **8.9**    | 17.8  | **4.7** | **35.9**   | 12.2  | **5.6**  | **84.8**   | **4.8** | **43.2**  |
> |      | Explan  | 5.7      | 8.5   | 6.9   | 6.8   | **1.1** | **19.3**   | 10.7  | **2.2** | **25.9**   | 8.2   | **1.3** | **18.8**   | **1.5**  | **21.3**  | 8.9   | **3.2** | **56.1**   | 12.9  | **4.4** | **51.8**   | 9.1   | **2.2**  | **31.9**   | **3.3** | **46.6**  |
>
> Table 3: The few-shot performance of MAPO with SOTA prompt optimizing baselines in downstream tasks. The metrics "Average-1," "(↑)-1," and "(↑(%))-1" are intended for comparisons that include baseline results across all datasets, such as those for Finetuning. Conversely, "Average-2," "(↑)-2," and "(↑(%))-2" are specifically designed for comparisons involving only the SST-2, Yelp P., MR, and CR datasets, such as those for Autoprompt. In these metrics, "(↑)" represents the absolute increase, while "(↑(%))" indicates the relative percentage increase.
> |   |              | SST-2 | YelpP. | MR   | CR   | RTE  | QNLI | SNLI | MNLI | MRPC | Average-1 | Average-2 | (↑)-1 | (↑(%))-1 | (↑)-2   | (↑(%))-2 |
> |---|--------------|-------|--------|------|------|------|------|------|------|------|-----------|-----------|-------|----------|---------|----------|
> | F | Finetuning   | 80.6  | 88.7   | 67.4 | 73.3 | 58.6 | 60.2 | 54.6 | 47.8 | 77.4 | 67.6      | -         | -     | -        | -       | -        |
> | C | Softprompt   | 73.8  | 88.6   | 74.1 | 75.9 | 54.7 | 49.7 | 36.1 | 33.2 | 51.6 | 59.7      | -         | -     | -        | -       | -        |
> |   | Black-Box    | 89.1  | 93.2   | 86.6 | 87.4 | 52.6 | 48.8 | 46.6 | 42.9 | 61.6 | 67.6      | -         | -     | -        | -       | -        |
> |   | Autoprompt   | 75.0  | 79.8   | 62.0 | 57.5 | -    | -    | -    | -    | -    | -         | 68.6      |       |          |         |          |
> | D | Manual       | 82.8  | 83.0   | 80.9 | 79.6 | 51.6 | 50.8 | 31.1 | 51.7 | 67.4 | 64.3      | -         | -     | -        | -       | -        |
> |   | In-Context   | 85.9  | 89.6   | 80.6 | 85.5 | 60.4 | 53.8 | 47.1 | 53.4 | 45.8 | 66.9      | -         | -     | -        | -       | -        |
> |   | Instructions | 89.0  | 84.4   | 85.2 | 80.8 | -    | -    | -    | -    | -    | -         | 84.9      | -     | -        | -       | -        |
> |   | GrIPS        | 87.1  | 88.2   | 86.1 | 80.0 | -    | -    | -    | -    | -    | -         | 85.4      | -     | -        | -       | -        |
> |   | RLprompt     | 92.5  | 95.1   | 87.1 | 89.5 | -    | -    | -    | -    | -    | -         | **91.1**  | -     | -        | -       | -        |
> |   | TEMPERA      | 91.9  | 92.6   | 88.0 | 91.1 | 60.3 | 57.4 | 56.4 | 45.2 | 74.0 | **73**    | -         | -     | -        | -       | -        |
> |   | AMA          | 95.7  | -      | -    | -    | 75.1 | -    | -    | -    | -    | -         | -         | -     | -        | -       | -        |
> | D | MAPO         | 96.1  | 93.5   | 90.2 | 88.9 | 75.3 | 63.1 | 60.0 | 55.7 | 79.3 | **78**    | **92.2**  | **5** | **6.8**  | **1.1** | **1.2**  |
>
>
> Table 4: Zero-shot domain transfer performance based on BLOOM and GPT-J with original and MAPO-Optimized prompts. CLS: Classification. The (↑ ) and ↑represent the absolute improvement scores of MAPO-optimized prompts compared with original prompts in each dataset and task, respectively. The (↑(%) ) and ↑(%) represent the relative improvement percentages of MAPO-optimized prompts compared with original prompts in each dataset and task, respectively.
> | Task   | Dataset    | BLOOM | +MAPO | **(↑)** | **↑**   | **(↑(%))** | **↑(%)** | GPT-J | +MAPO | **(↑)** | **↑**   | **(↑(%))** | **↑(%)** |
> |--------|------------|-------|-------|---------|---------|------------|----------|-------|-------|---------|---------|------------|----------|
> | Coref. | xwinograd  | 60.1  | 60.6  | **0.5** | **0.5** | **0.9**    | **0.9**  | -     | -     | **-**   | **-**   | **-**      | **-**    |
> | NLU    | BoolQ      | 67.9  | 68.2  | **0.3** | **0.6** | **0.4**    | **0.9**  | 67.2  | 67.9  | **0.7** | **0.3** | **1.0**    | **0.5**  |
> |        | CB         | 77.6  | 78.1  | **0.5** |         | **0.6**    |          | 83.9  | 84.2  | **0.3** |         | **0.4**    |          |
> |        | COPA       | 74.0  | 75.0  | **1.0** |         | **1.4**    |          | 84.0  | 84.2  | **0.2** |         | **0.2**    |          |
> |        | MultiRC    | 59.7  | 60.4  | **0.7** |         | **1.2**    |          | 63.8  | 64.1  | **0.3** |         | **0.5**    |          |
> |        | ReCoRD     | 69.8  | 70.2  | **0.4** |         | **0.6**    |          | 74.4  | 74.7  | **0.3** |         | **0.4**    |          |
> |        | WiC        | 61.4  | 62.0  | **0.6** |         | **1.0**    |          | 61.0  | 61.3  | **0.3** |         | **0.5**    |          |
> |        | WSC        | 64.4  | 65.1  | **0.7** |         | **1.1**    |          | 77.9  | 78.1  | **0.2** |         | **0.3**    |          |
> | NLI    | ANLIR1     | 31.5  | 32.1  | **0.6** | **0.5** | **1.9**    | **1.3**  | 37.8  | 38.2  | **0.4** | **0.3** | **1.1**    | **0.7**  |
> |        | ANLIR2     | 35.1  | 35.4  | **0.3** |         | **0.9**    |          | 37.9  | 38.3  | **0.4** |         | **1.1**    |          |
> |        | ANLIR3     | 37.1  | 37.8  | **0.7** |         | **1.9**    |          | 40.9  | 41.1  | **0.2** |         | **0.5**    |          |
> |        | StoryCloze | 79.0  | 79.5  | **0.5** |         | **0.6**    |          | 87.8  | 87.9  | **0.1** |         | **0.1**    |          |
> | CLS    | Amazon     | 65.2  | 67.7  | **2.5** | **2.3** | **3.8**    | **3.3**  | 68.2  | 69.4  | **1.2** | **1.2** | **1.8**    | **1.6**  |
> |        | DBPedia    | 70.5  | 72.5  | **2.0** |         | **2.8**    |          | 83.9  | 85.1  | **1.2** |         | **1.4**    |          |
> | QA     | DROP       | 67.9  | 69.9  | **2.0** | **1.8** | **2.9**    | **5.6**  | 51.6  | 52.8  | **1.2** | **1.0** | **2.3**    | **3.2**  |
> |        | NQ         | 15.1  | 16.1  | **1.0** |         | **6.6**    |          | 19.6  | 20.8  | **1.2** |         | **6.1**    |          |
> |        | RealTimeQA | 29.0  | 31.5  | **2.5** |         | **8.6**    |          | 36.0  | 37.2  | **1.2** |         | **3.3**    |          |
> |        | WebQs      | 34.8  | 36.3  | **1.5** |         | **4.3**    |          | 44.1  | 44.6  | **0.5** |         | **1.1**    |          |
>
> Table 5: Zero-shot domain transfer performance based on LLaMA with original and MAPO-Optimized prompts. CLS: Classification. The (↑ ) and ↑represent the absolute improvement scores of MAPO-optimized prompts compared with original prompts in each dataset and task, respectively. The (↑(%) ) and ↑(%) represent the relative improvement percentages of MAPO-optimized prompts compared with original prompts in each dataset and task, respectively.
> | Task | Dataset    | LLaMA | +MAPO | **(↑)** | **↑**   | **(↑(%))** | **↑(%)** |
> |------|------------|-------|-------|---------|---------|------------|----------|
> | RS   | BoolQ      | 76.5  | 76.7  | **0.2** | **0.3** | **0.3**    | **0.5**  |
> |      | PIQA       | 79.8  | 80.0  | **0.2** |         | **0.3**    |          |
> |      | SIQA       | 48.9  | 49.0  | **0.1** |         | **0.2**    |          |
> |      | HellaSwag  | 76.1  | 76.5  | **0.4** |         | **0.5**    |          |
> |      | WinoGrande | 70.1  | 70.5  | **0.4** |         | **0.6**    |          |
> |      | ARC-e      | 72.8  | 73.2  | **0.4** |         | **0.6**    |          |
> |      | ARC-c      | 47.6  | 47.8  | **0.2** |         | **0.4**    |          |
> |      | OBQA       | 57.2  | 57.9  | **0.7** |         | **1.2**    |          |
> | QA   | NQ         | 16.8  | 18.1  | **1.3** | **1.1** | **7.7**    | **4.7**  |
> |      | RACE       | 50.0  | 50.8  | **0.8** |         | **1.6**    |          |
>
>
> Due to the extensive length of the revised tables, we have divided them according to each specific question. In the final version, we will incorporate both absolute improvement scores and relative improvement percentages for the "+SFT" results. Additionally, a figure illustrating the outcomes for various temperature settings will also be included in the final version.
>
> Thank you very much for your valuable suggestions and comments on our manuscript. We hope that our responses have addressed your concerns. We would be most grateful if you could consider **raising your score** in light of these clarifications. Thank you once again for your time and consideration!

---

### Meta-Review · Area_Chair_7d5j · 2023-09-19

**Recommendation:** 3

**Metareview:**

This paper studies an interesting problem. It first shows the performance variations across large language models (LLMs) given the same set of prompts. It then adaptively optimizes the prompt for each LLM given a specific downstream task. The proposed method, MAPO, mainly consists of an SFT stage and an RL stage. The RL stage is composed of three main components: PPO, SFT approximation and maintaining generalization. The proposed method yields promising improvements.

As noted by the reviewers, the original submission had quite a few omissions such as ablations (SFT, temperature variations) as well as confusions such as the use of "Human feedback" while they actually just do model feedback; "most effective candidate prompts that will produce outputs closely aligned with the ground-truth outputs, not with the outputs generated by the original prompt as initially stated", etc. The authors have provided the required information in their detailed rebuttal. These should be added in the main paper / supplementary.

---

### Decision · Program_Chairs · 2023-10-07

**Decision:**

Accept-Findings

**Comment:**

This paper studies an interesting problem. It first shows the performance variations across large language models (LLMs) given the same set of prompts. It then adaptively optimizes the prompt for each LLM given a specific downstream task. The proposed method, MAPO, mainly consists of an SFT stage and an RL stage. The RL stage is composed of three main components: PPO, SFT approximation and maintaining generalization. The proposed method yields promising improvements.

As noted by the reviewers, the original submission had quite a few omissions such as ablations (SFT, temperature variations) as well as confusions such as the use of "Human feedback" while they actually just do model feedback; "most effective candidate prompts that will produce outputs closely aligned with the ground-truth outputs, not with the outputs generated by the original prompt as initially stated", etc. The authors have provided the required information in their detailed rebuttal. These should be added in the main paper / supplementary.